# Regret Minimization in Linear Bandits with Offline Data via Extended D-optimal Exploration.

**Sushant Vijayan** *
*School of Technology and Computer Science*
*TIFR, Mumbai*

*sushant.vijayanq@tifr.res.in*

**Arun Suggala**
*Google Deepmind*

*arunss@google.com*

**Karthikeyan Shanmugam**
*Google Deepmind*

*karthikeyanvs@google.com*

**Soumyabrata Pal**
*Adobe Research*

*soumyabratap@adobe.com*

Reviewed on OpenReview: *https://openreview.net/forum?id=4WcK8gKgCi*

## Abstract

We consider the problem of online regret minimization in stochastic linear bandits with access to prior observations (*i.e.,* offline data) from the underlying bandit model. This setting is highly relevant to numerous applications where extensive offline data is often available, such as in recommendation systems, personalized healthcare, and online advertising. Consequently, this problem has been studied intensively in recent works such as Banerjee et al. (2022); Wagenmaker & Pacchiano (2022); Agrawal et al. (2023); Hao et al. (2023); Cheung & Lyu (2024). We introduce the Offline-Online Phased Elimination (`OOPE`) algorithm, that effectively incorporates the offline data to substantially reduce the online regret compared to prior work. To leverage offline information prudently, `OOPE` uses an extended D-optimal design within each exploration phase. We show that `OOPE` achieves an online regret is $\tilde{O}(\sqrt{d_{\text{eff}}T \log(|\mathcal{A}|T)} + d^2)$, where $\mathcal{A}$ is the action set, $d$ is the dimension and $T$ is the online horizon. $d_{\text{eff}}$ $(\leq d)$ is the *effective problem dimension* which measures the number of poorly explored directions in offline data and depends on the eigen-spectrum $(\lambda_k)_{k \in [d]}$ of the Gram matrix of the offline data. Thus the eigen-spectrum $(\lambda_k)_{k \in [d]}$ is a quantitative measure of the *quality* of offline data. If the offline data is poorly explored ($d_{\text{eff}} \approx d$), we recover the established regret bounds for purely online linear bandits. Conversely, when offline data is abundant ($T_{\text{off}} \gg T$) and well-explored ($d_{\text{eff}} = o(1)$), the online regret reduces substantially. Additionally, we provide the first known minimax regret lower bounds in this setting that depend explicitly on the quality of the offline data. These lower bounds establish the optimality of our algorithm [1] in regimes where offline data is either well-explored or poorly explored. Finally, by using a Frank-Wolfe approximation to the extended optimal design we further improve the $O(d^2)$ term to $O\left(\frac{d^2}{d_{\text{eff}}} \min\{d_{\text{eff}}, 1\}\right)$, which can be substantial in high dimensions with moderate quality of offline data $d_{\text{eff}} = \Omega(1)$.

## 1 Introduction

Since the seminal works of Thompson (1933); Robbins (1952); Lai et al. (1985), bandit optimization has been extensively studied (Cesa-Bianchi & Lugosi, 2006; Gittins et al., 2011; Agrawal & Goyal, 2012; Lattimore

---

*Work done as student researcher at Google DeepMind, Bengaluru.
[1]Optimal within log factors in $T, T_{\text{off}}$ and additive constants in $d$

& Szepesvari, 2017; Lattimore & Szepesvári, 2020), and has found numerous applications in areas such as personalized medicine (Lu et al., 2021), recommendation systems (Li et al., 2010), and hyper-parameter tuning in ML (Golovin et al., 2017). In the typical bandit setting, the learner faces an unknown environment. It can repeatedly interact with the environment by taking an action and receives some noisy feedback. The goal of the learner is to converge to an optimal action with largest reward, as quickly as possible. However, as the environment is unknown, it leads to an exploration-exploitation trade-off where the learner has to get a good balance between exploiting the best actions found till now, and trying out, potentially better rewarding unexplored actions. Numerous algorithms have been proposed for achieving the right balance between exploration-exploitation; some of these include Thompson Sampling (TS) (Thompson, 1933; Agrawal & Goyal, 2012), Upper Confidence Bound (UCB) (Auer et al., 2002), Phased Elimination (PE) algorithms (Valko et al., 2014; Lattimore & Szepesvári, 2020).

Despite the wide applicability of bandit optimization algorithms, they require significant exploration to figure out the optimal action. This limits their direct application in domains like healthcare and education, where exploration carries cost or ethical concerns (Kapp, 2006). For instance, consider mobile health interventions involving health apps and wearables, where one wants to identify optimal nudges for promoting physical activity. Since human subjects are involved, federal regulations constrain the extent of experimentation. However, the learner can access relevant prior data in the form of offline logs in these settings. Number of works like (Banerjee et al., 2022; Oetomo et al., 2023; Wagenmaker & Pacchiano, 2022; Agrawal et al., 2023; Hao et al., 2023; Cheung & Lyu, 2024) propose algorithms that leverage such prior data to reduce the need for online exploration, facilitating faster convergence to optimal policies.

We consider the setting of linear stochastic bandits, where the expected reward of an action $a \in \mathcal{A}$ is a linear function of the action's feature vector: $\mathbb{E}[r_a] = \langle a, \theta \rangle$. Here $\theta \in \mathbb{R}^d$ is the unknown parameter vector, and with a slight overload of notation we let '$a$' represent the feature vector of arm $a$. The learner is given access to $T_{\text{off}}$ many offline observations generated by a *non-adaptive* policy $\pi_{\text{off}}$ which pulls action $a$ with probability $\pi_{\text{off}}(a)$, and observes a noisy reward with mean $\langle a, \theta \rangle$. The learner is allowed to perform $T$ rounds of online interaction, with the goal of maximizing the cumulative rewards collected over these $T$ rounds. Two natural questions that arise are:

*Under what conditions does offline data contribute to a significant reduction in regret?*
*Can we design optimal algorithms for this setting?*

Prior research has explored this problem predominantly in the context of Multi-Armed Bandits (MABs), which is a special case of linear bandits (Shivaswamy & Joachims, 2012; Banerjee et al., 2022; Hao et al., 2023; Cheung & Lyu, 2024). Techniques such as warm-started UCB and $\epsilon$-greedy algorithms (Shivaswamy & Joachims, 2012; Cheung & Lyu, 2024), and the artificial-replay meta-algorithm that simulates online actions using historical data (Banerjee et al., 2022) have been proposed. However, the direct extension of these approaches to the linear bandit setting presents challenges. Existing warm-started LinUCB (Linear UCB) analysis yields sub-optimal regret guarantees (see Section A.10). In Banerjee et al. (2022) Artificial Replay was shown to have regret bounds equivalent to those of warm started LinUCB and LinTS (Linear algorithms and hence also sub-optimal in our settings. Hao et al. (2023) and Oetomo et al. (2023) proposed a warm-started TS procedure for MABs and linear bandits respectively, in the presence of historic data. However, TS is known to achieve a worst-case regret of $\Omega(d^{3/2}\sqrt{T})$ even in the purely online setting, which is sub-optimal (Hamidi & Bayati, 2020). Finally, we note that all these works have focused solely on regret upper bounds (except for MABs in Cheung & Lyu (2024)), with a notable absence of lower bound analyses. Thus, designing provably near *optimal* algorithms for linear bandits with access to offline data remains a significant open question.

**Current work**: We introduce the Offline-Online Phased Elimination (`OOPE`) algorithm for regret minimization in linear bandits in the online phase with access to offline data (we will hereafter denote this as `OO` setting[2]). `OOPE` is based on the phased elimination algorithm in the purely online setting (see Valko et al. (2014), Chapter 22 in Lattimore & Szepesvári (2020)). The algorithm operates in phases whose lengths increase exponentially. Each phase consists of two distinct components: (a) *Exploration:* an exploratory policy is chosen to maximize the information gain about $\theta$ and (b) *Exploitation:* arms deemed suboptimal

---

[2] `OO` refers to Online with Offline setting

according to the current estimate of the unknown parameter $\theta$ are eliminated. The rejection threshold geometrically decreases in each phase. In this way, `OOPE` eliminates arms whose sub-optimality gaps becomes progressively smaller with each phase.

The underlying design principle of Phased Elimination is well known, the key challenge is to effectively utilize the offline data in the exploration phases. We use a simple experimental design (see equation (2)) that has a weighted mixture of the offline Grammian matrix with the Grammian matrix obtained from the online samples of that phase. This novel design is a generalization of the classical D-optimal design (see (Kiefer, 1960; Pukelsheim, 2006)).

We make two crucial design choices : (a) `OOPE` only incorporates previously unseen offline data in each new exploration phase. This maintains statistical independence of a given exploration phase from others and is important in obtaining *concentration* guarantees that underpins `OOPE`'s regret bounds and (b) we carefully choose the weighting $\alpha_l$ of the offline Grammian matrix. If the weighting $\alpha_l$ is too high, we unnecessarily use too many offline samples in that phase leading to higher number of online samples (hence, increased regret) used in later phases when the offline samples are exhausted. On the other hand, if the current $\alpha_l$ is too low, the online samples increases in the current phase when the live arms are quite sub-optimal and could contribute to the regret. The weighting in the experimental design must carefully balance this trade-off between the regret in the current phase with potential regret in future phases in order to obtain provably optimal regret guarantees.

**Contributions**: The following are our main contributions to regret minimization in linear bandits for the `OO` setting :

- We propose a phased elimination algorithm `OOPE` that incorporates offline data in its exploration through a novel extended D-optimal design. We establish a worst case regret bound of $\tilde{O}\left(\sqrt{d_{\text{eff}}T\log\left(|\mathcal{A}|T\right)} + d^2\right)$ that varies depending on the *quality* of the offline data. Our bounds depend on the *effective dimension* $d_{\text{eff}}$, which measures the problem's difficulty given an offline dataset. This result also shows that a quantitative measure of the offline data quality is given by the eigen-spectrum of the offline Gram matrix and highlights that regret reduction depends strongly on the quality of offline data available.

- We introduce the notion of minimax regret for problem classes defined by the normalized eigenspectrum of the offline Gram matrix and provide corresponding lower bounds for it. These lower bounds establish the optimality [3] of `OOPE` in well-explored and poorly-explored offline data regimes. To the best of our knowledge, this is the first lower bound that specifically incorporates the quality of the offline data through its eigen-spectrum in `OO` setting for linear bandits.

- In high-dimensional settings (e.g., $d = \Omega(\sqrt{T})$), the $d^2$ term in the `OOPE` regret can be dominant. To improve the regret in such settings, we introduce `OOPE-FW` which relies on Frank-Wolfe (FW) (Jaggi, 2013) to provide an approximate solution to the extended D-optimal design. This leads to an approximately optimal design with much smaller support than the $O(d^2)$ support in `OOPE` and a worst-case regret bound of `OOPE-FW` of $\tilde{O}\left(\sqrt{d_{\text{eff}}T\log\left(|\mathcal{A}|T\right)} + \frac{d^2}{d_{\text{eff}}}\right)$. This is an improvement over the regret of `OOPE` when $d_{\text{eff}} = \Omega(1)$.

In proving these guarantees, we overcome a number of technical issues that arise as a result of incorporating the offline data in the `OO` setting :

(a) *Deriving tight bounds on confidence width in extended D-optimal design:* The width of confidence sets plays a fundamental role in the regret analysis of bandit algorithms (Abbasi-Yadkori et al., 2011; Russo & Van Roy, 2013; Lattimore & Szepesvári, 2020). In contrast to the purely online setting where the Kiefer-Wolfowitz theorem (see Kiefer (1960)) provides an exact expression, the extended D-optimal design incorporating offline data lacks a straightforward analytical expression for the confidence width. To address this, we provide a tight characterisation of the confidence width of extended D-optimal design in terms of the offline eigen-spectrum, and confidence widths of the offline data (see Lemma 4.4).

---

[3]up to log factors in $T, T_{\text{off}}$ and an additive constant in $d$.

(b) *Incorporating quality of offline data into lower bounds*: The usual minimax lower bound in linear bandits (see section 24.1 in Lattimore & Szepesvári (2020)) utilizes an action set construction on the d-dimensional hypercube to derive the $d\sqrt{T}$ lower bound. However, a straightforward extension to the current setting gives a weaker lower bound that does not capture the quality of the offline data through the Grammian eigen-spectrum. We instead come up with a new construction in which we perturb the hypercube as a function of the offline proportions $\pi_{\text{off}}$ so that the Grammian eigenvectors align with the standard basis. This enables us to derive a lower bound in terms of the offline Grammian eigen-spectrum.

(c) *Complicated dual of extended D-optimal design for Frank-Wolfe analysis:* Effective initialization of the Frank-Wolfe (FW) subroutine within `OOPE-FW` is essential for achieving its improved regret guarantees. In the purely online setting, the typical analysis (see (Todd, 2016)) of FW uses this initialization along with properties of the dual problem - a Minimum Volume Enclosing Ellipsoid (MVEE) [4] problem. A challenge in extending this analysis to the `OO` setting is that the corresponding dual constraints are no longer of an enclosing ellipsoid. To tackle this, we come up with a novel feasibility relation (Lemma B.7) for relating the dual problem to the usual MVEE problem.

**Organization of the paper.** In section 2 we provide an in-depth review of previous work relevant to our problem setting in the literature. We formally introduce the problem setting, notations, and relevant background in Section 3. Section 4 presents the `OOPE` algorithm, and derives its regret guarantee, along with matching lower bounds. To address high-dimensional scenarios, Section 5 presents `OOPE-FW`, along with a theoretical analysis yielding improved regret bounds. Empirical validations of the theoretical results are presented in Section 6. Comprehensive proofs for the stated theorems and other supplemental results are presented in the Appendix.

## 2   Related Work

Regret minimization in the `OO` setting for linear bandits is closely related to a number of other problems studied in the literature. Below, we provide a non-exhaustive review of research works that are relevant to the problem we consider.

**Regret Minimization in Online Linear Bandits.**   Dani et al. (2008) provided one of the earliest known regret bounds of $\tilde{O}(d\sqrt{T})$ for purely online regret minimization without any offline data. For the same problem, Abbasi-Yadkori et al. (2011) developed the OFUL algorithm which again achieves $\tilde{O}(d\sqrt{T})$ regret, but with improved log dependence. Although optimal for exponentially sized arm sets ($|\mathcal{A}| = \Omega(2^d)$), these regret bounds are sub-optimal in the sub-exponential arm setting. Subsequent research has developed algorithms to address this drawback. The Phased Elimination algorithm achieves a regret of $O(\sqrt{dT\log(|\mathcal{A}|T)} + d^2)$ (Lattimore & Szepesvári, 2020) which only has $\log(|\mathcal{A}|)$ dependence. Our `OOPE` algorithm is based on phased elimination, which has better regret guarantees than UCB style algorithms in the sub-exponential arm setting. When the set of arms is dynamic and the losses chosen by an adversary, Chu et al. (2011) proposed the SUPLINUCB algorithm which achieves $O(\sqrt{dT\log^3|\mathcal{A}|T})$ regret. The EXP2 algorithm (with John's exploration) of Bubeck et al. (2012) achieves a regret of $O(\sqrt{dT\log|\mathcal{A}|T})$, for $T \geq d^2$. Li et al. (2019) improved the analysis of SUPLINUCB and provided nearly matching minimax lower bounds.

**Incorporating Offline Data into Bandits and Reinforcement Learning.** Several works have investigated regret minimization in `OO` setting (Shivaswamy & Joachims, 2012; Banerjee et al., 2022; Hao et al., 2023; Cheung & Lyu, 2024) for MABs. Most of them study warm started algorithms like Upper Confidence Bound (UCB), Thompson Sampling (TS) and $\epsilon$-greedy. Banerjee et al. (2022) proposed artificial-replay algorithm that is shown to attain similar bounds like the warm-started approaches mentioned above. Bayesian regret bounds were obtained for warm started TS by Hao et al. (2023) in the MAB setting. They assume a particular stationary offline data generation that is subsumed in our framework. Furthermore, their regret does not vanish as the quantity and quality of offline data increase. Cheung & Lyu (2024) provide a lower bound which helps establish minimax optimality of the warm-started UCB in the `OO` setting for MABs. A similar lower bound for MABs is reported in Sentenac et al. (2025), where, in addition to the online regret,

---

[4]see Lemma 5.1 for a precise definition of MVEE

the proposed algorithm (a mix of UCB for online settings and LCB for offline settings) also maintains sub-linear regret with the offline data generation policy. Oetomo et al. (2023) considered warm-started approach to linear bandits directly and provided regret upper bounds but no lower bounds. We provide an example which shows (section A.10 and lower bound results for TS of Hamidi & Bayati (2020)) that warm-starting approaches with typical regret analysis in linear bandits can have sub-optimal dimension factors in the important case of well-explored and plentiful offline data. Cai et al. (2022) and Kausik et al. (2024) consider regret minimization in contextual bandits for the MAB and linear bandit settings respectively. Cai et al. (2022) studied the problem under a potential co-variate shift in the offline data and provided an algorithm that achieves the minimax optimal regret guarantees. This work, when specialized to the case of MAB in OO setting gives rates which are only tight for well explored offline data. Kausik et al. (2024) studied the problem with a latent linear bandit setting and provide regret bounds that have improved dimensionality factors but the regret bound does not vanish with increasing quantity and quality of offline data. Warm-starting contextual bandits with potentially shifted data was considered in Zhang et al. (2019). However, as shown in Appendix A.1 of Cheung & Lyu (2024) the proposed algorithm in Zhang et al. (2019) can have regret of $\tilde{O}(T^{2/3})$ in the OO setting. Agrawal et al. (2023) developed algorithms for Best Arm Identification (BAI) in fixed confidence setting with offline data for MABs while Yang et al. (2025) studied the same problem with a potentially biased offline data.

The problem of leveraging offline data to improve the performance in the online phase has also been studied intensely in the Reinforcement Learning (RL) literature in the past few years under the name of hybrid-RL. Xie et al. (2021); Song et al. (2022); Wagenmaker & Pacchiano (2022); Ball et al. (2023); Li et al. (2023) used offline data for finding the optimal policy in online RL using as few samples as possible (similar to BAI in fixed confidence setting). In contrast, Zhou et al. (2023) derive high probability regret bounds under boundedness assumptions on coverage, transfer and concentration coefficients that are hard to verify in practice. Bu et al. (2020) analyzed the related dynamic pricing problem with offline data using a stylized demand model. Agnihotri et al. (2024), a follow-up work of Hao et al. (2023), studies regret minimization where the offline data contains user preferences rather than reward feedback. They analyze a warm started posterior sampling and provide its Bayesian regret.

**Offline Learning.** In this problem, the learner is given an offline dataset consisting of trajectories of a Markov Decision Process (MDP) and has to identify the optimal policy with high accuracy. Unlike the OO setting, in offline learning there is no adaptive online learning phase. The goal is to optimize identification measures like simple regret and error probability of misidentifying the optimal policy. A number of works like Rajaraman et al. (2020); Rashidinejad et al. (2021); Xiao et al. (2021); Cheng et al. (2022) have shown that pessimistic algorithms (like Lower Confidence Bound (LCB)) have good guarantees in this setting.

**Online Learning with Advice.** Many works have considered online learning with additional information [5] in the full (for e.g., Rakhlin & Sridharan (2013); Steinhardt & Liang (2014)) and bandit (for e.g., Mannor & Shamir (2011); Wei & Luo (2018); Wei et al. (2020); Sharma et al. (2020); Tennenholtz et al. (2021); Cutkosky et al. (2022)) feedback settings respectively. In this problem before each arm-pull, the learner also has additional information about the unknown system that can be incorporated into its decision. The regret guarantees in the above works are typically weaker than prior work in OO setting (like Shivaswamy & Joachims (2012); Cheung & Lyu (2024); Sentenac et al. (2025)) when the offline data is well-explored and plentiful[6]. This is due to the additional information typically being less informative than good quality offline data.

**Multi-task Bandit Learning.** This is a related line of work on solving multiple bandit problems jointly by aggregating observed rewards where the underlying unknown parameters are shared or are similar in certain statistical sense. Wang et al. (2021) solved the multi-armed bandit problem where the arm means are close. Gentile et al. (2014); Korda et al. (2016) extended the problem to linear bandits and assume a cluster structure on parameters. Lazaric et al. (2013); Soare et al. (2014); Osadchiy et al. (2022); Balcan et al. (2022) considered a sequential arrival of tasks in the context of multi-armed bandits and linear bandits.

---

[5]variously called as advice, side-information or hints in the literature.

[6]Roughly, this is what we mean by *good quality.* offline data

For a particular task, the samples from previous tasks can be represented as offline data but note that all samples have the flexibility of being chosen by the agent. In contrast, in our setting, we do not have any control over how the offline samples are being generated.

## 3 Preliminaries and Problem Formulation

We consider a linear bandit setting with a finite set of arms $\mathcal{A} \subset \mathbb{R}^d$, where $|\mathcal{A}| < \infty$. The learner knows the set $\mathcal{A}$. We make the following assumption on $\mathcal{A}$:

**Assumption 3.1.** *The set of arms $\mathcal{A}$ spans the $d$-dimensional Euclidean space, i.e.,* $\text{span}(\mathcal{A}) = \mathbb{R}^d$.

This assumption is made for simplicity and ease of exposition. It can be removed by performing the analysis within the subspace $\text{span}(\mathcal{A})$. When arm $a_t \in \mathcal{A}$ is sampled at time step $t$, we observe a reward $y_{a_t} = \langle \theta^*, a_t \rangle + \eta_{a_t,t}$. Here, $\theta^* \in \mathbb{R}^d$ is the unknown parameter vector of the bandit model, and $\eta_{a_t,t}$ is a zero-mean, 1-sub-gaussian real-valued random variable, drawn independently of past actions and rewards. Although presented for 1-sub-Gaussian noise, our results can be straightforwardly generalized to $\sigma$-sub-Gaussian noise. We define $a^* := \text{argmax}_{a \in \mathcal{A}} \langle \theta^*, a \rangle$ as the arm with the highest expected reward (ties are broken arbitrarily).

In the offline-online (OO) setting, the learner has access to $T_{\text{off}}$ offline samples, denoted as $\mathcal{D}_{\text{off}} = \{(a_t, y_{a_t})\}_{t=1}^{T_{\text{off}}}$, from the bandit model. Let $\pi_{\text{off}}(a)$ be the fraction of offline samples drawn from arm $a$; thus, $\sum_{a \in \mathcal{A}} \pi_{\text{off}}(a) = 1$, and the number of offline samples from arm $a$ is $\pi_{\text{off}}(a)T_{\text{off}}$.

**Assumption 3.2.** *The reward-generating distribution for each arm remains unchanged between the offline and online stages.*

The lack of *drift* between distributions in offline and online phase models, situations where the offline data is quite pertinent to the online phase. We leave the study of the more general case of known bounded *drift* between offline and online phases to future work.

We make the following assumption about how $\mathcal{D}_{\text{off}}$ was collected:

**Assumption 3.3.** *The offline data $\mathcal{D}_{off}$ was collected using a fixed design $\pi_{off}$ over the arms $\mathcal{A}$.*

If an arbitrary adaptive policy were used to collect offline data, it would be impossible to analyze the online regret without additional information on the specific offline policy. Therefore, we restrict our analysis to fixed-design offline data collection. This fixed design setting has not been previously studied, is non-trivial and practically important: In many scenarios, a fixed pilot design is initially used, and a need later arises to switch to an adaptive strategy while still leveraging the previously collected data. This condition has also been assumed in several related previous works like [Shivaswamy & Joachims (2012); Banerjee et al. (2022); Hao et al. (2023); Cheung & Lyu (2024)].

The learner interacts with the bandit model for an additional $T$ online rounds. The learner's goal is to choose arms $a_t$ at each online round $t \in \{1, \ldots, T\}$ to minimize the expected online regret, defined as:

$$\mathcal{R}(T, T_{\text{off}}, \mathcal{A}, \pi_{\text{off}}, \text{Alg}) := T\langle \theta^*, a^* \rangle - \sum_{t=1}^{T} \mathbb{E}[\langle \theta^*, a_t \rangle].$$

The expectation $\mathbb{E}[\cdot]$ is over the realizations of the offline and online noise, as well as any randomness in the learning algorithm. For ease of exposition, we will often write the online regret of an Algorithm Alg as $\mathcal{R}(\text{Alg})$ and suppress the dependence on $T, T_{\text{off}}, \mathcal{A}, \pi_{\text{off}}$.

The main difference from the standard formulation of regret minimization is the availability of offline samples. We refer to the setting where $T_{\text{off}} = 0$ as the *pure online* case. We make the following technical assumption:

**Assumption 3.4.** For every $a \in supp(\pi_{\text{off}}) \subset \mathcal{A}$, we have:

$$\pi_{\text{off}}(a)T_{\text{off}} \geq 3 \max \left\{ \lfloor \log_2 \sqrt{(T + T_{\text{off}})/4d \log(4|\mathcal{A}T|)} + 1 \rfloor, \lfloor \log_2 \sqrt{T_{\text{off}}/4 \log(4|\mathcal{A}T|)} + 1 \rfloor \right\}.$$

The assumption states that all arms in the support of $\pi_{\text{off}}$ has at least $\tilde{\Omega}(\log(\sqrt{(T + T_{\text{off}})/d}))$ samples. This technical assumption is required to ensure that our algorithm does not request excessive offline samples (see

Proposition 4.3). In the regime where $T_{\text{off}} >> d^d$ this assumption can be satisfied by incorporating only those arms which have sufficient offline data while minimally perturbing $\pi_{\text{off}}$. Another way to relax this assumption is to run an initial exploration phase where arms are pulled so that the assumption is satisfied at the cost of an additional $\tilde{O}(|\text{supp}(\pi_{\text{off}})| \log(\sqrt{(T + T_{\text{off}})/d}))$ regret.

**Notation:** The probability simplex over $\mathcal{A}$ is denoted by $\Delta(\mathcal{A})$. For any $\pi \in \Delta(\mathcal{A})$, $\pi(a)$ is the probability assigned to arm $a \in \mathcal{A}$. The norm induced by a positive definite (PD) matrix $B$ on a vector $x \in \mathbb{R}^d$ is $||x||_B := \sqrt{x^T B x}$. The standard partial ordering for symmetric matrices by the non-negative orthant is denoted by $\succeq$. For a design $\pi \in \Delta(\mathcal{A})$, we define the matrix $V_\pi := \sum_{a \in \mathcal{A}} \pi(a) a a^T$, which is a positive semi-definite (PSD) matrix. $\lambda_k(B)$ denotes the $k$-th smallest eigenvalue of a matrix $B$. $Tr(B)$ and $det(B)$ denote the trace and determinant of a matrix $B$, respectively. Recall that $\pi_{\text{off}} \in \Delta(\mathcal{A})$ is the offline arm pull proportion, and $\mathcal{D}_{\text{off}}$ represents the offline samples. For a design $\pi \in \Delta(\mathcal{A})$ and a subset of arms $\mathcal{B} \subseteq \mathcal{A}$, define $g_{\mathcal{B}}(\pi) := \max_{a \in \mathcal{B}} ||a||^2_{V_\pi^{-1}}$. This quantity is related to the maximum variance of the Ordinary Least Squares (OLS) estimator constructed using design $\pi$ [7]. The sub-optimality gap for an arm $a$ is $\Delta_a := \langle \theta^*, a^* \rangle - \langle \theta^*, a \rangle$. Let $\mathcal{A}^*$ be the set of optimal arms. We define $\Delta_{\max} := \max_{a \in \mathcal{A} \backslash \mathcal{A}^*} \Delta_a$ and $\Delta_{\min} := \min_{a \in \mathcal{A} \backslash \mathcal{A}^*} \Delta_a$ as the maximum and minimum sub-optimality gap respectively. For a positive definite matrix $H$ and a positive constant $c$, the ellipsoid $\xi(H, c)$ is defined as $\xi(H, c) := \{x \in \mathbb{R}^d \mid x^t H x \leq c\}$. $vol(S)$ and $conv(S)$ denote the volume and convex hull of a set $S$, respectively.

**Optimal Design Background:** We employ criteria from the theory of optimal experimental design (Pukelsheim, 2006; Fedorov, 2013) to develop our exploration schedule. Formally, a design $\pi^* \in \Delta(\mathcal{A})$ is *D-optimal* if $\pi^* \in \underset{\pi \in \Delta(\mathcal{A})}{\arg\max} \log(\det(V_\pi))$. Similarly, a design $\pi^*$ is *G-optimal* if $\pi^* \in \underset{\pi \in \Delta(\mathcal{A})}{\arg\min} g_{\mathcal{A}}(\pi)$. The Kiefer-Wolfowitz (KW) theorem (Kiefer, 1960) remarkably establishes that D-optimal and G-optimal designs share the same optimizing design $\pi^*$, and for such an optimal design $\pi^*$, $g_{\mathcal{A}}(\pi^*) = d$. Furthermore, one can show that the dual of D-optimal design optimization is related to finding the Minimum Volume Enclosing Ellipsoid (MVEE) for the set of arms $\mathcal{A}$ (Titterington, 1975).

## 4 Offline-Online Phased Elimination (OOPE)

Our proposed algorithm OOPE proceeds in distinct phases, each requiring geometrically increasing number of offline and online samples. Each phase consists of two parts: an *exploration part*, in which we carefully sample arms based on a well chosen design and an *elimination part* where we eliminate sub-optimal arms based on the observed rewards in the exploration part. At the end of the each phase we maintain a set of "live" arms $\mathcal{A}_l$, that is, arms that have survived elimination (see line 27 of Algorithm 1). During the exploration part of a phase $l$, $n^l_{\text{on}}(a)$ new online samples are collected from each arm $a \in \mathcal{A}_l$ to gain more information about $\theta^*$. Further we augment these samples that have been gathered online with additional $n^l_{\text{off}}(a)$ samples. After the phase $l$ exploration ends, we compute an OLS estimate $\hat{\theta}_l$ based on the samples $n^l_{\text{on}}(a)$, $n^l_{\text{off}}(a)$. Then, in the elimination part, all the live arms $a \in \mathcal{A}_l$ that are estimated to have sub-optimality gaps greater than $2\epsilon_l$ with respect to the estimate $\hat{\theta}_l$ are eliminated. Note, we do not re-use the same offline samples in multiple phases and this ensures statistical independence across phases. We continue this interleaved process of exploration and elimination till the exhaustion of online samples. Next, we introduce concepts important to our algorithm OOPE and give a precise description of OOPE.

**Effective dimension $d_{\text{eff}}$:** The *effective* dimension $d_{\text{eff}}$ is defined as:

$$d_{\text{eff}} := \min\left( \sum_{k=1}^{d} \frac{1}{1 + \frac{T_{\text{off}}}{T} \frac{\lambda_k(V_{\pi_{\text{off}}})}{\max_a ||a||^2}}, \frac{T}{T_{\text{off}}} g_{\mathcal{A}}(\pi_{\text{off}}) \right). \tag{1}$$

Note that $d_{\text{eff}} \leq d$. The effective dimension represents the remaining direction of $\theta^*$ left to explore after incorporating the offline data. The extent of exploration of each direction depends on the eigenvalues of $V_{\pi_{\text{off}}}$, the total number of offline samples $T_{\text{off}}$ and the offline confidence width $g_{\mathcal{A}}(\pi_{\text{off}})$. In the pure online

---

[7] see chapter 20 of Lattimore & Szepesvári (2020) for exact details.

---

**Algorithm 1** OFFLINE ONLINE PHASED ELIMINATION (`OOPE`).

---

1: **Input:** $T, \mathcal{A}, \mathcal{D}_{\text{off}}, \pi_{\text{off}}, T_{\text{off}}$.
2: $l \leftarrow 1$, $\mathcal{A}_l \leftarrow \mathcal{A}$, $s \leftarrow 0$, $N_{\text{off}} \leftarrow [0]^{|\mathcal{A}|}$, mid $\leftarrow$ True.
3: Compute $d_{\text{eff}}$ as in equation (1).
4: **while** $s < T$ **do**
5:     **if** $|\mathcal{A}_l| > 1$ **then**
6:         $\epsilon_l \leftarrow 2^{-l}$; $\alpha_l \leftarrow \frac{T_{\text{off}}}{T_{\text{off}}+T}$
7:         $\pi_{l,\text{on}}^* \leftarrow \underset{\pi \in \Delta(\mathcal{A}_l)}{\operatorname{argmax}} \log\left(\det\left(V_{(1-\alpha_l)\pi+\alpha_l \pi_{\text{off}}}\right)\right)$
8:         $\tilde{\pi}_l^\star \leftarrow (1-\alpha_l)\pi_{l,\text{on}}^* + \alpha_l \pi_{\text{off}}$;   $g(\tilde{\pi}_l^\star) \leftarrow \underset{a \in \mathcal{A}_l}{\max}||a||^2_{V_{\tilde{\pi}^*}^{-1}}$.
9:         **for** $a \in \mathcal{A}$ **do**
10:             $n_{\text{on}}^l(a) \leftarrow \left\lceil \frac{3 d_{\text{eff}} \pi_{l,on}^*(a) \log(4l^2|\mathcal{A}|T)}{\epsilon_l^2} \right\rceil$.
11:             $n_{\text{off}}^l(a) \leftarrow \left\lceil \frac{2\alpha_l \pi_{\text{off}}(a) g(\tilde{\pi}_l^\star) \log(4l^2|\mathcal{A}|T)}{\epsilon_l^2} \right\rceil$.
12:             **if** $s + n_{\text{on}}^l(a) <= T$ **then**
13:                 $s \leftarrow s + n_{\text{on}}^l(a)$; $Y_{\text{on},a}^l \leftarrow n_{\text{on}}^l(a)$ new samples from arm $a$.
14:             **else**
15:                 $n_{\text{on}}^l(a) \leftarrow T - s$; $s \leftarrow T$; $Y_{\text{on},a}^l \leftarrow n_{\text{on}}^l(a)$ new samples from arm $a$.
16:                 mid $\leftarrow$ False; **break**.          *// Breaks inner loop when online samples exhausted.*
17:             **end if**
18:             **if** $N_{\text{off}}(a) + n_{\text{off}}^l(a) <= \pi_{\text{off}}(a)T_{\text{off}}$ **then**
19:                 $N_{\text{off}}(a) \leftarrow N_{\text{off}}(a) + n_{\text{off}}^l(a)$
20:                 $Y_{\text{off},a}^l \leftarrow n_{\text{off}}^l(a)$ offline samples from arm $a$.
21:             **else**
22:                 $n_{\text{off}}^l(a) \leftarrow \pi_{\text{off}}(a)T_{\text{off}} - N_{\text{off}}(a)$; $N_{\text{off}}(a) \leftarrow \pi_{\text{off}}(a)T_{\text{off}}$.
23:                 $Y_{\text{off},a}^l \leftarrow n_{\text{off}}^l(a)$ offline samples from arm $a$. *// Ensures no arm requires excess offline samples.*
24:             **end if**
25:         **end for**               *// Elimination occurs in all but last phase.*
26:         **if** mid **then**
27:             $\hat{\theta}_l \leftarrow \left(\sum_a (n_{\text{off}}^l(a) + n_{\text{on}}^l(a))aa^t\right)^{-1}\left(\sum_a \left(\sum_{k=1}^{n_{\text{off},a}^l} Y_{\text{off},a,k}^l + \sum_{k=1}^{n_{\text{on},a}^l} Y_{\text{on},a,k}^l\right)a\right)$.
28:             $\mathcal{A}_{l+1} \leftarrow \mathcal{A}_l \setminus \{a \in \mathcal{A}_l : \underset{a' \in \mathcal{A}_l}{\max}\langle a' - a, \hat{\theta}_l\rangle \geq 2\epsilon_l\}$.
29:             $Y_{\text{off}} \leftarrow Y_{\text{off}} \setminus \{\cup_a Y_{\text{off},a}^l\}$, $l \leftarrow l+1$.
30:         **end if**
31:     **else**
32:         Pull arm $a \in \mathcal{A}_l$.
33:         $s \leftarrow s+1$.
34:     **end if**
35: **end while**

---

setting, we have that $d_{\text{eff}} = d$. `OOPE` estimates $d_{\text{eff}}$ by computing the eigenspectrum of $V_{\pi_{\text{off}}}$ and plugging the values of $T_{\text{off}}, T$ into the formula equation (1).

As an example, we set $\mathcal{A}$ as the standard orthonormal basis in $\mathbb{R}^d$ and $T_{\text{off}} >> T$. If we consider the uniform offline exploration $\pi_{\text{off}}(a) = \frac{1}{d}$, then we have $\forall k$, $\lambda_k = 1/d$, $g_{\mathcal{A}}(\pi_{\text{off}}) = d$ and $d_{\text{eff}} = \min(d^2 T/(T + T_{\text{off}}), dT/T_{\text{off}}) = dT/T_{\text{off}}$. Now consider, the offline design $\pi_{\text{off}}(a_0) = 1$ for some specific $a_0 \in \mathcal{A}$, then we have $0 = \lambda_1 = \cdots = \lambda_{d-1}$, $\lambda_d = 1$, $g_{\mathcal{A}}(\pi_{\text{off}}) = \infty$ and $d_{\text{eff}} = \min(d - 1 + T/(T + T_{\text{off}}), \infty) \approx d - 1$. In the uniform $\pi_{\text{off}}$ case, there is well-rounded exploration of all directions in the offline phase and hence $d_{\text{eff}}$ is quite small, while $\pi_{\text{off}} = \delta_{a_0}$ has $d - 1$ directions

left to be explored in the online phase.

**Online Exploration Policy:** In a phase $l$, we choose an exploration design $\pi^*_{l,\text{on}}$ which maximizes the information gain about the unknown parameter $\theta^*$. Mathematically, we have:

$$\pi^*_{l,\text{on}} \in \underset{\pi \in \Delta(\mathcal{A}_l)}{\arg\max} \log \left( \det \left( V_{(1-\alpha_l)\pi + \alpha_l \pi_{\text{off}}} \right) \right). \tag{2}$$

Here $\alpha_l \in [0,1]$ corresponds to the relative weight given to offline to online samples in phase $l$. This objective naturally extends the $D$-optimal design found in experimental design literature (Fedorov, 2013) to incorporate offline data. This optimization maybe interpreted as minimizing the volume of the confidence ellipsoid of an Ordinary Least Square (OLS) estimator for $\theta^*$ given the access to offline data. In the pure online setting, the optimization objective only considers online samples (*i.e.,* $\alpha_l = 0$).

**Online Arm Pulls:** In each exploration part, we pull the live arms according to the policy defined in equation (2). In particular, we sample each arm $a \in \mathcal{A}_l$, $n^l_{\text{on}}(a)$ times, so that we have an $\epsilon_l$ accurate estimate of the unknown parameter $\theta$:

$$n^l_{\text{on}}(a) \triangleq \left\lceil \frac{2(1-\alpha_l)\pi^*_{l,\text{on}}(a)g(\tilde{\pi}^\star_l)\log(4l^2|\mathcal{A}|T)}{\epsilon_l^2} \right\rceil. \tag{3}$$

Here, $g(\tilde{\pi}^\star_l) \triangleq g_{\mathcal{A}_l}((1-\alpha_l)\pi^*_{l,\text{on}} + \alpha_l\pi_{\text{off}})$ is the maximum confidence width obtained using a mixture of offline samples generated using $\pi_{\text{off}}$, and online samples generated using $\pi^*_{l,\text{on}}$ (recall $g_{\mathcal{B}}(\pi) \triangleq \max_{a \in \mathcal{B}} \|a\|^2_{V_\pi^{-1}}$). In order for us to prove the correctness of the OOPE we actually use slightly more online samples (since $(1-\alpha_l)g(\tilde{\pi}^\star_l) \le d_{\text{eff}}$, see Lemma 4.4.) :

$$n^l_{\text{on}}(a) \triangleq \left\lceil \frac{3d_{\text{eff}}\pi^*_{l,\text{on}}(a)\log(4l^2|\mathcal{A}|T)}{\epsilon_l^2} \right\rceil. \tag{4}$$

This slightly increased samples only increases regret bound by a small constant multiple but enables us to upper bound the maximum number of phases possible (Lemma 4.2).

**Offline Data Allocation:** In each phase $l$, we use $n^l_{\text{off}}(a)$ offline samples for arm $a \in \mathcal{A}_l$, where $n^l_{\text{off}}(a)$ is defined as:

$$n^l_{\text{off}}(a) \triangleq \left\lceil \frac{2\alpha_l\pi_{\text{off}}(a)g(\tilde{\pi}^\star_l)\log(4l^2|\mathcal{A}|T)}{\epsilon_l^2} \right\rceil. \tag{5}$$

We set $\alpha_l := \frac{T_{\text{off}}}{T_{\text{off}}+T}$. This choice of $\alpha_l$ is *crucial* for the success of the algorithm. If $\alpha_l$ is set too low then we would unnecessarily use excess online samples where the offline samples would have sufficed and incur regret. If $\alpha_l$ is set too high then we consume too much offline samples for a given confidence width and hence have to use too many online samples in the latter phases.

To better understand our choice of $\alpha_l$, consider the situation where $\epsilon_l = 2^{-l}$. Then the regret in each phase can be upper bounded by the confidence width times the total number of online samples in phase $l$: $(1-\alpha_l)2^l g(\tilde{\pi}^\star_l)$. Since $g(\tilde{\pi}^\star_l)$ doesn't have an analytical expression, we replace it with a simpler upper bound $\frac{g(\pi_{\text{off}})}{\alpha_l}$ (this follows from $V_{\tilde{\pi}^\star_l} \succeq \alpha_l V_{\pi_{\text{off}}}$, see Lemma 4.4). The optimal schedule $\alpha_l$ minimizes the cumulative regret upper bound $\sum_l (1-\alpha_l)2^l \frac{g(\pi_{\text{off}})}{\alpha_l}$, under the constraints that all the $T$ online samples and at most $T_{\text{off}}$ offline samples are utilized.

The following proposition shows that our choice of $\alpha_l$ is a simple and good approximation to the optimal solution for this constrained optimization problem. The proof of this proposition is given in Appendix A.1.

**Proposition 4.1.** *The value of the following constrained optimization*

$$\min_{\substack{l_M \in \mathbb{N} \\ \alpha_l \in [0,1] \\ 1 \leq l \leq l_M}} \quad \sum_{l=1}^{l_M} 2^l \frac{1 - \alpha_l}{\alpha_l}$$

$$s.t. \quad \sum_{l=1}^{l_M} (1 - \alpha_l) 2^{2l} \geq T \tag{6}$$

$$\sum_{l=1}^{l_M} \alpha_l 2^{2l} \leq T_{off}$$

*is $\Omega\left(\frac{T}{\sqrt{T + T_{off}}}\right)$. In the regime where $T_{off} > T$, the choice $\alpha_l = \frac{T_{off}}{T + T_{off}}$ and $l_M = \theta(\log_2 \sqrt{T + T_{off}})$ is feasible and has an objective value of $O\left(\frac{T}{\sqrt{T + T_{off}}}\right)$ for the optimization equation (6). Moreover the choice $\alpha_l = \frac{T_{off}}{T + T_{off}}$ and $l_M = \theta(\log_2 \sqrt{T + T_{off}})$ is optimal amongst schedules where $\alpha_l$ is held fixed for each phase $l \leq l_M$.*

The constraints in the optimization equation (6) are obtained by observing the fact that the number of offline samples $\sum_{a \in \mathcal{A}} n_{off}^l(a) \propto \alpha_l 2^{2l}$ (equation (5)) and online samples $\sum_{a \in \mathcal{A}_l} n_{on}^l(a) \propto (1 - \alpha_l) 2^{2l}$ (equation (3)) used in each phase $l$. We do not re-use the offline data across different exploration phases. This ensures statistical independence across phases and helps in obtaining concentration inequalities useful for our regret analysis. The above calculation roughly captures the regret bound calculation given the sample constraint on offline and online samples. However, a more precise analysis is required to prove the correctness of OOPE and provide tight regret bounds accounting for per-arm offline sample constraints, potential last phase incompleteness in online samples, no-exploration in certain directions in offline data and increased online sample usage as per equation (4).

**Elimination:** After the exploration part, we have the elimination part. We utilize the offline samples $n_{off}^l(a)$ and online samples $n_{on}^l(a)$ collected in the exploration part to construct an OLS estimator $\hat{\theta}_l$ for $\theta^*$ as follows:

$$\hat{\theta}_l = \left( \sum_a (n_{off}^l(a) + n_{on}^l(a)) a a^t \right)^{-1} \left( \sum_a \left( \sum_{k=1}^{n_{off}^l(a)} Y_{off,a,k}^l + \sum_{k=1}^{n_{on}^l(a)} Y_{on,a,k}^l \right) a \right)$$

Here $Y_{on,a,k}^l, Y_{off,a,k}^l$ represent the sample values of the offline and online samples respectively collected (see lines 13-22 in Algorithm 1) in the exploration part.

Then we eliminate those arms that are suboptimal wrt $\hat{\theta}_l$, i.e., eliminate arms $a$ which satisfy the inequality $\max_{a' \in \mathcal{A}_l} \langle a', \hat{\theta}_l \rangle \geq \langle a, \hat{\theta}_l \rangle + 2\epsilon_l$. The elimination threshold $\epsilon_l = 2^{-l}$ becomes more stringent for latter phases and consequently the samples considered in each phase $l$ increases exponentially. Intuitively, it is because one requires more samples to reject arms with smaller sub-optimality gaps. The elimination part is only executed in all phases except the last one (see line 25 in Algorithm 1). This is designed so because there is no need to further prune live arms $\mathcal{A}_l$ after the online samples are exhausted.

## 4.1 Properties and correctness of OOPE

In this subsection we will prove that OOPE does not request excess online or offline samples while still maintaining the requisite concentration results that help in proving tight high probability regret bounds. The logical condition in lines 12-16 in Algorithm 1 ensures OOPE does not request excess online samples, while the logical condition of lines 18-22 ensure that no arm is requested excess offline samples. We next establish the following upper-bound on the number of phases in which elimination occurs:

**Lemma 4.2.** *Denote the total number of phases upto and including the penultimate phase as $l_M$ (the last phase is one in which we exhaust online samples). We define the function $H^{-1} : \mathbb{R}_{\geq 0} \longrightarrow \mathbb{N} \cup \{0\}$ as:*

$$H^{-1}(x) = \max \left\{ n \mid \sum_{l=1}^{n} 4^l \log(4l^2 |\mathcal{A}| T) \leq x, n \in \mathbb{N} \cup \{0\} \right\}.$$

*Then we have that :*

$$l_M \leq H^{-1} \left( \frac{T}{3d_{eff}} \right).$$

The proof of Lemma 4.2 is presented in Appendix A.2. Using this bound we can show the following property of `OOPE`:

**Proposition 4.3.** *For every arm $a \in supp(\pi_{off})$, the total number of offline samples requested by `OOPE` of arm $a$ till phase $l_M$ does not exceed $\pi_{off}(a) T_{off}$.*

The proof of Proposition 4.3 is presented in Appendix A.3. Proposition 4.3 guarantees that we have enough offline samples for the requisite concentration results [8] to hold in all phases in which elimination takes place. This property is useful in enabling us to derive tight regret bounds for `OOPE`.

## 4.2 Regret bound for `OOPE`

In pure online settings, the Kiefer-Wolfowitz Theorem guarantees that $g(\tilde{\pi}_l^\star) = d$. In `OO` setting, this equality no longer holds. A key technical contribution of our work is to derive a generalized bound [9] in the next lemma showing that `OO` setting, $g(\tilde{\pi}_l^\star)$ is bounded by $d_{\text{eff}}$, which captures the spectral decay of the offline data. The proof is presented in Appendix A.4.

**Lemma 4.4** (Confidence width of D-optimal designs with offline data)**.** *The optimizer of (2) computed in phase $l$ satisfies the following relation:*

$$d = (1 - \alpha_l) g(\tilde{\pi}_l^\star) + \alpha_l \sum_{a \in \mathcal{A}} \pi_{off}(a) \|a\|_{V_{\tilde{\pi}_l^\star}^{-1}}^2, \tag{7}$$

*with $\tilde{\pi}_l^\star = (1 - \alpha_l) \pi_{l,on}^* + \alpha_l \pi_{off}$ and for all $\pi \in \Delta(\mathcal{A}_l)$ we have:*

$$d \leq (1 - \alpha_l) g_{\mathcal{A}_l}((1 - \alpha_l)\pi + \alpha_l \pi_{off}) + \alpha_l \sum_{a \in \mathcal{A}} \pi_{off}(a) \|a\|_{V_{\tilde{\pi}_l^\star}^{-1}}^2. \tag{8}$$

*Using these relations, we have the bound:*

$$(1 - \alpha_l) g(\tilde{\pi}_l^\star) \leq d_{eff}.$$

The lemma relates the maximum confidence width $g(\tilde{\pi}_l^\star)$ with the effective dimension $d_{\text{eff}}$. The maximum confidence width $g(\pi)$ for a design $\pi$, determines the concentration in the OLS estimator. In the purely online case with $\alpha_l = 0$, we have by Kiefer Wolfowitz Theorem, $g(\tilde{\pi}_l^\star) = d$. We note that with the offline data ($\alpha_l > 0$) this relationship no longer holds true. The upper bound in Lemma 4.4 replaces the Kiefer-Wolfowitz result in the presence of offline data and is key to deriving the regret bounds below.

We now present our main theorem which provides a bound on the regret of `OOPE`.

**Theorem 4.5** (Regret Bound)**.** *The `OOPE` algorithm satisfies the following regret bound with probability $1 - \frac{1}{T}$,*

$$\mathcal{R}(\text{OOPE}) \leq 16 \sqrt{6 d_{eff} T \log(4 l_{max}^2 |\mathcal{A}| T)} + 4d(d + 1) \tag{9}$$

*where $l_{max} = \left\lfloor \log_2 \sqrt{4 + \frac{T}{d_{eff} \log(4|\mathcal{A}|T)}} \right\rfloor$ and $d_{eff}$ is the effective dimension defined in equation (1).*

---

[8]see Step 1 of the proof of Theorem 4.5.

[9]This result subsumes the Kiefer-Wolfowitz theorem in online settings.

The proof of Theorem 4.5 is presented in the Appendix A.5. The regret bound is directly dependent on $d_{\text{eff}}$. In the pure online setting as there is no offline data we have $d_{\text{eff}} = d$ and we recover the online regret bound. The proof of Theorem 4.5 crucially relies on the confidence width $g(\tilde{\pi}_l^{\star})$ of the unknown parameter $\theta^*$, at phase $l$. Lemma 4.4 helps us derive a tight upper bound for this quantity.

*Remark* 4.6 (Extension to infinite sets). Theorem 4.5 can be extended to infinite action spaces that are a compact subset of $\mathbb{R}^d$. To see this, consider an $\epsilon$-net of $\mathcal{A}$ whose size is $O(1/\epsilon^d)$. By running OOPE on the $\epsilon$-net, we incur an additional regret of at most $T\epsilon$. By choosing $\epsilon = \tilde{O}(\sqrt{\frac{d_{\text{eff}}d}{T}} + \frac{d^2}{T})$, we obtain $\tilde{O}(\sqrt{d_{\text{eff}}dT} + d^2)$ regret. The additional factor of $d$ in $\sqrt{d_{\text{eff}}dT}$ is due to exponential in $d$ many arms we consider in the $\epsilon$-net.

*Remark* 4.7 (Making OOPE Horizon-Free). We can use the idea of the doubling trick with geometrically increasing horizons and full restarts (see, for example Besson & Kaufmann (2018)). Besson & Kaufmann (2018) shows that for algorithms with $O(\sqrt{T})$ minimax regret, the doubling trick makes the algorithms horizon-free at the cost of some absolute constant multipliers to the regret bound. For OOPE , we set $\alpha_l^i = \frac{T_{\text{off}}}{T_{\text{off}} + T^i}$ and estimate $d_{\text{eff}}^i$ for the $i^{th}$ episode. We note that for each restart, we reuse the entire offline data $\mathcal{D}_{off}$. This reuse of offline data preserves the eigenspectrum of the original data and is why we choose the "full" restart variant of Besson & Kaufmann (2018) instead of partial or no-restart variants.

**Multi-Armed Bandit (MAB) case:** This is an important special case of Linear bandits. The set $\mathcal{A}$ in MAB consists of the standard orthonormal basis of $\mathbb{R}^d$. For this special case, we give an analytical solution to the optimization (2):

**Proposition 4.8.** *Denote the set of unique values in $\pi_{off}$ vector when restricted to only the co-ordinates of the live arms $\mathcal{A}_l$ as:*

$$0 < \pi_{off_{(1)}} < \pi_{off_{(2)}} < \ldots \pi_{off_{(r)}} \le 1$$

*with each value $\pi_{off_{(i)}}$ being taken $m_i$ times for all $i \in [r]$ (note $r \le |\mathcal{A}_l|$). Define the sequence:*

$$\beta_i = \begin{cases} 1, & \text{for } i = 1 \\ \frac{1}{1 + \sum_{j=1}^{i-1} m_j(\pi_{off_{(i)}} - \pi_{off_{(j)}})}, & \text{for } 1 < i \le r \\ 0 & \text{for } i = r + 1. \end{cases}$$

*Then, for some $i \in [r]$ we have that $\beta_{i+1} \le \alpha_l < \beta_i$ and that:*

$$g(\tilde{\pi}_l^{\star}) = \frac{\sum_{j=1}^i m_j}{\alpha_l(\sum_{j=1}^i m_j \pi_{off_{(j)}}) + 1 - \alpha_l}. \tag{10}$$

*Further, the support of $\pi_{l,on}^*$ is $S_i = \{a \in \mathcal{A}_l \mid \pi_{off}(a) \le \pi_{off_{(i)}}\}$ and the optimizer is:*

$$\pi_{l,on}^*(a) = \frac{1}{|S_i|} + \frac{\alpha_l}{1 - \alpha_l}\left[\frac{\pi_{off}(S_i)}{|S_i|} - \pi_{off}(a)\right].$$

The proof of the proposition is in Appendix A.6. This allows us to avoid the computational overhead of solving equation (2) numerically. As far as we know, this is the first analytical solution when offline data is incorporated into D-optimal design. We recover the purely online MAB case when we set $\alpha_l = 0$, $\mathcal{A}_l = \mathcal{A}$, with $g(\tilde{\pi}_l^{\star}) = d$ and $\pi_{l,on}^*(a) = \frac{1}{d}$.

*Remark* 4.9. It can be shown that OOPE specialized to the MAB setting is minimax optimal for the well-explored and poorly explored regimes. The minimax lower bounds for the OO setting in MAB has been obtained in Cheung & Lyu (2024).

### 4.3 Lower bound for minimax regret.

We informally define our problem class $\mathcal{P}_{v,T_{\text{off}},T}^d$ to be characterised by the parameters $d, T, T_{\text{off}}$ and the d-dimensional vector $v$, where :

$$v_i = \frac{\lambda_i(V_{\pi_{\text{off}}})}{\max_{a \in \mathcal{A}} ||a||^2}.$$

is the vector of normalized eigenvalues of $V_{\pi_{\text{off}}}$. Intuitively the parameters $T_{\text{off}}$ and $v$ captures the amount and quality of the offline data respectively. For a more formal definition please see section A.7. We further assume that every bandit instance in $\mathcal{P}^d_{v,T_{\text{off}},T}$ is such that $|\mathcal{A}| \leq O(d^d)$. For this class, one defines the minimax regret $\mathcal{R}_{\text{minmax}}(\mathcal{P}^d_{v,T_{\text{off}},T})$ informally as the best possible regret for the worst problem instance from $\mathcal{P}^d_{v,T_{\text{off}},T}$. [10]

**Proposition 4.10.** *For any problem class $\mathcal{P}^d_{v,T_{off},T}$ with $|\mathcal{A}| \leq O(d^d)$ we have that:*

$$\mathcal{R}_{minmax}(\mathcal{P}^d_{v,T_{off},T}) \geq \frac{\sqrt{T}exp(-2)}{8} \sup_{\substack{w \in \Delta_d \\ \forall i, w_i \geq v_i}} \sum_{i=1}^{d} \frac{1}{\sqrt{1 + \frac{T_{off}}{T}\frac{v_i}{w_i}}}$$

*where $\Delta_d$ is d-dimensional simplex.*

The proof is provided in the Appendix A.7. The crux of the proof is finding a hard instance by perturbing the online hard instance of hypercubes (see Chapter 24 Lattimore & Szepesvári (2020)) based on the offline spectrum $v_i$'s. A simple approach of rotating the hypercubes $\mathcal{A}, \Theta$ by the eigenvectors of $V_{\pi_{\text{off}}}$ will become inconsistent due to the circular dependence of $V_{\pi_{\text{off}}}$ on $\mathcal{A}$. We carefully construct a hard instance that is consistent while perturbing the hypercube based on $v_i$'s.

The lower bound is a concave optimization program. We are able to characterize its dual in Lemma A.9. Utilizing this dual, we can show (see Lemma A.10) that for $k \in [d]$, such that $v_i = 0$ for $i < k$ and $0 < v_i$ for $i \geq k$:

$$\frac{(1 - \sum_i v_i)T_{\text{off}}}{2v_d(1 + \frac{T_{\text{off}}}{T})^{3/2}T} + (k-1) + \frac{(d-k+1)}{\sqrt{1 + \frac{T_{\text{off}}}{T}}} \geq \sup_{\substack{w \in \Delta_d \\ \forall i, w_i \geq v_i}} \sum_{i=1}^{d} \frac{1}{\sqrt{1 + \frac{T_{\text{off}}}{T}\frac{v_i}{w_i}}} \geq (k-1) + \frac{(d-k+1)}{\sqrt{1 + \frac{T_{\text{off}}(\sum_i v_i)}{T}}}. \quad (11)$$

Now consider the case when the offline data is well explored, i.e., $g(\pi_{\text{off}}) = O(d)$[11] or equally $v_i = \Omega(1/d)$ for all $i$, with $T_{\text{off}} \gg T$ then the effective dimension is $d_{\text{eff}} \leq O(dT/T_{\text{off}})$. Substituting $|A| = d^d$ in Theorem 4.5, gives us an upper bound $\mathcal{R}_{\text{minmax}}(\mathcal{P}^d_{v,T_{\text{off}},T}) \leq O(dT\log(Td)/\sqrt{T + T_{\text{off}}} + d^2)$. Using the bounds in equation (11) (here $k = 1$) we obtain that $\mathcal{R}_{\text{minmax}}(\mathcal{P}^d_{v,T_{\text{off}},T}) \geq \Omega(dT/\sqrt{T + T_{\text{off}}})$. OOPE is thus, minimax optimal up to logarithmic factors and an additive constant when large amount of offline data is well explored.

In the case where there are lots of poorly explored directions in offline data, that is $k = \Omega(d)$ in above, we get that $\mathcal{R}_{\text{minmax}}(\mathcal{P}^d_{v,T_{\text{off}},T}) \geq \Omega(d\sqrt{T})$. Applying again, Theorem 4.5 the OOPE regret bound we get that $d_{\text{eff}} = \theta(d)$ and hence $\mathcal{R}_{\text{minmax}}(\mathcal{P}^d_{v,T_{\text{off}},T}) \leq O(d\sqrt{T})$ and OOPE is again minimax optimal in this regime. We summarise the above discussion:

**Corollary 4.11.** *For problem classes $\mathcal{P}^d_{v,T_{off},T}$ with $T = \Omega(max(d\sqrt{T_{off}}, d^3))$, where either the offline data is well explored, that is $g(\pi_{off}) = \theta(d)$ or when the offline data is poorly explored, that is $k = \Omega(d)$ in equation (11), then OOPE is minimax optimal (modulo logarithmic factors and an additive constant) over these problem classes.*

A detailed calculation for the corollary is carried out in section A.8. In the case where there are only a few poorly explored directions, that is $k = o(d)$ in equation (11), there is a gap in the upper and lower bounds: $\tilde{O}(\sqrt{dkT}) \geq \mathcal{R}_{\text{minmax}}(\mathcal{P}^d_{v,T_{\text{off}},T}) \geq \theta(k\sqrt{T})$. We believe this slack comes from weakening of the upper bound on the confidence width $g(\tilde{\pi}^*)$ in analysis of Theorem 4.5 in this regime. Table 1 summarizes the above discussion.

*Remark* 4.12 (Gap between OOPE's upper bound and the lower bound). We believe the gap is purely analytical and essentially boils down to how we upper-bound the term $\lambda_d(V_{\pi^*_{l,on}})$ by $\max_a ||a||^2$ in proof of Lemma 4.4 in Appendix A.4. This bound does not reflect the dependence on the offline eigenspectrum and is just a uniform bound. This bound can be tight when the offline data has many underexplored directions, but it is quite weak in more well-explored settings. Improving this bound, to incorporate the offline spectrum in a more nuanced way, is challenging, and we leave it as further work.

---

[10]see eqn. equation (32) in section A.7.
[11]This happens when all directions are well explored in offline data, for *e.g.,* uniform offline arm pulls in orthogonal action sets.

| **Offline Regime** | $k$ | $g(\pi_{\mathbf{off}})$ | $\mathcal{R}_{\mathtt{OOPE}}$ | **Lower Bound** |
|---|---|---|---|---|
| Well-explored | 1 | $\theta(d)$ | $\tilde{O}(\frac{dT}{\sqrt{T+T_{\mathrm{off}}}})$ | $\Omega(\frac{dT}{\sqrt{T+T_{\mathrm{off}}}})$ |
| Moderately-explored | $o(d)$ | $\infty$ | $\tilde{O}(\sqrt{dkT})$ | $\Omega(k\sqrt{T})$ |
| Under-explored | $\Omega(d)$ | $\infty$ | $\tilde{O}(d\sqrt{T})$ | $\Omega(d\sqrt{T})$ |

Table 1: Our upper and lower bounds comparison in various offline regimes.

**Comparison with warm-started LinUCB:** One approach to incorporate offline data into LinUCB is to create the initial confidence ellipsoid utilizing all the offline data, and pull the arms which maximize the UCB index in the subsequent online rounds. In Proposition A.11, we present the existing regret guarantees for warm started LinUCB. In Appendix A.10, we present a simple problem setting where this regret bound is $d^{1/2}$ worse than the regret bound of $\mathtt{OOPE}$. The LinUCB bound has a dependence on $||\theta||_{V_0}$, where $V_0$ is the initial Gram matrix. When we "warm-start" LinUCB with $V_0 = T_{\mathrm{off}} V_{\pi_{\mathrm{off}}} + I$, $||\theta||_{V_0}$ on the hard instance of $\mathcal{A} = \{\pm 1\}^d, \Theta = \{\pm\sqrt{\frac{d}{(T_{\mathrm{off}}+T)}}\}^d$ can become $\Omega(d)$. This is an instance where LinUCB, either by warm-starting or by cold-starting ($V_0 = I$) attains the same regret bound $O(d^{3/2}T/\sqrt{T_{\mathrm{off}}})$ while in contrast $\mathtt{OOPE}$ gets $O(\frac{dT}{\sqrt{T_{\mathrm{off}}}})$ rates.

*Remark* 4.13. In practice, $\mathtt{OOPE}$ has certain additional benefits over LinUCB. First is computational in nature. Each iteration of LinUCB requires computing the UCB index [12] for each sample. Whereas in $\mathtt{OOPE}$, the D-optimal design problem is only solved $\log(T + T_{\mathrm{off}})$ times at the beginning of each phase. This makes $\mathtt{OOPE}$ computationally more efficient than UCB and hence appealing in practice.

## 5 Improving the dimension dependence in regret

The bound in Theorem 4.5 has an additional $O(d^2)$ term. This is due to the support of $\pi^*_{l,\mathrm{on}}$, which is at most $d(d+1)/2$. In some scenarios, this term in the regret can be important, for instance, if $d_{\mathrm{eff}} = \Omega(1)$, $|\mathcal{A}| = \Omega(d^2)$ and $T = O(d^2)$. Thus, this term can be a source of regret even if the typical dominant term $\tilde{O}(\sqrt{d_{\mathrm{eff}}T})$ is small. To address this, we compute an $\epsilon$-approximate solution to (2) using Frank-Wolfe that has $O(d/\epsilon \log\log(d))$ support points while ensuring the regret has only increased by at most $\sqrt{d\epsilon T}$ when using this approximate exploration schedule. We are trading the regret benefits of the optimal exploration for the smaller support size of the approximate solution.

To approximately solve the optimization in (2) we will use a version of Frank-Wolfe (FW) algorithm where at each update step at most one new arm is added. We will start with a carefully chosen initialization that has $O(d)$ support, and show that FW converges to an $\epsilon$-approximate solution in $O(d\log\log(d) + d/\epsilon)$ steps ensuring the second term in the regret is $O(d\log\log(d) + d/\epsilon)$ rather than $O(d^2)$ as in the previous section. This yields the following improved bound in the regime when $d = \Omega(1)$ which we state informally.

**Theorem** (Informal Improved support regret Bound). *The $\mathtt{OOPE}$ algorithm, where each phase l uses Frank-Wolfe iterations upto an accuracy $\epsilon = d_{eff}/d$, obtains a regret of $\tilde{O}(\sqrt{d_{eff}T\log(|\mathcal{A}|T)} + \frac{d^2}{d_{eff}})$ with prob. $1 - \frac{1}{T}$.*

When $d_{\mathrm{eff}} = \Omega(1)$, the above regret bound is a strict improvement over the regret of $\mathtt{OOPE}$.

**Dual problem:** An important tool in this analysis is the dual problem of equation (2). The dual of the maximization (2) is a minimum volume ellipsoid problem but the convex constraints are not the usual enclosing of arms condition that arise in the purely online setting (see chapter 2 of Todd (2016) for more details of the pure online case.). We recover the usual dual of Minimum Volume Enclosing Ellipsoid (MVEE) problem when we set $\alpha_l = 0$ in the following lemma (proof in Appendix B.1).

**Lemma 5.1.** *(Strong Duality) Consider the minimization problem*

$$\mathcal{P}(\mathcal{A}_l, \alpha_l) := \min_{H \succ 0} \quad -\log(\det(H))$$

---

[12]The rarely switching version (Theorem 4 of Abbasi-Yadkori et al. (2011)) requires $O(d\log(T + T_{\mathrm{off}}))$ updates that has a $d$ dependence that $\mathtt{OOPE}$ does not have.

*such that for all $a \in \mathcal{A}_l$,*

$$(1 - \alpha_l)a^t H a + \alpha_l Tr(V_{\pi_{off}} H) \leq d, \tag{12}$$

*This is dual to the optimization problem in (2) given by :*

$$\mathcal{D}(\mathcal{A}_l, \alpha_l) := \max_{\pi \in \Delta(\mathcal{A}_l)} \log\left( \det\left( (1 - \alpha_l)V_\pi + \alpha_l V_{\pi_{off}} \right) \right),$$

*that is, $\mathcal{P}(\mathcal{A}_l, \alpha_l) = \mathcal{D}(\mathcal{A}_l, \alpha_l)$.*

**Initialization and its Information Gain bound:** The initialization of Frank-Wolfe is the construction given in Kumar & Yildirim (2005). The pseudocode for the initialization is in Appendix B.2. At a high level, the initialization choose $d$ arms, each of which optimizes a well chosen linear function. Once the $d$ support points (arms) are selected, we put a uniform measure on them and use it as our initialization $\pi_l^{(0)}$ for the Frank-Wolfe procedure. For any exploration phase $l$, define the *information gain* function:

$$d(\pi, \mathcal{A}_l, \alpha) := \log(\det(\alpha_l V_{\pi_{off}} + (1 - \alpha_l)V_\pi), \tag{13}$$

where $\pi \in \Delta(\mathcal{A}_l)$. We have the following bound on $d(\pi_l^{(0)}, \mathcal{A}_l, \alpha_l)$:

**Proposition 5.2.** *For exploration phase $l$, let as before $\pi_{l,on}^*$ denote the optimal solution to (2), then with the above initialization $\pi_l^{(0)}$ we have: $d(\pi_{l,on}^*, \mathcal{A}_l, \alpha_l) - d(\pi_l^{(0)}, \mathcal{A}_l, \alpha_l) \leq d \log \frac{d^5}{(1-\alpha_l)}$.*

The proof of the proposition is in Appendix B.3. The proposition shows that the initialization is not too far from the optimal value with only $O(d)$ support points. The dual transforms the problem of maximizing the information gain into a geometric problem of minimizing the volume of an ellipsoid subject to certain constraints equation (12). The proof relates this geometric problem to the usual MVEE by means of a new feasibility relation amongst these two class of problems and scale invariance of the volume of MVEE (see Appendix B.7 and B.8). The use of feasibility relation and scale invariance is a novel addition to the typical online FW analysis on these types of problems (see Todd (2016)). Finally we bound this transformed MVEE using the following property of the initialization - $vol(conv(B)) \geq \frac{1}{d!}vol(conv(\mathcal{A}_l))$ (see Proposition B.3) where $B$ is the support of initialization $\pi_l^{(0)}$ obtained from $\mathcal{A}_l$.

### 5.1 Frank-Wolfe (FW) iterations after initialization

In this section we show that performing $t = \tilde{O}(d)$ iterations starting from $\pi_n^{(0)}$ is enough to guarantee a good solution $\pi_n^t$ with only $\tilde{O}(d)$ support. We first specify the FW updates. Then, we describe a potential function that FW update implicitly keeps tracks that directly translates to tighter regret bounds after $t = \tilde{O}(d)$ iterations.

**Definition 5.3.** *If $\pi$ is probability distribution on $\mathcal{V} \subseteq \mathcal{A}$ and if $(1 - \alpha)\sum_a \pi(a)aa^t + \alpha V_{\pi_{off}}$ is non singular then define $H(\pi) := \left( (1 - \alpha)\sum_a \pi(a)aa^t + \alpha V_{\pi_{off}} \right)^{-1}$.*

**Frank-Wolfe Algorithm:** We start from the initialization $\pi_l^{(0)}$ and apply Frank-Wolfe (FW) update specified in Algorithm 2.

The FW update adds atmost only one new arm in the support of the solution in each iteration. The arm added has the largest slack as defined in line 3 of Algorithm 2.

**Slack in equation (8) as potential:** In the previous section, we saw that the equality in Lemma 4.4 is crucial for establishing the regret bound in terms of $d_{eff}$. From the same Lemma, a sub-optimal online design $\pi \in \Delta(\mathcal{A}_l)$ would satisfy the inequality 8. Let $\tilde{\pi} = (1 - \alpha)\pi + \alpha\pi_{off}$. In our FW updates, we track the *potential*:

$$\delta(\pi_l^{(t)}) = \frac{(1 - \alpha)g(\tilde{\pi}_l^{(t)}) + \alpha \sum_{a \in \mathcal{A}} \pi_{off}(a)\|a\|^2_{V^{-1}_{\tilde{\pi}_l^{(t)}}}}{d} - 1,$$

---

**Algorithm 2** FRANK-WOLFE FOR OO SETTING

---

1: **Input:** $\epsilon, \mathcal{A}_l, \pi_l^{(0)}, V_{\pi_{\text{off}}}, \alpha$.
2: $t \leftarrow 0$.
3: $w \leftarrow \left( Tr\left( H(\pi_l^{(t)}) \left( (1-\alpha)aa^t + \alpha V_{\pi_{\text{off}}} \right) \right) \right)_{a \in \mathcal{A}_l}$ ; $\delta(\pi_l^{(t)}) = \frac{\max_a w_a}{d} - 1$, $a_+ \leftarrow \underset{a}{argmax} \; w_a$.
4: **while** $\epsilon < \delta(\pi_l^{(t)})$ **do**
5: $\quad \beta \leftarrow \frac{(w_{a_+} - d)}{(d-1)w_{a_+}}$, $\pi^{(+)} \leftarrow (1+\beta)^{-1}(\pi_l^{(t)} + \beta \mathbb{1}_{\{a_+\}})$.
6: $\quad t \leftarrow t + 1; \pi_l^{(t)} \leftarrow \pi^{(+)}$.
7: $\quad w \leftarrow \left( Tr\left( H(\pi_l^{(t)}) \left( (1-\alpha)aa^t + \alpha V_{\pi_{\text{off}}} \right) \right) \right)_{a \in \mathcal{A}_l}$.
8: $\quad \delta(\pi_l^{(t)}) = \frac{\max_a w_a}{d} - 1$, $a_+ \leftarrow \underset{a}{argmax} \; w_a$.
9: **end while**
10: **return:** $\pi_l^{(t)}$.

---

which is precisely the slack in equation (8) for $\pi_l^{(t)}$. It can also be re-written as $\delta(\pi_l^{(t)}) = \frac{w_{a_+}(\pi_n)}{d} - 1$ (Lines 3 or 8 of Algorithm 2). We next show that, if $\delta(\pi_l^{(t)})$ is large, then the FW update has a larger per iteration improvement in its information gain function (its objective) $d(\pi_l^{(t)}, \mathcal{A}_l, \alpha)$. Formally,

**Lemma 5.4.** *In the phase l, the per-iteration improvement in information gain of the FW update is given by:*

$$d(\pi_l^{(t+1)}, \mathcal{A}_l, \alpha) - d(\pi_l^{(t)}, \mathcal{A}_l, \alpha) \geq m(\delta(\pi_l^{(t)})) := \log(\delta(\pi_l^{(t)})) - \frac{\delta(\pi_l^{(t)})}{1 + \delta(\pi_l^{(t)})} \tag{14}$$

The proof is presented in Appendix B.4. A novel step in proving this lemma is using a *reverse* Jensen inequality from Merhav (2022) to lower bound the FW update improvement in the presence of offline data unlike the typical analysis done in Todd (2016).

**Bounding number of iterations to achieve slack** $\delta(\pi_l^{(t)}) \sim O(1)$ : Using the properties of the function $m(\delta)$ and Lemma 5.4, we bound the number of iterations of FW needed to get a small enough potential $\delta(\pi_l^{(t)})$ as follows:

**Proposition 5.5.** *The number of iterations required for the FW updates in Algorithm 2 to reach an iterate with slack $\delta(\pi_l^{(t)}) < \delta_0$ from $\pi_l^{(0)}$ is at most*

$$t = \frac{d}{1 - \frac{\delta_0}{(1+\delta_0)\log(1+\delta_0)}} \log\left( \frac{d}{\delta_0} \log\left( \frac{d^5}{1 - \alpha_l} \right) \right).$$

*The approximation $\pi_l^{(t)}$ has the property $|\text{supp}(\pi_l^{(t)})| \leq t + d$ and satisfies the following inequalities:*

$$d \leq (1 - \alpha_l)g(\tilde{\pi}_l^{(t)}) + \alpha_l \sum_{a \in \mathcal{A}} \pi_{off}(a)\|a\|_{V_{\tilde{\pi}_l^{(t)}}^{-1}}^2 \leq d(1 + \delta(\pi_l^{(t)})),$$

*where $\tilde{\pi}_l^{(t)} = (1 - \alpha_l)\pi_l^{(t)} + \alpha_l \pi_{off}$.*

The proof is given in Appendix B.5. The convergence rate does slow down as $\delta_0$ approaches zero as shown in the following lemma:

**Theorem** (Lemma 3.9, Todd (2016))**.** *The number of iterations to reduce the slack from $0 < \delta_0 < 1$ to $\delta_0/2$ is at most $\frac{14d}{\delta_0}$.*

Although proof in Todd (2016) is for the pure online case, it is straightforward to modify it to obtain the same result in the OO setting and is omitted. One can thus start with the Kumar & Yildirim (2005) initialization

(Algorithm 3) and run the FW Algorithm 2. We can split the upper bound analysis of number of iterations into two phases : one where $\delta(\pi_l^{(t)}) \geq 1$ and the second where $\delta(\pi_l^{(t)}) < 1$. In the first phase from Proposition 5.5 (we set $\delta_0 = 1$) we get the iterations is atmost $4d \log(d \log(d^5/(1 - \alpha_l))) + d$. In the second phase the number of iterations is bounded by $14d(1 + 2 + 2^2 + \cdots + 2^k)$ where $k = \log_2(1/\epsilon)$ to get $28d/\epsilon$ iterations. We have shown that:

**Corollary 5.6.** *The total number of iteration FW takes with Kumar & Yildirim (2005) initialization to attain a slack of $\epsilon < 1$ is $4d \log(d \log(d^5/(1 - \alpha_l))) + d + \frac{28d}{\epsilon}$.*

### 5.2 `OOPE-FW` and its Regret with improved dependence on $d$.

`OOPE-FW` **Algorithm**: We modify the `OOPE` Algorithm 1, where in each iteration instead of solving the optimization 2 exactly we solve it approximately in each phase $l$. We use the Kumar & Yildirim (2005) initialization (Algorithm 3) on the live arms $\mathcal{A}_l$ and then run the FW updates (Algorithm 2) from this initialization till a slack $\delta(\pi_l^{(t)}) \leq \frac{d_{\text{eff}}}{d}$ is reached. One has the following bound for the confidence-width $g_{\mathcal{A}_l}(\tilde{\pi}_l^{(t)})$ when we use the initialization 3 with FW (Algorithm 1) updates:

**Proposition 5.7.** *When `OOPE-FW` with FW iterations run till $\delta(\pi_l^{(t)}) \leq \frac{d_{\text{eff}}}{d}$ we have that:*

$$(1 - \alpha_l) g_{\mathcal{A}_l}(\tilde{\pi}_l^{(t)}) \leq 2d_{\text{eff}} \tag{15}$$

*where $\tilde{\pi}_l^{(t)} = (1 - \alpha_l)\pi_l^{(t)} + \alpha_l \pi_{\text{off}}$.*

The proof is presented in Appendix B.6. In each phase $l$ of `OOPE-FW` we use online samples:

$$n_{\text{on}, fw}^l(a) = \left\lceil \frac{6d_{\text{eff}} \pi_l^{(t)}(a) \log(4l^2 |\mathcal{A}| T)}{\epsilon_l^2} \right\rceil, \tag{16}$$

and offline samples:

$$n_{\text{off}, fw}^l(a) = \left\lceil \frac{2\alpha_l \pi_{\text{off}}(a) g(\tilde{\pi}_l^{(t)}) \log(4l^2 |\mathcal{A}| T)}{\epsilon_l^2} \right\rceil. \tag{17}$$

These particular number of samples ensure the requisite concentration inequalities hold. The `OOPE-FW` algorithm does not demand excess offline samples and online samples just like `OOPE`. We can derive a similar result to Proposition 4.3 for `OOPE-FW`, which implies offline sample for each arm $a$ does not exhaust until the last phase (when the online samples are exhausted).

We present the regret bound of `OOPE-FW`:

**Theorem 5.8** (`OOPE-FW` Regret Bound)**.** *The `OOPE-FW` algorithm has the following regret bound hold with probability $1 - \frac{1}{T}$,*

$$\mathcal{R}(\text{OOPE-FW}) \leq 32\sqrt{3d_{\text{eff}} T \log(4l_{max}^2 |\mathcal{A}| T)} + 8d + \frac{224d^2}{d_{\text{eff}}}$$

$$+ 32d \log\left(d \log\left(\left(\frac{d^5(T + T_{\text{off}})}{T}\right)\right)\right)$$

*where $d_{\text{eff}}$ is the effective dimension defined in equation (1) and $l_{max}$ is the same as defined in Theorem 4.5.*

The proof is given in Appendix B.7. The first term $\tilde{O}(\sqrt{d_{\text{eff}} T})$ has an additional constant $\sqrt{2}$ in `OOPE-FW` compared to `OOPE`. However, when $d_{\text{eff}} = \Omega(1)$, the above the bound on the second term is a strict improvement over the regret of `OOPE`. One can of course choose the version of phased elimination for the problem parameters (like $d, T, T_{\text{off}}, \pi_{\text{off}}$) and choose between `OOPE` and `OOPE-FW`. We record it as a corollary:

**Corollary 5.9.** *Using the appropriate phased elimination variant in `OO` setting, we can obtain with probability at least $1 - \frac{1}{T}$ a regret of $\tilde{O}\left(\sqrt{d_{\text{eff}} T} + \min\{d^2, \frac{d^2}{d_{\text{eff}}}\}\right)$.*

## 6 Experiments

In this section we present empirical evidence validating our theoretical insights.

**Simulation setting.** We select the unknown parameter $\theta$ uniformly from the unit sphere in $\mathbb{R}^d$. We choose the set of arms $\mathcal{A}$ also independently and uniformly from the unit sphere. The noise of each arm is set to be i.i.d standard normal distribution $\mathcal{N}(0,1)$. The offline data is generated by first choosing $T_{\text{off}}$ and the number of arms with offline data $n_{support}$. We ensure $2 \leq n_{support} \leq \min\{|\mathcal{A}|, T_{\text{off}}\}$. Then a partition of $T_{\text{off}}$ with size $n_{support} - 1$ is chosen uniformly, that is, any partition has probability $\frac{1}{\binom{T_{\text{off}}}{n_{support}-1}}$. The offline fraction $\pi_{\text{off}}$ is computed from this random partition and $T_{\text{off}}$. We set $20 \leq d \leq 30$, $d \leq |\mathcal{A}| \leq d^2$, $T \in [10^3, 10^5]$ and $T_{\text{off}} \in [10^4, 10^6]$ in all our experiments. Every horizon is run for 50 times, and the average regret along with a confidence interval is reported.

**Results.** We implement `OOPE` as prescribed in Section 4. With the parameters chosen as in Table 2,

| Parameter | Value |
|:---:|:---:|
| $d$ | 20 |
| $|\mathcal{A}|$ | 40 |
| $T$ | $10^4$ |
| $n_{support}$ | 40 |

Table 2: Parameter values for numerical experiment in Figure 1.

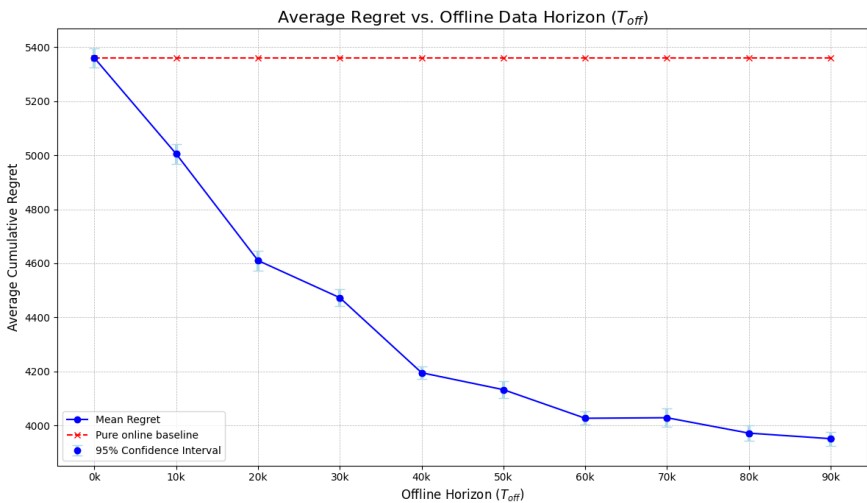

Figure 1: (**Improved performance with increasing offline data**) Plot showing lower regret with increasing offline samples for a fixed $\pi_{\text{off}}$. A purely online baseline is presented for comparison.

Figure 1 shows that `OOPE` performs better, i.e. incurs lower regret, as the number of offline samples increases. As a basic baseline, we compare with pure online Phased Elmination (dashed red). For this experiment we first sample a random partition uniformly with $T_{\text{off}} = 10^5$ and compute $\pi_{\text{off}}$. We regenerate offline data again for the shorter offline horizons holding the $\pi_{\text{off}}$ as fixed.

Next, we compare the performance of `OOPE` with warm-started UCB (LinUCB) and Thompson Sampling (LinTS). In the case of LinUCB the Grammian matrix is updated as $V_0 = T_{\text{off}}V_{\pi_{\text{off}}} + I$, while for LinTS the initial gaussian prior is set with empirical estimate of $\theta$ (OLS) and the variance-covariance matrix $\Sigma$ from the offline data. See Chapter 20 of Lattimore & Szepesvári (2020) and Appendix A.9 for details on the warm started LinUCB. We implement warm-started LinTS according to the details found in Agrawal & Goyal (2012); Oetomo et al. (2023). We choose the parameters as in Table 3. Figure 2 shows the better performance of `OOPE` compared to warm-started LinUCB and LinTS.

| Parameter | Value |
|:---:|:---:|
| $d$ | 20 |
| $|\mathcal{A}|$ | 40 |
| $T_{\text{off}}$ | $10^5$ |
| $n_{support}$ | 40 |

Table 3: Parameter values for numerical experiment in Figure 2.

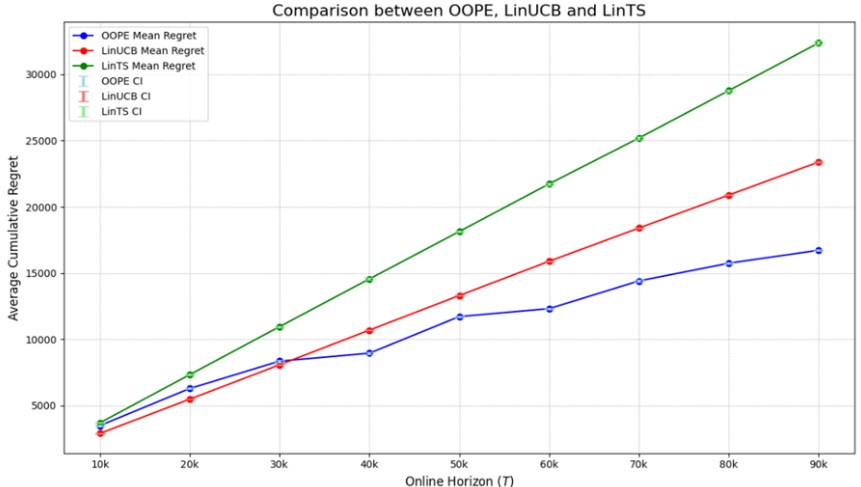

Figure 2: (**Comparison of `OOPE` versus warm-started LinUCB and LinTS.**) Plot showing better performance of `OOPE`.

As mentioned in Section 5, the support term can be a significant contributor to the regret performance of `OOPE`, particularly in larger online horizons, with large number of arms (i.e $|\mathcal{A}| = \Omega(d^2)$) and moderate effective dimension $d_{\text{eff}}$. The next experiment compares the performance of `OOPE-FW` with `OOPE` in such settings. We set the parameters as in Table 4. The Figure 3 shows the improved performance of `OOPE-FW`

| Parameter | Value |
|:---:|:---:|
| $d$ | 30 |
| $|\mathcal{A}|$ | 900 |
| $T_{\text{off}}$ | $10^6$ |
| $n_{support}$ | 100 |

Table 4: Parameter values for numerical experiment in Figure 3.

in such circumstances. The improved performance of `OOPE-FW` empirically is due to small support sizes of $\pi^*_{l,\text{on}}$ compared to `OOPE` in the initial phases, while still obtaining comparable $g(\tilde{\pi}^*_l)$.

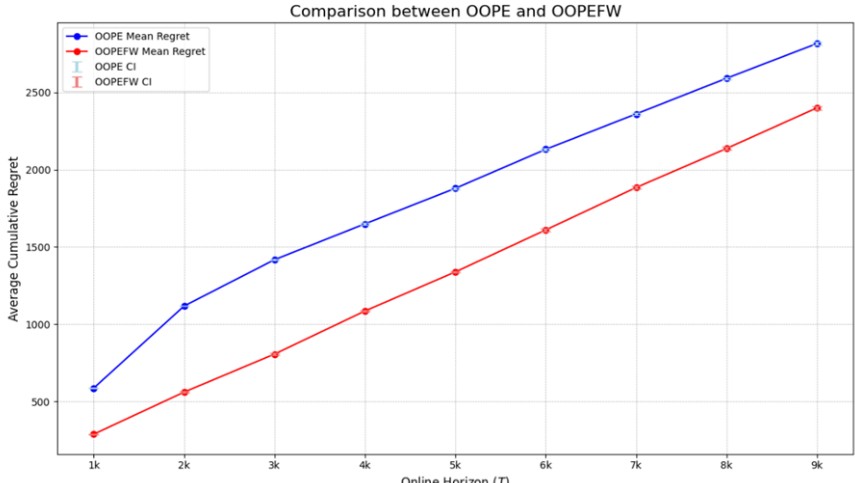

Figure 3: (**Comparison of `OOPE` versus `OOPE-FW`.**) Plot showing better performance of `OOPE-FW` in settings small $T, d_{\text{eff}}$ and large $|\mathcal{A}|$.

The regret and support terms are a complicated function of $d, T_{\text{off}}, \lambda$ and $T$. This makes it challenging to understand precisely when `OOPE-FW` outperforms `OOPE`. We perform a series of synthetic experiments where we fix $T, \lambda$ (the offline eigenspectrum) and vary $d_{\text{eff}}$ (or equivalently $T_{\text{off}}$). We do this for three different online horizons $T = \{2000, 20000, 100000\}$ (short, medium and long) and three different dimensions $d = \{17, 20, 22\}$. The table 5 presents the parameters chosen.

| Parameter | Value |
|---|---|
| $d$ | $\{17, 20, 22\}$ |
| $T$ | $\{2000, 20000, 100000\}$ |
| $|\mathcal{A}|$ | $d^2$ |
| $d_{\text{eff}}$ | $[1, d]$ |
| $n_{support}$ | $3d$ |

Table 5: Parameter values for numerical experiment in Figure 4.

In Figure 4 we plot the $\Delta Regret = \mathcal{R}(\texttt{OOPE}) - \mathcal{R}(\texttt{OOPE-FW})$ with the effective dimension $d_{\text{eff}}$. We do this for each dimension $d$ in $\{17, 20, 22\}$. For each $d$, we further plot the regret differences for three different horizons: 2000 (short), 20,000 (medium), and 100,000 (long) online horizons. Further, we quantify the spread of the offline eigenspectrum as $\kappa = \frac{\lambda_{max}}{\lambda_{min}}$. We try to ensure the spread $\kappa$ is in the same range $22 - 26$ for all instances, for the results to be comparable.

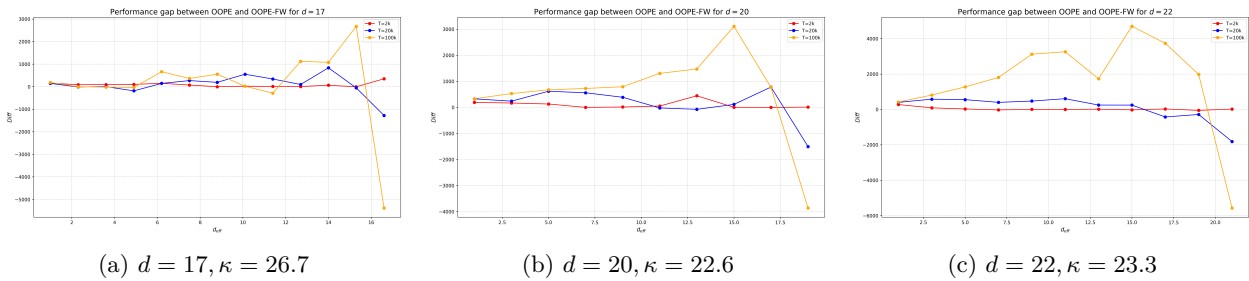

(a) $d = 17, \kappa = 26.7$        (b) $d = 20, \kappa = 22.6$        (c) $d = 22, \kappa = 23.3$

Figure 4: Performance gap ($\Delta$ Regret) versus $d_{\text{eff}}$ across different dimensions and horizons $T$. As $d$ increases, the advantage of `OOPE-FW` grows for moderate $d_{eff}$, but becomes worse than `OOPE` at high $d_{eff}$.

In Figure 4 we observe that for medium and longer online horizons ($T = 20000, 100000$) `OOPE` significantly outperforms `OOPE-FW`, when the effective dimension is large ($d_{\text{eff}} \approx d$). This is due to the additional constant multiple the $\tilde{O}(\sqrt{d_{\text{eff}}T})$ term has in `OOPE-FW`, which represents the 'cost' of using a Frank-Wolfe approximation. Conversely, for moderate $d_{\text{eff}}$, `OOPE-FW` outperforms `OOPE`. This aligns with when the support terms become dominant; `OOPE-FW` has the better $\frac{d^2}{d_{\text{eff}}}$ support size scaling. Additionally, we observe that this outperformance increases with $d$.

For small $d_{\text{eff}}$, the performance becomes comparable since the support terms for both `OOPE` and `OOPE-FW` scale similarly at $O(d^2)$. For shorter horizons ($T = 2000$), the learner has at most one phase of elimination. In this case, the efficiency gains of `OOPE-FW`'s sparse support are suppressed because the horizon expires before the saved exploration time can be converted into an exploitation dividend.

In summary, the differences between `OOPE` and `OOPE-FW` occur at medium to longer online horizons, and for medium to large $d_{\text{eff}}$. At large $d_{\text{eff}}$, `OOPE` does better, but for moderate `OOPE-FW` outperforms in a manner that increases with $d$.

## 7 Conclusion and Future Work

We study the problem of regret minimization in linear bandits with access to offline data. We propose a phased elimination algorithm `OOPE`, based on a generalized notion of D-optimal design, which achieves significantly lower regret. We identify an *effective* dimension ($d_{\text{eff}}$) based on the offline Grammian eigenspectrum that quantitatively captures this. Consequently we obtain an improved regret of $O(\sqrt{d_{\text{eff}}T\log(|\mathcal{A}|)} + d^2)$. This matches a novel minimax lower bound upto $\log(dT)$ factors and additive constants in *well & poorly* explored offline data regimes. The lower bound was derived using novel offline quality dependent perturbations of the hypercube. In settings with small $T$, $d_{\text{eff}}$ and large number of arms, the $O(d^2)$ support term might dominate the $\tilde{O}(\sqrt{d_{\text{eff}}T})$. To overcome this, we propose a Frank-Wolfe variant of `OOPE`, called `OOPE-FW`. Our theoretical insights are further validated with synthetic numerical experiments.

**Future Work.** The current work assumes a stochastic offline data generation process. It will be useful to relax this assumption and study the case when offline data comes from adaptive policies. In many practical situations the online data comes from a slightly shifted $\theta^*$ vis-a-vis the offline data, with a certain shift budget known a priori (see for example Cheung & Lyu (2024)). It will be important to study extensions of `OO` setting with such drift.

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

## A  Proof of results in Section 4

### A.1  Proof of Proposition 4.1.

*Proof.* We first fix $l_M = k$ for a fixed $k \in \mathbb{N}$. We optimize over $\alpha_l$, $1 \leq l \leq k$, first and deal with optimizing over $k$ later. We observe the optimization:

$$\inf_{\alpha_l \in [0,1], \forall l \in [k]} \sum_{l=1}^{k} 2^{\ell} \frac{(1 - \alpha_l)}{\alpha_l}$$
$$\text{s.t. } \min\left\{\frac{4^{k+1} - 4}{3} - T, T_{\text{off}}\right\} \geq \sum_{l=1}^{k} 2^{2l} \alpha_l \tag{18}$$

is a convex program. We study the Fenchel Dual of this convex program. We use Theorem Fenchel Duality here with the following choices (see section 7.10-7.12 in Luenberger (1997) for more details):

$$f(\alpha) = \sum_{\ell=1}^{k} 2^{\ell} \frac{(1 - \alpha_\ell)}{\alpha_\ell}, \quad C = \left\{\alpha \in \mathbb{R}^k \big| \alpha \in [0,1]^k\right\}$$

$$g(\alpha) = 0, D = \left\{\alpha \in \mathbb{R}^k \left| \min\left\{\frac{4^{k+1} - 4}{3} - T, T_{\text{off}}\right\} \geq \sum_{\ell=1}^{h} \alpha_\ell 2^{2\ell}\right.\right\}.$$

One can then find the respective conjugate domains and functionals as follows:

$$f^*(x^*) = \sum_{l=1}^{k} \left(x_l^* \mathbb{1}_{\{x_l^* + 2^l \geq 0\}} - \left(2^{\frac{l+2}{2}} \sqrt{|x_l^*|} - 2^l\right) \mathbb{1}_{\{x_l^* + 2^l < 0\}}\right), \quad C^* = \mathbb{R}^k$$

$$g^*(x^*) = -r \min\left\{\frac{4^{k+1} - 4}{3} - T, T_{\text{off}}\right\}, D^* = \left\{-r(2^{2l})_{l \in [k]} \big| r \geq 0\right\}.$$

We note that $C^* \cap D^*$ is just the negative ray along the vector $(2^{2l})_{l \in [k]}$. Using the above conjugate functionals and domains one then has that the dual problem is:

$$\max_{r \geq 0} \left(-r \min\left\{\frac{4^{k+1} - 4}{3} - T, T_{\text{off}}\right\} + \sum_{l=1}^{k} \left(r 2^{2l} \mathbb{1}_{\{1 \geq r 2^l\}} + (2^{3l/2+1} \sqrt{r} - 2^l) \mathbb{1}_{\{1 < r 2^l\}}\right)\right).$$

Now, in general it is hard to write down an analytical expression for this dual optimization. However if we restrict the optimization over $r$ to $[0, 2^{-k}]$ we see that the value of the dual is atleast:

$$2^{-k} \max\left\{T, \frac{4^{k+1} - 4}{3} - T_{\text{off}}\right\}.$$

Minimizing this lower bound over $k \in \mathbb{N}$, we observe that the optimal $k^* = \theta(\log_2 \sqrt{T + T_{\text{off}}})$ and the resulting value of the optimization is atleast $\Omega\left(\frac{T}{\sqrt{T + T_{\text{off}}}}\right)$.

In the primal minimization if we choose $k = \log_2(\sqrt{3(T + T_{\text{off}}) + 4})$ then we see that the choice

$\alpha_l = \frac{T_{\text{off}}}{T + T_{\text{off}}}, \forall l \in [k]$ gets us an upper bound on the primal minimization of $2^k \frac{T}{T_{\text{off}}} = O\left(\sqrt{T + T_{\text{off}}} \frac{T}{T_{\text{off}}}\right)$. In the regime of $T < T_{\text{off}}$ we then have that this is $O\left(\frac{T}{\sqrt{T + T_{\text{off}}}}\right)$.

Next, we show that amongst all schedules that hold $\alpha_l = \alpha$ is fixed across $l \leq l_M = k$ our choice of $\alpha_l$ and $l_M$ is optimal. In this class of static $\alpha$ the optimization becomes:

$$\inf_{\alpha \in [0,1], k \in \mathbb{N}} \frac{(1 - \alpha)}{\alpha}(2^{k+1} - 2)$$
$$\text{s.t. } \min\left\{\frac{4^{k+1} - 4}{3} - T, T_{\text{off}}\right\} \geq \alpha\frac{(4^{k+1} - 4)}{3} \tag{19}$$

which simplifies to :

$$\min_{k \in \mathbb{N}}(2^{k+1} - 2)\left(\frac{1}{\min\left\{1 - \frac{3T}{4^{k+1} - 4}, \frac{3T_{\text{off}}}{4^{k+1} - 4}\right\}} - 1\right).$$

We consider two cases:
**Case 1:** $4^{k+1} \geq 3(T + T_{\text{off}}) + 4$**.** In this case the optimization becomes:

$$\min_{k+1 \geq \lceil \log_2(\sqrt{3(T + T_{\text{off}}) + 4})\rceil} (2^{k+1} - 2)\left(\frac{4^{k+1} - 4}{3T_{\text{off}}} - 1\right).$$

As the objective is an increasing function of $k + 1$, we have that the optimal $k$ in this case is:

$$k^* = \lceil \log_2(\sqrt{3(T + T_{\text{off}}) + 4})\rceil - 1$$

and that:

$$\frac{T_{\text{off}}}{4(T + T_{\text{off}} + 1)} \leq \alpha^* = \frac{3T_{\text{off}}}{4^{k*+1} - 4} \leq \frac{T_{\text{off}}}{T + T_{\text{off}}}. \tag{20}$$

**Case 2:** $4^{k+1} \leq 3(T + T_{\text{off}}) + 4$**.** In this case we have that:

$$\min_{k+1 \leq \lfloor \log_2(\sqrt{3(T + T_{\text{off}}) + 4})\rfloor} \frac{1}{4 + (2^{k+1} - 2) - \frac{3T}{2^{k+1} - 2}}.$$

The objective in this case is a decreasing function of $k + 1$ and hence the optimal $k^*$ is given by:

$$k^* = \lfloor \log_2(\sqrt{3(T + T_{\text{off}}) + 4})\rfloor - 1$$

and that $\alpha^*$ is given by:

$$\alpha^* = 1 - \frac{3T}{4^{k*+1} - 4} \leq \frac{T_{\text{off}}}{T + T_{\text{off}}}.$$

This shows that $l_M = \theta(\log_2 \sqrt{T + T_{\text{off}}})$ and $\alpha = \theta\left(\frac{T_{\text{off}}}{T + T_{\text{off}}}\right)$ is optimal amongst schedules that hold $\alpha_l$ fixed across phases. $\qquad\qquad\square$

### A.2 Proof of Lemma 4.2.

From the definition of $l_M$ and the number of online samples required (equation (4)) in each phase we have that:

$$
\begin{aligned}
T &\geq \sum_{l=1}^{l_M} \sum_{a \in \mathcal{A}_l} n_{\text{on}}^l(a) \\
&\geq \sum_{l=1}^{l_M} \sum_{a \in \mathcal{A}_l} \frac{3 d_{\text{eff}} \pi_{l,\text{on}}^*(a)}{\epsilon_l^2} \log(4l^2 |\mathcal{A}| T) \\
&= \sum_{l=1}^{l_M} \frac{3 d_{\text{eff}}}{\epsilon_l^2} \log(4l^2 |\mathcal{A}| T) \\
&= 3 d_{\text{eff}} \sum_{l=1}^{l_M} 4^l \log(4l^2 |\mathcal{A}| T).
\end{aligned}
$$

From this inequality and definition of the $H^{-1}$ we have that:

$$
l_M \leq H^{-1} \left( \frac{T}{3 d_{\text{eff}}} \right).
$$

### A.3 Proof of Proposition 4.3.

It is useful to define the following function $H : \mathbb{N} \cup \{0\} \longrightarrow \mathbb{R}_{\geq 0}$ as:

$$
H(n) = \sum_{l=1}^{n} 4^l \log(4l^2 |\mathcal{A}| T).
$$

Now for a given arm $a \in supp(\pi_{\text{off}})$ the number of samples used upto phase $l_M$ is given by:

$$
\begin{aligned}
\sum_{l=1}^{l_M} n_{\text{off}}^l(a) &= \sum_{l=1}^{l_M} \left\lceil \frac{2 \alpha_l \pi_{\text{off}}(a) g(\tilde{\pi}_l^\star) \log(4l^2 |\mathcal{A}|/\delta)}{\epsilon_l^2} \right\rceil \\
&\leq \sum_{l=1}^{l_M} \frac{2 \alpha_l \pi_{\text{off}}(a) d_e \log(4l^2 |\mathcal{A}| T)}{(1 - \alpha_l) \epsilon_l^2} + l_M \\
&= \sum_{l=1}^{l_M} \frac{2 T_{\text{off}} \pi_{\text{off}}(a) d_{\text{eff}} \log(4l^2 |\mathcal{A}| T)}{T \epsilon_l^2} + l_M \\
&= \frac{2 T_{\text{off}} \pi_{\text{off}}(a) d_{\text{eff}}}{T} \sum_{l=1}^{l_M} 4^l \log(4l^2 |\mathcal{A}| T) + l_M \\
&= \frac{2 T_{\text{off}} \pi_{\text{off}}(a) d_{\text{eff}}}{T} H(l_M) + l_M \\
&\leq \frac{2 T_{\text{off}} \pi_{\text{off}}(a) d_{\text{eff}}}{T} \frac{T}{3 d_{\text{eff}}} + l_M \\
&= \frac{2}{3} \pi_{\text{off}}(a) T_{\text{off}} + l_M.
\end{aligned}
$$

where the last inequality uses Lemma 4.2. We next derive a bound on $l_M$ in a similar way to Lemma 4.2.

$$
\begin{aligned}
T &\geq \sum_{l=1}^{l_M} \sum_{a \in \mathcal{A}_l} n_{\text{on}}^l(a) \\
&\geq \sum_{l=1}^{l_M} \sum_{a \in \mathcal{A}_l} \frac{3 d_{\text{eff}} \pi_{l,\text{on}}^*(a)}{\epsilon_l^2} \log(4l^2 |\mathcal{A}| T) \\
&\geq 3 d_{\text{eff}} \log(4|\mathcal{A}|T) \sum_{l=1}^{l_M} 4^l \\
&= 4 d_{\text{eff}} \log(4|\mathcal{A}|T)(4^{l_M} - 1).
\end{aligned}
$$

Next we lower bound $d_{\text{eff}}$ using its definition 1:

$$
\sum_{k=1}^d \frac{1}{1 + \frac{T_{\text{off}}}{T} \frac{\lambda_k(V_{\pi_{\text{off}}})}{\max_a ||a||^2}} \geq \sum_{k=1}^d \frac{1}{1 + \frac{T_{\text{off}}}{T}} = \frac{dT}{T + T_{\text{off}}}
$$

where we have used the fact that $\lambda_k(V_{\pi_{\text{off}}}) \leq \max_a ||a||^2$. Similarly:

$$
\frac{T}{T_{\text{off}}} g_{\mathcal{A}}(\pi_{\text{off}}) = \frac{T}{T_{\text{off}}} \max_{a \in \mathcal{A}} \sum_{i=1}^d \frac{a_i^2}{\lambda_i(V_{\pi_{\text{off}}})} \geq \frac{T}{T_{\text{off}}},
$$

where the inequality again follows from $\lambda_k(V_{\pi_{\text{off}}}) \leq \max_a ||a||^2$. Combining the above two inequalities with definition 1 we have that:

$$
d_{\text{eff}} \geq \min\left\{\frac{dT}{T + T_{\text{off}}}, \frac{T}{T_{\text{off}}}\right\}.
$$

This implies that:

$$
\left\lfloor \log_2 \sqrt{\frac{\max\{\frac{T+T_{\text{off}}}{d}, T_{\text{off}}\}}{4\log(4|\mathcal{A}|T)} + 1} \right\rfloor \geq l_M
$$

But from Assumption 3.3 we know that the LHS is less than $\frac{\pi_{\text{off}}(a)T_{\text{off}}}{3}$. Thus one concludes that:

$$
l_M \leq \frac{\pi_{\text{off}}(a)T_{\text{off}}}{3}.
$$

Using this we have that :

$$
\sum_{l=1}^{l_M} n_{\text{off}}^l(a) \leq \frac{2}{3}\pi_{\text{off}}(a)T_{\text{off}} + l_M \leq \frac{2}{3}\pi_{\text{off}}(a)T_{\text{off}} + \frac{\pi_{\text{off}}(a)T_{\text{off}}}{3} = \pi_{\text{off}}(a)T_{\text{off}}.
$$

This concludes the proof of the proposition.

### A.4 Proof of Lemma 4.4.

For any $\pi = (1 - \alpha_l)\pi_{l,\text{on}} + \alpha_l \pi_{\text{off}}$, with $\pi_{l,\text{on}} \in \Delta(\mathcal{A}_l)$ we have:

$$
\begin{aligned}
\sum_{a \in \mathcal{A}} \pi(a) ||a||_{V_\pi^{-1}}^2 &= \sum_{a \in \mathcal{A}} \pi(a) a^t V_\pi^{-1} a \\
&= \sum_{a \in \mathcal{A}} \pi(a) Tr(aa^t V_\pi^{-1}) \\
&= Tr\left(\left(\sum_{a \in \mathcal{A}} \pi(a) aa^t\right) V_\pi^{-1}\right) \\
&= Tr\left(V_\pi V_\pi^{-1}\right) \\
&= d.
\end{aligned}
$$

But by definition of $g_{\mathcal{A}_l}(\pi) := \max_{a \in \mathcal{A}_l} ||a||^2_{V_\pi^{-1}}$ we have

$$d = \sum_{a \in \mathcal{A}} \pi(a)||a||^2_{V_\pi^{-1}} = (1 - \alpha_l) \sum_{a \in \mathcal{A}_l} \pi_{l,\text{on}}(a)||a||^2_{V_\pi^{-1}} + \alpha_l \sum_{a \in \mathcal{A}} \pi_{\text{off}}(a)||a||^2_{V_\pi^{-1}}$$

$$d \leq (1 - \alpha_l)g_{\mathcal{A}_l}(\pi) + \alpha_l \sum_{a \in \mathcal{A}} \pi_{\text{off}}(a)||a||^2_{V_\pi^{-1}}.$$

To show equation (7) we first observe that the optimization in (2) :

$$\max_{\pi_{l,\text{on}} \in \Delta(\mathcal{A}_l)} \log \left( \det \left( (1 - \alpha_l) \sum_a \pi_{l,\text{on}}(a)aa^t + \alpha_l \sum_a \pi_{\text{off}}(a)aa^t \right) \right)$$

is concave in $\pi_{l,\text{on}}$. Next, using the fact that the derivative of the objective wrt $\pi_{l,\text{on}}(a)$ from standard matrix calculus is $(1 - \alpha_l)||a||^2_{V_\pi^{-1}}$ and using first-order optimality conditions for a concave maximization we have:

$$0 \geq \sum_{a \in \mathcal{A}_l} (1 - \alpha_l)||a||^2_{V_{\tilde{\pi}_l^*}^{-1}} (\pi_{l,\text{on}}(a) - \pi^*_{l,\text{on}}(a)).$$

But this implies that since $\tilde{\pi}_l^\star = (1 - \alpha_l)\pi^*_{l,\text{on}} + \alpha_l \pi_{\text{off}}$ we have:

$$\alpha_l \sum_{a \in \mathcal{A}} \pi_{\text{off}}(a)||a||^2_{V_{\tilde{\pi}_l^*}^{-1}} + (1 - \alpha_l) \sum_{a \in \mathcal{A}_l} \pi^*_{l,\text{on}}(a)||a||^2_{V_{\tilde{\pi}_l^*}^{-1}} \geq \alpha_l \sum_{a \in \mathcal{A}} \pi_{\text{off}}(a)||a||^2_{V_{\tilde{\pi}_l^*}^{-1}} + (1 - \alpha_l) \sum_{a \in \mathcal{A}_l} \pi_{l,\text{on}}(a)||a||^2_{V_{\tilde{\pi}_l^*}^{-1}}$$

$$\sum_{a \in \mathcal{A}} \tilde{\pi}^*(a)||a||^2_{V_{\tilde{\pi}_l^*}^{-1}} \geq \alpha_l \sum_{a \in \mathcal{A}} \pi_{\text{off}}(a)||a||^2_{V_{\tilde{\pi}_l^*}^{-1}} + (1 - \alpha_l) \sum_{a \in \mathcal{A}_l} \pi_{l,\text{on}}(a)||a||^2_{V_{\tilde{\pi}_l^*}^{-1}}$$

$$d \geq \alpha_l \sum_{a \in \mathcal{A}} \pi_{\text{off}}(a)||a||^2_{V_{\tilde{\pi}_l^*}^{-1}} + (1 - \alpha_l) \sum_{a \in \mathcal{A}_l} \pi_{l,\text{on}}(a)||a||^2_{V_{\tilde{\pi}_l^*}^{-1}}.$$

But since $\pi_{l,\text{on}}$ is arbitrary element from $\Delta(\mathcal{A}_l)$ we have that

$$d \geq \alpha_l \sum_{a \in \mathcal{A}} \pi_{\text{off}}(a)||a||^2_{V_{\tilde{\pi}_l^*}^{-1}} + (1 - \alpha_l)g(\tilde{\pi}_l^\star). \tag{21}$$

Combining (21) with the earlier inequality gives us (7):

$$d = \alpha_l \sum_{a \in \mathcal{A}} \pi_{\text{off}}(a)||a||^2_{V_{\tilde{\pi}_l^*}^{-1}} + (1 - \alpha_l)g(\tilde{\pi}_l^\star).$$

Applying to this the Woodbury Matrix identity $(A + B)^{-1} = A^{-1} - A^{-1}(I + AB^\dagger)^{-1}$ to this, where $B^\dagger$ represents the pseudoinverse, we get:

$$(1 - \alpha_l)g(\tilde{\pi}_l^\star) = Tr\left( \left( I + \frac{\alpha}{1 - \alpha}V_{\pi_{\text{off}}}V^\dagger_{\pi^*_{l,\text{on}}} \right)^{-1} \right).$$

Now using the following linear algebra result for any two PSD matrices $A, B$:

$$\lambda_k(A)\lambda_1(B) \leq \lambda_k(AB) \leq \lambda_k(A)\lambda_d(B), \tag{22}$$

we then have that (we set $B = I$, $A = I + \frac{\alpha}{1-\alpha}V_{\pi_{\text{off}}}V^\dagger_{\pi^*_{l,\text{on}}}$)

$$1 + \frac{\alpha_l}{1 - \alpha_l}\frac{\lambda_k(V_{\pi_{\text{off}}})}{\lambda_d(V_{\pi^*_{l,\text{on}}})} \leq \lambda_k\left( I + \frac{\alpha_l}{1 - \alpha_l}V_{\pi_{\text{off}}}V^\dagger_{\pi^*_{l,\text{on}}} \right). \tag{23}$$

Using (23) in the representation above we get :

$$(1 - \alpha_l)g(\tilde{\pi}_l^\star) \leq \sum_{k=1}^d \frac{1}{1 + \frac{\alpha_l}{1-\alpha_l}\frac{\lambda_k(V_{\pi_{\text{off}}})}{\lambda_d(V_{\pi^*_{l,\text{on}}})}}.$$

Using the fact that $\lambda_d(V_{\pi^*_{l,\mathrm{on}}}) \leq Tr(V_{\pi^*_{l,\mathrm{on}}}) \leq \max_a ||a||^2$ and $\alpha_l = \frac{T_{\mathrm{off}}}{T+T_{\mathrm{off}}}$, we finally have:

$$(1-\alpha_l)g(\tilde{\pi}_l^\star) \leq \sum_{k=1}^d \frac{1}{1+\frac{T_{\mathrm{off}}}{T}\frac{\lambda_k(V_{\pi_{\mathrm{off}}})}{\max_a ||a||^2}}. \tag{24}$$

A simple but alternate useful bound for the confidence width is:

$$g(\tilde{\pi}_l^\star) \leq g(\pi_{\mathrm{off}})/\alpha_l \tag{25}$$

where $g(\pi_{\mathrm{off}}) \triangleq \max_{a \in \mathcal{A}} ||a||^2_{V_{\pi_{\mathrm{off}}}^{-1}}$ and follows from the fact that $V_{\tilde{\pi}^*} \succeq \alpha_l V_{\pi_{\mathrm{off}}}$. Note that this bound is useful in the regime where the offline is well explored, i.e, $g(\pi_{\mathrm{off}}) = \theta(d)$, but can be pretty loose when the offline data is not uniformly well explored in some directions (in particular $g(\pi_{\mathrm{off}}) = \infty$). Combining equation (25) with equation (23) and using the definition of $d_{\mathrm{eff}}$ (equation (1)) finally gives us:

$$(1-\alpha_l)g(\tilde{\pi}_l^\star) \leq d_{\mathrm{eff}}.$$

## A.5   Proof of Theorem 4.5.

We divide the proof into the following key steps:

**Step 1: Concentration result for "clean" execution of `OOPE`.**

Let us define the event $\xi_{a,l}(\mathcal{V})$ for any arm $a$ and subset $\mathcal{V} \subseteq \mathcal{A}$ as

$$\xi_{a,l}(\mathcal{V}) := \{|\langle a, \theta^* - \hat{\theta}_l \rangle| \leq \epsilon_l\}$$

where $\hat{\theta}_l$ is the OLS estimator constructed from the offline samples and online samples in phase $l$. We assume the set of "live" arms is $\mathcal{V}$ and the online samples are taken only from this collection of live arms. We then have that:

$$\mathbb{P}\left(\bigcup_{l=1}^\infty \bigcup_{a \in \mathcal{A}_l} \{\xi^c_{a,l}(\mathcal{A}_l)\}\right) \leq \sum_{l=1}^\infty \mathbb{P}\left(\bigcup_{a \in \mathcal{A}_l} \{\xi^c_{a,l}(\mathcal{A}_l)\}\right)$$

$$= \sum_{l=1}^\infty \sum_{\mathcal{V} \subseteq \mathcal{A}} \mathbb{P}\left(\bigcup_{a \in \mathcal{V}} \{\xi^c_{a,l}(\mathcal{V}), \mathcal{A}_l = \mathcal{V}\}\right).$$

We note that the estimate $\hat{\theta}_l$ is independent from the event $\{\mathcal{A}_l = \mathcal{V}\}$ as the offline samples are only ever used in one phase and never thereafter in estimating $\hat{\theta}_l$. This ensures no dependency is created between these two events. Thus

$$\mathbb{P}\left(\bigcup_{l=1}^\infty \bigcup_{a \in \mathcal{A}_l} \{\xi^c_{a,l}(\mathcal{A}_l)\}\right) \leq \sum_{l=1}^\infty \sum_{\mathcal{V} \subseteq \mathcal{A}} \mathbb{P}\left(\bigcup_{a \in \mathcal{V}} \{\xi^c_{a,l}(\mathcal{V})\}\right)\mathbb{P}\left(\{\mathcal{A}_l = \mathcal{V}\}\right)$$

$$\leq \sum_{l=1}^\infty \sum_{\mathcal{V} \subseteq \mathcal{A}} \sum_{a \in \mathcal{V}} \mathbb{P}\left(\{\xi^c_{a,l}(\mathcal{V})\}\right)\mathbb{P}\left(\{\mathcal{A}_l = \mathcal{V}\}\right) \tag{26}$$

Now we state a useful sub-gaussian concentration result:

**Lemma A.1.** *Given the offline samples $n^l_{on}(a)$ and online samples $n^l_{off}(a)$ are as defined in equation (4) and equation (5) respectively, we have that*

$$\mathbb{P}(|\langle a, \theta^* - \hat{\theta}_l \rangle| \geq \epsilon_l) \leq \frac{1}{2Tl^2|\mathcal{A}|}. \tag{27}$$

*Proof of Lemma A.1.* Define $n^l(a) = n^l_{\mathrm{on}}(a) + n^l_{\mathrm{off}}(a)$, where

$$n^l_{\mathrm{on}}(a) = \left\lceil \frac{3d_{\mathrm{eff}}\pi^*_{l,\mathrm{on}}(a)}{\epsilon_l^2} \log(4l^2|\mathcal{A}|T) \right\rceil$$

and

$$n_{\text{off}}^l(a) = \left\lceil \frac{2\alpha_l \pi_{\text{off}}(a) g(\tilde{\pi}_l^\star) \log(4l^2|\mathcal{A}|T)}{\epsilon_l^2} \right\rceil.$$

are the online and offline samples, respectively, of arm $a$ utilized in phase $l$ to construct $\hat{\theta}_l$. Further, define the positive definite matrix $V := \sum_a n^l(a) aa^t$. Then, we have that:

$$V \succeq \left( \alpha_l \frac{2g(\tilde{\pi}_l^\star) \log(4l^2|\mathcal{A}|T)}{\epsilon_l^2} \sum_a \pi_{\text{off}}(a) aa^t + (1-\alpha_l) \frac{3d_{\text{eff}} \pi_{l,\text{on}}^*(a) \log(4l^2|\mathcal{A}|T)}{\epsilon_l^2} \sum_a \pi_{l,\text{on}}^*(a) aa^t \right)$$

$$\succeq \frac{2g(\tilde{\pi}_l^\star) \log(4l^2|\mathcal{A}|T)}{\epsilon_l^2} \left( \alpha_l \sum_a \pi_{\text{off}}(a) aa^t + (1-\alpha_l) \sum_a \pi_{l,\text{on}}^*(a) aa^t \right)$$

$$= \frac{2g(\tilde{\pi}_l^\star) \log(4l^2|\mathcal{A}|T)}{\epsilon_l^2} V_{\tilde{\pi}^*}.$$

Here $\succeq$ refers to the standard partial ordering of positive definiteness on the space of $d \times d$ matrices. The second inequality above is obtained from $2(1-\alpha_l)g(\tilde{\pi}_l^\star) \leq 3d_{\text{eff}}$ (see Lemma 4.4). Consequently, for all $a \in \mathcal{A}$ we have:

$$\sqrt{2||a||_{V^{-1}}^2 \log(4l^2|\mathcal{A}|T)} \leq \epsilon_l \sqrt{\frac{||a||_{V_{\tilde{\pi}_l^*}^{-1}}^2}{g(\tilde{\pi}_l^*)}} \leq \epsilon_l.$$

Then, utilising the standard sub-gaussain concentration result for OLS estimator (see Chapter 20 in Lattimore & Szepesvári (2020) for a derivation):

$$\mathbb{P}(|\langle a, \theta^* - \hat{\theta}_l \rangle| \geq \sqrt{2||a||_{V^{-1}}^2 \log(1/\epsilon_0)}) \leq 2\epsilon_0$$

we have that

$$\mathbb{P}(|\langle a, \theta^* - \hat{\theta}_l \rangle| \geq \epsilon_l) \leq \mathbb{P}\left( |\langle a, \theta^* - \hat{\theta}_l \rangle| \geq \sqrt{2||a||_{V^{-1}}^2 \log(4l^2|\mathcal{A}|T)} \right)$$

$$\leq \frac{1}{2Tl^2|\mathcal{A}|}.$$

This concludes the proof of lemma. $\qquad\square$

Substituting the bound (27) into equation (26) we get:

$$\mathbb{P}\left( \bigcup_{l=1}^\infty \bigcup_{a \in \mathcal{A}_l} \{\xi_{a,l}^c(\mathcal{A}_l)\} \right) \leq \sum_{l=1}^\infty \sum_{\mathcal{V} \subseteq \mathcal{A}} \sum_{a \in \mathcal{V}} \frac{1}{2Tl^2|\mathcal{A}|} \mathbb{P}\left( \{\mathcal{A}_l = \mathcal{V}\} \right)$$

$$\leq \sum_{l=1}^\infty \frac{1}{2Tl^2}$$

$$\leq \frac{1}{T}.$$

Thus, for the rest of the regret bound proof we will work within the "clean" execution event: $\bigcap_{l=1}^\infty \bigcap_{a \in \mathcal{A}_l} \{\xi_{a,l}(\mathcal{A}_l)\}$.

**Step 2: In clean execution suboptimal arms are eliminated while the optimal arms aren't.**

If an optimal arm $a^*$ is in $\mathcal{A}_l$ then

$$\langle a - a^*, \hat{\theta}_l \rangle = \langle a, \hat{\theta}_l - \theta^* \rangle - \langle a^*, \hat{\theta}_l - \theta^* \rangle + \langle a - a^*, \theta^* \rangle$$

$$\leq 2\epsilon_l.$$

This means that $a^*$ is also in $\mathcal{A}_{l+1}$ in the good set. Thus by induction it is clear that in the good set the best arm is not eliminated.

For $a$ such that $\langle a^* - a, \theta^* \rangle > 4\epsilon_l$ we have that

$$\max_{a' \in \mathcal{A}_l} \langle a' - a, \hat{\theta}_l \rangle \geq \langle a^* - a, \hat{\theta}_l \rangle$$
$$= \langle a^*, \hat{\theta}_l - \theta^* \rangle - \langle a, \hat{\theta}_l - \theta^* \rangle + \langle a^* - a, \theta^* \rangle$$
$$> 2\epsilon_l,$$

and hence get eliminated after phase $l$ in the good set. This means in the next phase $l + 1$ we have the property that for all $a \in \mathcal{A}_{l+1}$ we have that $\langle a^* - a, \theta^* \rangle \leq 4\epsilon_l = 8\epsilon_{l+1}$. Thus, in phase $l$, all arms $a$ with a sub-optimality gap greater than $4\epsilon_l$ is eliminated.

A further consequence of the above result is that for $l \geq \log_2(8\Delta_{min}^{-1})$ we have $\mathcal{A}_l \subseteq \mathcal{A}^*$, i.e, only optimal arms survives.

**Step 3: Upper bounding the regret with the confidence width $d_{\text{eff}}$ and $|\text{supp}(\pi_{l,\text{on}}^*)|$.**

Let $n_{n,a}$ denote number of times arm $a$ has been pulled within the online horizon $T$. Then, within the good set the regret is upper bounded by:

$$\mathcal{R}(\text{OOPE}) = \sum_{a \in \mathcal{A} \setminus \{a^*\}: \Delta_a \leq v} \Delta_a n_{n,a} + \sum_{a \in \mathcal{A} \setminus \{a^*\}: \Delta_a > v} \Delta_a n_{n,a}$$
$$\leq vT + \sum_{a \in \mathcal{A} \setminus \{a^*\}: \Delta_a > v} \Delta_a n_{n,a}$$
$$\leq vT + \sum_{l=1}^{l_M} \sum_{a \in \mathcal{A} \setminus \{a^*\}: \Delta_a > v} \Delta_a n_{\text{on}}^l(a) \mathbb{1}_{\{a \in \mathcal{A}_l\}}.$$

where $l_M = \min(l_{max}, \log_2(8v^{-1}))$. This is because from Step 2 we know that any suboptimal arms that survive into phase $l$ has a sub-optimality gap of atmost $8\epsilon_l$, and $l \leq \log_2(8v^{-1})$. The other bound $l_{max}$ denotes an upper bound for the very last phase in which the online samples are exhausted. From Proposition 4.3 we know that the offline samples if they exhaust will happen only in the last phase as well. Thus, we can proceed by assuming the concentration result from Step 1 holds in all but the last phase. For the last phase, since there is no elimination, we can bound the regret from the last phase by assuming excess online and offline samples as per equation (4), even though in reality they would have been exhausted. This is so because the online regret can only increase with more online samples and the concentration does not matter in the last round since we don't have elimination in the last round.

Substituting the equation (4) for $n_{\text{on}}^l(a)$ we get

$$\mathcal{R}(\text{OOPE}) \leq vT + \sum_{l=1}^{l_M} \sum_{a \in \mathcal{A} \setminus \{a^*\}: \Delta_a > v} 8\epsilon_l \left\lceil \frac{3d_{\text{eff}} \pi_{l,\text{on}}^*(a)}{\epsilon_l^2} \log(4l^2 |\mathcal{A}|T) \right\rceil$$
$$\leq vT + \sum_{l=1}^{l_M} \frac{24 d_{\text{eff}}}{\epsilon_l} \log(4l^2 |\mathcal{A}|T) + \sum_{l=1}^{l_M} 8\epsilon_l |\text{supp}(\pi_{l,\text{on}}^*)|$$
$$\leq vT + 24 d_{\text{eff}} \log(4l_{max}^2 |\mathcal{A}|T) \left( \sum_{l=1}^{l_M} 2^l \right) + \sum_{l=1}^{l_M} 8\epsilon_l |\text{supp}(\pi_{l,\text{on}}^*)|$$

**Step 4: Upper bound on $|\text{supp}(\pi_{l,\text{on}}^*)|$.**

Now we shall show, just as in Kiefer-Wolfowitz theorem (see Theorem 21.1 in Lattimore & Szepesvári (2020)), there exists an optimizer $\pi_{l,\text{on}}^*$ such that $|supp(\pi_{l,\text{on}}^*)| \leq \frac{d(d+1)}{2}$. The Lagrangian for the concave optimization problem (2) is

$$\mathcal{L} = \log \left( \det \left( (1 - \alpha_l) \sum_a \pi_{l,\text{on}}(a) aa^t + \alpha_l \sum_a \pi_{\text{off}}(a) aa^t \right) \right) - \mu(\sum_a \pi_{l,\text{on}}(a) - 1) - \sum_a \lambda_a \pi_{l,\text{on}}(a).$$

where $\mu, \lambda_a$ are the multipliers. The Karush-Kuhn-Tucker conditions which are sufficient and necessary give:

$$||a||^2_{V_{\tilde{\pi}}^{-1}} - \mu - \lambda_a = 0,$$
$$\lambda_a \pi_{l,\text{on}}(a) = 0,$$
$$\sum_a \pi_{l,\text{on}}(a) = 1.$$

Here $\tilde{\pi}(a) = (1 - \alpha_\ell)\pi_{l,\text{on}}(a) + \alpha_\ell \pi_{\text{off}}(a)$. We have that if $a \in supp(\pi^*_{l,\text{on}})$ then $\lambda_a = 0$. Therefore, we observe that $\forall a \in supp(\pi_{l,\text{on}})$, we have

$$||a||^2_{V_{\tilde{\pi}^*_l}^{-1}} = \mu > 0.$$

Now suppose $|supp(\pi^*_{l,\text{on}})| > \frac{d(d+1)}{2}$ then as the dimension of space of symmetric matrices is only $\frac{d(d+1)}{2}$ we know there exists a $v$ such that

$$\sum_{a \in supp(\pi^*_{l,\text{on}})} v(a)aa^t = 0.$$

Then from the earlier observation we have

$$\mu \sum_{a \in supp(\pi^*_{l,\text{on}})} v(a) = \sum_{a \in supp(\pi^*_{l,\text{on}})} v(a)||a||^2_{V_{\tilde{\pi}^*_l}^{-1}} = Tr(V_{\tilde{\pi}^*_l}^{-1} \sum_{a \in supp(\pi^*_{l,\text{on}})} v(a)aa^t) = 0.$$

Thus $\sum_{a \in supp(\pi^*_{l,\text{on}})} v(a) = 0$. Then we notice that there exists a perturbation of $\pi^*_n$ with $v$, i.e, $\widehat{\pi^*_{l,\text{on}}} = \pi^*_{l,\text{on}} + tv$ such that some of support points have zero mass and still have an unchanged optimal objective value. And hence from $\pi^*_{l,\text{on}}$ we have produced a new optimal solution $\widehat{\pi^*_{l,\text{on}}}$ that has strictly smaller subset as the support. By induction on the support size we conclude that there exists a $\pi^*_{l,\text{on}}$ such that $|supp(\pi^*_{l,\text{on}})| \le \frac{d(d+1)}{2}$.

**Step 5: The final bound.**

Using the result in Step 4 we get a regret bound of

$$\mathcal{R}(\texttt{OOPE}) \le vT + 24d_{\text{eff}} \log(4l^2_{max}|\mathcal{A}|T) \left( \sum_{l=1}^{l_M} 2^l \right) + \sum_{l=1}^{l_M} 8\epsilon_l \frac{d(d+1)}{2}$$

$$\le vT + 48d_{\text{eff}} \log(4l^2_{max}|\mathcal{A}|T) \min \left\{ \frac{8}{v}, \sqrt{4 + \frac{T}{d_{\text{eff}} \log(4|\mathcal{A}|T)}} \right\} + 4d(d+1). \tag{28}$$

Optimizing over $v$ gives the bound:

$$\mathcal{R}(\texttt{OOPE}) \le 16\sqrt{6d_{\text{eff}}T \log(4l^2_{max}|\mathcal{A}|T)} + 4d(d+1), \tag{29}$$

under clean execution. But as we know from Step 1 that set arises with probability atleast $1 - \frac{1}{T}$ and thus the proof of Theorem 4.5 is complete.

## A.6 Proof of proposition 4.8.

We first derive a useful lemma that will help with the proof of proposition.

**Lemma A.2.** *We can upper bound the confidence width as:*

$$g(\tilde{\pi}^\star_l) \le \frac{1}{\alpha_l \pi_{off}(a_o) + (1 - \alpha_l)\pi^*_{l,on}(a_o)} \tag{30}$$

*for every $a_0 \in supp(\pi^*_{l,on})$. Furthermore, when $|\mathcal{A}| = d$ the inequality holds as an equality for each $a_0 \in supp(\pi^*_{l,on})$.*

*Proof.* Using the Cauchy-Binet formula (see chapter 3 in Tao (2012)) we have the following relation for $a_1, a_2, \ldots, a_n \in \mathbb{R}^d$ :

$$det\left(\sum_{i=1}^{n} a_i a_i^t\right) = \sum_{S \subset [n]:|S|=d} det\left(\sum_{a_i \in S} a_i a_i^t\right)$$

and the simple algebraic fact:

$$det\left(\sum_{i=1}^{d} c_i a_i a_i^t\right) = \left(\prod_{i=1}^{d} c_i\right) det\left(\sum_{i=1}^{d} a_i a_i^t\right)$$

whenever $c_i \geq 0$, for each $i \in [d]$. Combining these two facts one then has $\pi_l \in \Delta(\mathcal{A}_l)$:

$$det(\sum_{a \in \mathcal{A}} \alpha_l \pi_{\text{off}}(a) a a^t + \sum_{a \in \mathcal{A}_l} (1-\alpha_l)\pi_l(a) a a^t) = \sum_{\substack{|S|=d \\ S \subset \mathcal{A}}} \left(\prod_{a \in S \backslash \mathcal{A}_l} \alpha_l \pi_{\text{off}}(a)\right) \left(\prod_{a \in \mathcal{A}_l \cap S} (\alpha_l \pi_{\text{off}}(a) + (1-\alpha_l)\pi_l(a))\right) det\left(\sum_{a \in S} a a^t\right).$$

Using the identity $\frac{\partial}{\partial t}\Big|_{t=0} \log det(B + t u u^t) = \langle u, B^{-1} u \rangle$ we then have that for $a_o \in \mathcal{A}_l$:

$$||a_o||_{V_{\tilde{\pi}^\star l}}^2 = \frac{\sum_{\substack{|S|=d, a_o \in S \\ S \subset \mathcal{A}}} \left(\prod_{a \in S \backslash \mathcal{A}_l} \alpha_l \pi_{\text{off}}(a)\right) \left(\prod_{a \in \mathcal{A}_l \cap S \backslash \{a_o\}} (\alpha_l \pi_{\text{off}}(a) + (1-\alpha_l)\pi_{l,\text{on}}^*(a))\right) det\left(\sum_{a \in S} a a^t\right)}{\sum_{\substack{|S|=d \\ S \subset \mathcal{A}}} \left(\prod_{a \in S \backslash \mathcal{A}_l} \alpha_l \pi_{\text{off}}(a)\right) \left(\prod_{a \in \mathcal{A}_l \cap S} (\alpha_l \pi_{\text{off}}(a) + (1-\alpha_l)\pi_{l,\text{on}}^*(a))\right) det\left(\sum_{a \in S} a a^t\right)}.$$

Multiplying the above equation with $(\alpha_l \pi_{\text{off}}(a_o) + (1 - \alpha_l)\pi_{l,\text{on}}^*(a_o))$ we get that:

$$(\alpha_l \pi_{\text{off}}(a_o)+(1-\alpha_l)\pi_l^*(a_o))||a_o||_{V_{\tilde{\pi}^\star l}}^2 = \frac{\sum_{\substack{|S|=d, a_o \in S \\ S \subset \mathcal{A}}} \left(\prod_{a \in S \backslash \mathcal{A}_l} \pi_{\text{off}}(a)\right) \left(\prod_{a \in \mathcal{A}_l \cap S} (\alpha_l \pi_{\text{off}}(a) + (1-\alpha_l)\pi_l^*(a))\right) det\left(\sum_{a \in S} a a^t\right)}{\sum_{\substack{|S|=d \\ S \subset \mathcal{A}}} \left(\prod_{a \in S \backslash \mathcal{A}_l} \pi_{\text{off}}(a)\right) \left(\prod_{a \in \mathcal{A}_l \cap S} (\alpha_l \pi_{\text{off}}(a) + (1-\alpha_l)\pi_l^*(a))\right) det\left(\sum_{a \in S} a a^t\right)}.$$

We observe that the RHS of the above equality is a determinantal probability and hence less than 1. Now from the proof of Lemma 4.4 we know that if $a_0 \in supp(\pi_{l,\text{on}}^*)$ then $||a_o||_{V_{\tilde{\pi}_l^\star}}^2 = g(\tilde{\pi}_l^\star)$. Combining this we get that :

$$(\alpha_l \pi_{\text{off}}(a_o) + (1-\alpha_l)\pi_{l,\text{on}}^*(a_o))g(\tilde{\pi}_l^\star) \leq 1, \tag{31}$$

for all $a_0 \in supp(\pi_{l,\text{on}}^*)$.

If $|\mathcal{A}| = d$ then the above inequality is in fact an equality since $S = \mathcal{A}$ necessarily. This concludes the proof the lemma. $\qquad\square$

One can derive from the lemma's result 30 the following upper bound on the confidence width:

$$g(\tilde{\pi}_l^\star) \leq \max_{B \subset \mathcal{A}_l} \left\{ \frac{|B|}{\alpha_l \pi_{\text{off}}(B) + (1 - \alpha_l)} \right\}$$

**Proof of Proposition 4.8**:

*Proof.* As remarked above, for the MAB case with $|\mathcal{A}| = d$ we have that:

$$(\alpha_l \pi_{\text{off}}(a_o) + (1-\alpha_l)\pi_{l,\text{on}}^*(a_o))||a_o||_{V_{\tilde{\pi}^*}}^2 = 1$$

for every $a_0 \in supp(\pi_{l,\text{on}}^*)$. Combining this with the fact that $g(\tilde{\pi}^*)$ is equal for all support points of $\pi_{l,\text{on}}^*$ mean that if we can identify a set $S \subseteq \mathcal{A}_l$ with the following properties:

1. $\alpha_l \max_{a \in S} \pi_{\text{off}}(a) < \frac{\alpha_l \pi_{\text{off}}(S) + (1-\alpha_l)}{|S|}$.

2. $\frac{\alpha_l \pi_{\text{off}}(S) + (1-\alpha_l)}{|S|} \leq (1 - \alpha_l) + \alpha_l \min_{a \in S} \pi_{\text{off}}(a)$.

3. $\frac{\alpha_l \pi_{\text{off}}(S) + (1-\alpha_l)}{|S|} \leq \alpha_l \min_{a \in \mathcal{A}_l \setminus S} \pi_{\text{off}}(a)$

then $S$ is the support set of $\pi^*_{l,\text{on}}$ and for all $a \in S$ we have that:

$$\pi^*_{l,\text{on}}(a) = \frac{1}{|S|} + \frac{\alpha_l}{1 - \alpha_l}\left[\frac{\pi_{\text{off}}(S)}{|S|} - \pi_{\text{off}}(a)\right]$$

and

$$g(\tilde{\pi}^\star_l) = \frac{|S|}{\alpha_l \pi_{\text{off}}(S) + (1 - \alpha_l)}.$$

We shall now restrict our search of $S$ to sets of the following form:

$$S_i = \{a \in \mathcal{A}_l \mid \pi_{\text{off}}(a) \leq \pi_{\text{off}_{(i)}}\}$$

for some $i \in [r]$. Writing out the three properties to be satisfied by $S$ we see that for $S_i$ it translates into the following three inequalities:

$$\alpha_l \pi_{\text{off}_{(i)}} < \frac{\alpha_l(\sum_{j=1}^i m_j \pi_{\text{off}_{(j)}}) + (1 - \alpha_l)}{\sum_{j=1}^i m_j}$$

$$\frac{\alpha_l(\sum_{j=1}^i m_j \pi_{\text{off}_{(j)}}) + (1 - \alpha_l)}{\sum_{j=1}^i m_j} \leq (1 - \alpha_l) + \alpha_l \pi_{\text{off}_{(1)}}$$

$$\frac{\alpha_l(\sum_{j=1}^i m_j \pi_{\text{off}_{(j)}}) + (1 - \alpha_l)}{\sum_{j=1}^i m_j} \leq \alpha_l \pi_{\text{off}_{(i+1)}}$$

with the notation $\pi_{\text{off}_{(r+1)}} = \infty$. These inequalities can be straightforwardly re-written as:

$$\alpha_l < \frac{1}{1 + \sum_{j=1}^{i-1} m_j(\pi_{\text{off}_{(i)}} - \pi_{\text{off}_{(j)}})}$$

$$\alpha_l \leq \frac{1}{1 + \frac{\sum_{j=2}^{i-1} m_j(\pi_{\text{off}_{(j)}} - \pi_{\text{off}_{(1)}})}{(\sum_{j=1}^i m_j - 1)}}$$

$$\alpha_l \geq \frac{1}{1 + \sum_{j=1}^i m_j(\pi_{\text{off}_{(i+1)}} - \pi_{\text{off}_{(j)}})}.$$

We observe now that:

$$\sum_{j=1}^{i-1} m_j\big(\pi_{\text{off}_{(i)}} - \pi_{\text{off}_{(j)}}\big) \geq \sum_{j=1}^{i-1}\big(\pi_{\text{off}_{(j+1)}} - \pi_{\text{off}_{(j)}}\big) = \pi_{\text{off}_{(i)}} - \pi_{\text{off}_{(1)}}$$

$$\implies \Big(\sum_{j=1}^{i} m_j\Big)\left(\sum_{j=1}^{i-1} m_j\big(\pi_{\text{off}_{(i)}} - \pi_{\text{off}_{(j)}}\big)\right) \geq \Big(\sum_{j=1}^{i} m_j\Big)\big(\pi_{\text{off}_{(i)}} - \pi_{\text{off}_{(1)}}\big)$$

$$\iff \Big(\sum_{j=1}^{i} m_j - 1\Big)\left(\sum_{j=1}^{i-1} m_j\big(\pi_{\text{off}_{(i)}} - \pi_{\text{off}_{(j)}}\big)\right) \geq \Big(\sum_{j=1}^{i} m_j\Big)\big(\pi_{\text{off}_{(i)}} - \pi_{\text{off}_{(1)}}\big) - \left(\sum_{j=1}^{i-1} m_j\big(\pi_{\text{off}_{(i)}} - \pi_{\text{off}_{(j)}}\big)\right)$$

$$\iff \Big(\sum_{j=1}^{i} m_j - 1\Big)\left(\sum_{j=1}^{i-1} m_j\big(\pi_{\text{off}_{(i)}} - \pi_{\text{off}_{(j)}}\big)\right) \geq \sum_{j=1}^{i} m_j\big(\pi_{\text{off}_{(j)}} - \pi_{\text{off}_{(1)}}\big)$$

$$\iff \Big(\sum_{j=1}^{i} m_j - 1\Big)\left(\sum_{j=1}^{i-1} m_j\big(\pi_{\text{off}_{(i)}} - \pi_{\text{off}_{(j)}}\big)\right) \geq \sum_{j=2}^{i} m_j\big(\pi_{\text{off}_{(j)}} - \pi_{\text{off}_{(1)}}\big)$$

$$\iff \frac{1}{1 + \frac{\sum_{j=2}^{i-1} m_j\big(\pi_{\text{off}_{(j)}} - \pi_{\text{off}_{(1)}}\big)}{\big(\sum_{j=1}^{i} m_j - 1\big)}} \geq \frac{1}{1 + \sum_{j=1}^{i-1} m_j\big(\pi_{\text{off}_{(i)}} - \pi_{\text{off}_{(j)}}\big)}.$$

The last inequality then implies that for $S_i$ we have:

$$\frac{1}{1 + \sum_{j=1}^{i} m_j\big((\pi_{\text{off}_{(i+1)}} - \pi_{\text{off}_{(j)}})\big)} \leq \alpha_l < \frac{1}{1 + \sum_{j=1}^{i-1} m_j\big(\pi_{\text{off}_{(i)}} - \pi_{\text{off}_{(j)}}\big)}.$$

that is $\beta_{i+1} \leq \alpha_l < \beta_i$. The result follows from this, as the $\beta_i$'s split the interval $[0,1)$ into disjoint intervals, in which $\alpha_l$ must lie, since $0 \leq \alpha_l < 1$. The expression for $g(\tilde{\pi}_l^\star)$ is gotten by substituting $S = S_i$ whenever $\beta_{i+1} \leq \alpha_l < \beta_i$. □

### A.7 Minimax regret and lower bounds for it in presence of offline data.

A *problem instance* is an ordered quintuple $p := (\pi_{\text{off}}, \mathcal{A}, \theta, T_{\text{off}}, T)$ where $\pi_{\text{off}}$ is a measure on $\mathcal{A}$, $\mathcal{A} \subset R^d$, $\theta \in R^d$, and $T_{\text{off}}, T \in \mathbb{N}$. We define a *problem class* $\mathcal{P}$ to be a set of such problem instances $p$. In this work, we consider problem classes where all problem instances $p$ are such that the ratio of eigenvalues of the offline Grammian matrix $V_{\pi_{\text{off}}}$ to $\max_{a \in \mathcal{A}}||a||^2$ is held fixed, that is,

$$v_i = \frac{\lambda_i(V_{\pi_{\text{off}}})}{\max_{a \in \mathcal{A}}||a||^2}$$

for $i \in [d]$ is fixed. Further we impose the condition that $|\mathcal{A}| \leq (2d)^d$ for every instance in this class. Thus our problem classes are parametrized by $(d+3)$ parameters-$(d, v := (v_1, v_2, \ldots, v_d), T_{\text{off}}, T)$. We assume the *consistency condition* -

$$\sum_{i=1}^{d} v_i \leq 1$$

is satisfied by the parameter vector $v$ since:

$$\sum_{i=1}^{d} \lambda_i(V_{\pi_{\text{off}}}) = Tr(V_{\pi_{\text{off}}}) = \sum_{a \in \mathcal{A}} \pi_{\text{off}}(a)||a||^2 \leq \max_{a \in \mathcal{A}}||a||^2$$

$$\implies \sum_{i=1}^{d} v_i \leq 1.$$

We will denote such consistent problem classes as $\mathcal{P}^d_{v,T_{\text{off}},T}$.

We remark here that for any consistent set of parameters, the corresponding class $\mathcal{P}^d_{v,T_{\text{off}},T}$ is non-empty. We can just take the an action set consisting of scaled standard basis to see this.

Let us define the minimax regret value of a problem class as follows:

$$\mathcal{R}_{\text{minmax}}(\mathcal{P}^d_{v,T_{\text{off}},T}) := \inf_{S \in \mathcal{S}} \sup_{p \in \mathcal{P}^d_{v,T_{\text{off}},T}} \mathcal{R}_p(T,S), \tag{32}$$

where $S$ is an adaptive regret minimization algorithm for horizon $T$ which takes $T_{\text{off}}$ offline samples as input from a class $\mathcal{S}$ of such adaptive algorithms.

**Hard instance that lower bounds $\mathcal{R}_{\text{minmax}}(\mathcal{P}^d_{v,T_{\text{off}},T})$ and proof of Proposition 4.10.**

In this section we will construct a hard instance $p_0 \in \mathcal{P}^d_{v,T_{\text{off}},T}$ such that we can derive a lower bound to $\mathcal{R}_{\text{minmax}}(\mathcal{P}^d_{v,T_{\text{off}},T})$.

In this construction we assume that $\pi_{\text{off}}$ is such that $|supp(\pi_{\text{off}})| \leq (2d)^d$. We will assume our action set is of the form:

$$\mathcal{A} = \left\{ a \in R^d \middle| a_i \in \{\pm c_{1,i}, \pm c_{2,i}, \ldots, \pm c_{d,i}\}, \ \forall i \in [d] \right\} \tag{33}$$

where $c_{k,i} \in R$. We collect these $c_{k,i}$'s in a column vector and denote it as $C_i$ for each $i \in [d]$. The $C_i$'s will be chosen to optimize the lower bound. Let the set of possible $\theta$ (the unknown parameter) be:

$$\Theta = \{\theta \in R^d | \theta_i \in \{\pm\alpha_i\}\}$$

where we will choose $\alpha_i$ later on. We assume the noise is standard iid gaussian $\mathcal{N}(0,1)$.

Define the following sets for $k, i \in [d]$:

$$\mathcal{A}^+_{c_{k,i}} = \{a \in \mathcal{A} | a_i = |c_{k,i}|\}$$
$$\mathcal{A}^-_{c_{k,i}} = \{a \in \mathcal{A} | a_i = -|c_{k,i}|\}$$

*Remark* A.3. We make the following observation about the sets $\mathcal{A}^+_{c_{k,i}}, \mathcal{A}^-_{c_{k,i}}$:

1. In case for some $k, i \in [d]$ if $c_{k,i} = 0$ then we have $\mathcal{A}^+_{c_{k,i}} = \mathcal{A}^-_{c_{k,i}}$.

2. If $c_{k,i} \neq 0$, then we have that the sets $\mathcal{A}^+_{c_{k,i}}, \mathcal{A}^-_{c_{k,i}}$ are disjoint.

Now let us define a $d(d-1)/2$ collection $A^{ij}$ of $d \times d$ matrices indexed by the ordered pair $(i,j), i, j \in [d]$ with $i < j$ as follows:

$$A^{ij}_{k,l} = \pi_{\text{off}}(\mathcal{A}^+_{c_{k,i}} \cap \mathcal{A}^+_{c_{l,j}}) + \pi_{\text{off}}(\mathcal{A}^-_{c_{k,i}} \cap \mathcal{A}^-_{c_{l,j}}) - \pi_{\text{off}}(\mathcal{A}^+_{c_{k,i}} \cap \mathcal{A}^-_{c_{l,j}}) - \pi_{\text{off}}(\mathcal{A}^-_{c_{k,i}} \cap \mathcal{A}^+_{c_{l,j}}).$$

for $k, l \in [d]$.

Now we state a useful lemma:

**Lemma A.4.** *Given that $|supp(\pi_{off})| \leq (2d)^d$ over an action set of the form (33), we have that the following are equivalent:*

- *The offline Gram matrix $V_{\pi_o}$ (which is a function of both $\pi_{off}$ and $\mathcal{A}$) has the standard basis as its eigenvectors.*

- *The set of column vectors $C_i$, $i \in [d]$ is such that:*

$$C_i^t A^{ij} C_j = 0 \qquad (34)$$

*for all $i, j \in [d]$ and $i < j$.*

*Proof.* A necessary and sufficient condition for standard basis to be the eigenvectors is that offdiagonal entries of $V_{\pi_o}$ are zero, that is, for all $1 \leq i < j \leq d$:

$$
\begin{aligned}
0 &= (\sum_{\mathcal{A}} \pi_{\text{off}}(a) a a^t)_{i,j} \\
&= \sum_{\mathcal{A}} \pi_{\text{off}}(a)(a a^t)_{ij} \\
&= \sum_{k,l \in [d]} \sum_{a \in \mathcal{A}: |a_i| = |c_{k,i}|, |a_j| = |c_{l,j}|} \pi_{\text{off}}(a) a_i a_j \\
&\stackrel{(a)}{=} \sum_{k,l \in [d]} c_{k,i} c_{l,j} (\pi_{\text{off}}(\mathcal{A}^+_{c_{k,i}} \cap \mathcal{A}^+_{c_{l,j}}) + \pi_{\text{off}}(\mathcal{A}^-_{c_{k,i}} \cap \mathcal{A}^-_{c_{l,j}}) - \pi_{\text{off}}(\mathcal{A}^+_{c_{k,i}} \cap \mathcal{A}^-_{c_{l,j}}) - \pi_{\text{off}}(\mathcal{A}^-_{c_{k,i}} \cap \mathcal{A}^+_{c_{l,j}})) \\
&= \sum_{k,l \in [d]} c_{k,i} c_{l,j} A^{ij}_{k,l} \\
&= C_i^t A^{ij} C_j
\end{aligned}
$$

where $(a)$ follows from Remark A.3. This concludes the proof. $\qquad \square$

*Remark* A.5. If $c_{k,i} = 0$ then it is clear that $\forall j > i$ and $l \in [d]$ that $A^{ij}_{kl} = 0$.

**Lemma A.6.** *Suppose we have that $A \sim \pi_{off}$ and $\pi_{off}$ is such that it satisfies:*

1. *(Coordinate Independence). $\forall i, j, k, l \in [d]$*

$$\pi_{off}(A_i = c_{k,i}, A_j = c_{l,j}) = \pi_{off}(A_i = c_{k,i}) \pi_{off}(A_j = c_{l,j})$$

2. *($\pm$ Symmetricity). $\forall i, k \in [d]$*

$$\pi_{off}(A_i = c_{k,i}) = \pi_{off}(A_i = -c_{k,i})$$

*, that is, $\pi_{off}$ is independent co-ordinate wise and symmetrically distributed wrt to positive & negative values of a component $c_{k,i}$, then we have that $A^{ij} = 0$ for all $i < j$.*

*Proof.* From coordinate independence we have that:

$$
\begin{aligned}
\pi_{\text{off}}(\mathcal{A}^+_{c_{k,i}} \cap \mathcal{A}^+_{c_{l,j}}) &= \pi_{\text{off}}(\mathcal{A}^+_{c_{k,i}}) \pi_{\text{off}}(\mathcal{A}^+_{c_{l,j}}) \\
\pi_{\text{off}}(\mathcal{A}^-_{c_{k,i}} \cap \mathcal{A}^-_{c_{l,j}}) &= \pi_{\text{off}}(\mathcal{A}^-_{c_{k,i}}) \pi_{\text{off}}(\mathcal{A}^-_{c_{l,j}}) \\
\pi_{\text{off}}(\mathcal{A}^+_{c_{k,i}} \cap \mathcal{A}^-_{c_{l,j}}) &= \pi_{\text{off}}(\mathcal{A}^+_{c_{k,i}}) \pi_{\text{off}}(\mathcal{A}^-_{c_{l,j}}) \\
\pi_{\text{off}}(\mathcal{A}^-_{c_{k,i}} \cap \mathcal{A}^+_{c_{l,j}}) &= \pi_{\text{off}}(\mathcal{A}^-_{c_{k,i}}) \pi_{\text{off}}(\mathcal{A}^+_{c_{l,j}})
\end{aligned}
$$

for all $i, j, k, l \in [d]$. From $\pm$ symmetricity we have that:

$$\pi_{\text{off}}(\mathcal{A}^-_{c_{k,i}}) = \pi_{\text{off}}(\mathcal{A}^+_{c_{k,i}})$$

for all $i, k \in [d]$. Using these relations we get that:

$$
\begin{aligned}
A^{ij}_{k,l} &= \pi_{\text{off}}(\mathcal{A}^+_{c_{k,i}} \cap \mathcal{A}^+_{c_{l,j}}) + \pi_{\text{off}}(\mathcal{A}^-_{c_{k,i}} \cap \mathcal{A}^-_{c_{l,j}}) - \pi_{\text{off}}(\mathcal{A}^+_{c_{k,i}} \cap \mathcal{A}^-_{c_{l,j}}) - \pi_{\text{off}}(\mathcal{A}^-_{c_{k,i}} \cap \mathcal{A}^+_{c_{l,j}}) \\
&= \pi_{\text{off}}(\mathcal{A}^+_{c_{k,i}}) \pi_{\text{off}}(\mathcal{A}^+_{c_{l,j}}) + \pi_{\text{off}}(\mathcal{A}^-_{c_{k,i}}) \pi_{\text{off}}(\mathcal{A}^-_{c_{l,j}}) - \pi_{\text{off}}(\mathcal{A}^+_{c_{k,i}}) \pi_{\text{off}}(\mathcal{A}^-_{c_{l,j}}) - \pi_{\text{off}}(\mathcal{A}^-_{c_{k,i}}) \pi_{\text{off}}(\mathcal{A}^+_{c_{l,j}}) \\
&= \pi_{\text{off}}(\mathcal{A}^+_{c_{k,i}}) \pi_{\text{off}}(\mathcal{A}^+_{c_{l,j}}) + \pi_{\text{off}}(\mathcal{A}^+_{c_{k,i}}) \pi_{\text{off}}(\mathcal{A}^+_{c_{l,j}}) - \pi_{\text{off}}(\mathcal{A}^+_{c_{k,i}}) \pi_{\text{off}}(\mathcal{A}^+_{c_{l,j}}) - \pi_{\text{off}}(\mathcal{A}^+_{c_{k,i}}) \pi_{\text{off}}(\mathcal{A}^+_{c_{l,j}}) \\
&= 0,
\end{aligned}
$$

for every $i < j$ and $k, l \in [d]$. This shows that $A^{ij}$ is the zero matrix for this choice of $\pi_{\text{off}}$. $\qquad \square$

The upshot of the previous lemma is that if a measure $\pi_{\text{off}}$ satisfies the coordinate independence and $\pm$ symmetricity then by Lemma A.4 we know any choice of $C_i$ ensure that the eigenvectors of $V_{\pi_o}$ are the standard basis.

*Remark* A.7. It is straightforward to construct a $\pi_{\text{off}}$ which is coordinate independent and $\pm$ symmetric. Consider any $d$ mutually independent distributions on $[d]$. Denote them as $\pi_j$, $j \in [d]$ and $\pi_j \in \Delta_d$. Simply set $\pi_{\text{off}}(\mathcal{A}^+_{c_{k,i}} \cup \mathcal{A}^-_{c_{k,i}}) = \pi_i(k)$ for each $i, k \in [d]$. If it is the case that $c_{k,i} \neq 0$, then one further sets: $\pi_{\text{off}}(\mathcal{A}^+_{c_{k,i}}) = \pi_{\text{off}}(\mathcal{A}^-_{c_{k,i}}) = \pi_i(k)/2$ to ensure $\pm$ symmetricity. We note that coordinate wise independence and $\pm$ symmetricity are only partially restrictive of our choice of $\pi_{\text{off}}$, We still could make any arbitrary d-collection of $\pi_j$'s in the above construction.

Now we want the $\pi_{\text{off}}$ and $C_i$'s should satisfy for each $i \in [d]$:

$$v_i = \frac{\lambda_i}{\max\limits_{a \in \mathcal{A}} ||a||^2}. \tag{35}$$

We note that

$$\lambda_i = \sum_{k=1}^d \pi_{\text{off}}(\mathcal{A}^+_{c_{k,i}} \cup \mathcal{A}^-_{c_{k,i}})|c_{k,i}|^2$$

and

$$\max\limits_{a \in \mathcal{A}} ||a||^2 = \sum_{i=1}^d |c_{d,i}|^2$$

where we assume WLOG that $|c_{d,i}| \geq |c_{k,i}|$ for all $1 \leq k \leq (d-1)$ and $i \in [d]$. Re-writing the desired condition using the above equations we obtain:

$$\begin{aligned} v_i &= \frac{\lambda_i}{\max\limits_{a \in \mathcal{A}} ||a||^2} \\ &= \frac{\sum_{k=1}^d \pi_{\text{off}}(\mathcal{A}^+_{c_{k,i}} \cup \mathcal{A}^-_{c_{k,i}})|c_{k,i}|^2}{\sum_{i=1}^d |c_{d,i}|^2} \\ &= \frac{\sum_{k=1}^d \pi_i(k)|c_{k,i}|^2}{\sum_{i=1}^d |c_{d,i}|^2}. \end{aligned}$$

Now for any $w_i = \frac{|c_{d,i}|^2}{\sum_{i=1}^d |c_{d,i}|^2} \geq v_i$, it is clear from the above equation that there will always exist a $\pi_i \in \Delta_d$ and choice of $|c_{k,i}|^2 \leq |c_{d,i}|^2$ for $1 \leq k \leq (d-1)$ that satisfy condition equation (35).

Let $P_\theta, P_{\theta'}$ be the distribution on the offline+online samples for parameters $\theta$ and $\theta'$. Then we can decompose the KL divergence as:

$$\begin{aligned} D(P_\theta, P_{\theta'}) &= \mathbb{E}_\theta\Big[\sum_{t=-T_{\text{off}}}^0 D(\mathcal{N}(\langle A_t, \theta\rangle, 1), \mathcal{N}(\langle A_t, \theta'\rangle, 1))\Big] + \mathbb{E}_\theta\Big[\sum_{t=1}^T D(\mathcal{N}(\langle A_t, \theta\rangle, 1), \mathcal{N}(\langle A_t, \theta\rangle, 1))\Big] \\ &= \frac{1}{2}\sum_{t=-T_{\text{off}}}^0 (\langle A_t, \theta - \theta'\rangle)^2 + \frac{1}{2}\sum_{t=1}^T \mathbb{E}_\theta[(\langle A_t, \theta - \theta'\rangle)^2] \\ &= \frac{T_{\text{off}}}{2}||\theta - \theta'||^2_{V_{\pi_o}} + \frac{1}{2}\sum_{t=1}^T \mathbb{E}_\theta\big[||\theta - \theta'||^2_{A_t A_t^T}\big] \end{aligned}$$

where $A_t$ refers to the arm pull sequence and the randomness in the last equation is due to the sampling strategy $S$.

For $i \in [d]$ and $\theta \in \Theta$ we define

$$p_{\theta_i} := \mathbb{P}_\theta \left( \sum_{t=1}^T \mathbb{1}\{sgn(A_{ti} \neq sgn(\theta_i)\} \geq \frac{T}{2} \right).$$

Now let for every $\theta$ define $\tilde{\theta}_i \in \Theta$ such that $\tilde{\theta}_i = -\theta_i$ and $\forall j \neq i$, $\tilde{\theta}_j = \theta_j$.

We now describe our choice for $\alpha_i$:

$$\alpha_i = \frac{1}{|c_{d,i}|\sqrt{T + T_{\text{off}}\frac{v_i}{w_i}}}.$$

We apply Bretagnoulle-Huber lemma and use the definitions of $\alpha_i$ and the KL decomposition to get :

$$p_{\theta_i} + p_{\tilde{\theta}_i} \geq \frac{1}{2}exp\left( -\frac{T_{\text{off}}}{2}||\theta - \tilde{\theta}_i||^2_{V_{\pi_{\text{off}}}} - \frac{1}{2}\sum_{t=1}^T \mathbb{E}_\theta\left[||\theta - \tilde{\theta}_i||^2_{A_t A_t^T}\right]\right)$$

$$= \frac{1}{2}exp\left( -\frac{4\alpha_i^2 T_{\text{off}}\lambda_i(V_{\pi_{\text{off}}})}{2} - \frac{4\alpha_i^2}{2}\sum_{t=1}^T \mathbb{E}_\theta\left[A_{ti}^2\right]\right)$$

$$\overset{(a)}{\geq} \frac{1}{2}exp\left( -\frac{4\alpha_i^2 T_{\text{off}}\lambda_i(V_{\pi_{\text{off}}})}{2} - \frac{4T|c_{d,i}|^2\alpha_i^2}{2}\right)$$

$$\overset{(b)}{=} \frac{1}{2}exp\left( -2\alpha_i^2|c_{d,i}|^2\left(T + T_{\text{off}}\frac{v_i}{w_i}\right)\right)$$

$$= \frac{1}{2}exp(-2)$$

where $(a)$ follows because $|c_{d,i}| \geq |c_{k,i}|$ for $1 \leq k \leq (d-1)$ and $(b)$ follows because $w_i = \frac{|c_{d,i}|^2}{\max_{a \in \mathcal{A}}||a||^2}$.

Then we have for any $b \in R_+^d$ that

$$\sum_{\theta \in \Theta} \frac{1}{|\Theta|}\sum_{i=1}^d b_i p_{\theta_i} = \frac{1}{|\Theta|}\sum_{i=1}^d b_i \sum_{\theta \in \Theta} p_{\theta_i} \geq \frac{(\sum_{i=1}^d b_i)}{4}exp(-2).$$

In what follows we choose:

$$b_i = \frac{1}{\sqrt{T + T_{\text{off}}\frac{v_i}{w_i}}}$$

This guarantees the existence of a $\theta_0$ such that $\sum_{i=1}^d b_i p_{\theta_{0_i}} \geq \frac{(\sum_{i=1}^d b_i)}{4}exp(-2)$. Finally, we choose our problem instance $p_0 = (\pi_{\text{off}}, \mathcal{A}, \theta_0, T_{\text{off}}, T)$ where $\mathcal{A}$ is given by equation (33) and $\pi_{\text{off}}$ is a measure on this $\mathcal{A}$ that has coordinate independence and $\pm$ symmetricity. For this $p_0$ we have

$$\mathcal{R}_{p_0}(T, S) \overset{(a)}{=} \mathbb{E}_{\theta_0}\left[\sum_{t=1}^T \sum_{i=1}^d (|c_{d,i}|sgn(\theta_{0_i}) - A_{t,i})\theta_{0_i}\right]$$

$$= \sum_{i=1}^d \alpha_i \mathbb{E}_{\theta_0}\left[\sum_{t=1}^T (|c_{d,i}| + |A_{t,i}|)\mathbb{1}\{sgn(A_{ti}) \neq sgn(\theta_i)\} + (|c_{d,i}| - |A_{t,i}|)\mathbb{1}\{sgn(A_{ti}) = sgn(\theta_i)\}\right]$$

$$\geq \sum_{i=1}^d \frac{|c_{d,i}|}{|c_{d,i}|\sqrt{T + T_{\text{off}}\frac{v_i}{w_i}}}\mathbb{E}_{\theta_0}\left[\sum_{t=1}^T \mathbb{1}\{sgn(A_{ti}) \neq sgn(\theta_i)\}\right]$$

$$\geq \frac{T}{2}\sum_{i=1}^d \frac{1}{\sqrt{T + T_{\text{off}}\frac{v_i}{w_i}}}\mathbb{P}_{\theta_0}\left(\sum_{t=1}^T \mathbb{1}\{sgn(A_{ti} \neq sgn(\theta_i)\} \geq \frac{T}{2}\right)$$

$$\geq \frac{\sqrt{T}exp(-2)}{8}\sum_{i=1}^d \frac{1}{\sqrt{1 + \frac{T_{\text{off}}}{T}\frac{v_i}{w_i}}},$$

where $w_i \geq v_i$ for each $i$. $(a)$ follows from the fact the optimal arm has the same sign as $\theta_0$ in each coordinate. Now we know this construction works for every $w \in \Delta_d$ such that $w_i \geq v_i$. This implies:

$$\mathcal{R}_{p_0}(T,S) \geq \frac{\sqrt{T}\exp(-2)}{8} \sup_{\substack{w \in \Delta_d \\ \forall i, w_i \geq v_i}} \sum_{i=1}^{d} \frac{1}{\sqrt{1 + \frac{T_{\text{off}}}{T}\frac{v_i}{w_i}}}.$$

This gives a lower bound for the minimax value:

$$\mathcal{R}_{\text{minmax}}(\mathcal{P}_{v,T_{\text{off}},T}^d) \geq \frac{\sqrt{T}\exp(-2)}{8} \sup_{\substack{w \in \Delta_d \\ \forall i, w_i \geq v_i}} \sum_{i=1}^{d} \frac{1}{\sqrt{1 + \frac{T_{\text{off}}}{T}\frac{v_i}{w_i}}}$$

$$= \theta\left( \sqrt{T} \sup_{\substack{w \in \Delta_d \\ \forall i, w_i \geq v_i}} \sum_{i=1}^{d} \frac{1}{\sqrt{1 + \frac{T_{\text{off}}}{T}\frac{v_i}{w_i}}} \right) \tag{36}$$

**Properties of the lower bound equation (36).**

Let us define the following quantity:

$$d_e^{lb} := \sup_{\substack{w \in \Delta_d \\ \forall i, w_i \geq v_i}} \sum_{i=1}^{d} \frac{1}{\sqrt{1 + \frac{T_{\text{off}}}{T}\frac{v_i}{w_i}}}. \tag{37}$$

Then the lower bound equation (36) is just $\theta(\sqrt{T}d_e^{lb})$. The following are simple properties of $d_e^{lb}$ and the proof is omitted:

**Lemma A.8.** *We have that:*

1. *The optimization defining $d_e^{lb}$ in equation (37) is a concave program in $w$.*

2. *$d_e^{lb} \leq d$.*

3. *If $\sum_{i=1}^{d} v_i = 1$, $v_1 = v_2 = \cdots = v_{k-1} = 0$ and $0 < v_k \leq v_{k+1} \leq \cdots \leq v_d$, for some $k \in [d]$ then:*

$$d_e^{lb} = (k-1) + \frac{(d-k+1)}{\sqrt{1 + \frac{T_{off}}{T}}}$$

We provide a dual representation of $d_e^{lb}$ in the next lemma:

**Lemma A.9** (Dual representation of $d_e^{lb}$). *For simplicity assume that $0 < v_1 \leq \cdots \leq v_d$. Define for each $i \in [d]$, the following functions:*

$$\beta_i(x) = \begin{cases} v_i, & \text{for } x \leq \frac{-T_{off}}{2Tv_i(1+\frac{T_{off}}{T})^{3/2}} \\ z, & \text{for } \frac{-T_{off}}{2Tv_i(1+\frac{T_{off}}{T})^{3/2}} \leq x \leq \frac{-T_{off}v_i}{2T(1+\frac{T_{off}v_i}{T})^{3/2}} \\ 1, & \text{for } \frac{-T_{off}v_i}{2T(1+\frac{T_{off}v_i}{T})^{3/2}} \leq x \end{cases}$$

*where $z$ is the unique positive real root (which is guaranteed to exist exist under the condition on $x$) to the quartic polynomial:*

$$z\left(z + \frac{T_{off}v_i}{T}\right)^3 - \left(\frac{T_{off}v_i}{2Tx}\right)^2 = 0.$$

*We note that each $\beta_i$ are non-decreasing function of $x$. Then we have that:*

$$d_e^{lb} = \min_{x \in \mathbb{R}}\left\{ \sum_{i=1}^{d}\left( \beta_i(x)x + \frac{1}{\sqrt{1 + \frac{T_{off}v_i}{T\beta_i(x)}}} \right) - x \right\}. \tag{38}$$

*Furthermore, the minimizer $x^*$ (it need not be unique) for the dual problem is characterised by the necessary and sufficient condition:*

$$\sum_{i=1}^{d} \beta_i(x^*) = 1,$$

*with the unique primal optimizer $w^* = (\beta_1(x^*), \beta_2(x^*), \ldots, \beta_d(x^*))$.*

*Proof.* The basic idea is to use the following Fenchel Dual Theorem with appropriate choice of domain sets $C, D$ and $g$:

**Fenchel Duality.** *[Theorem 1, Section 7.12 Luenberger (1997)] Let $f$ and $g$ are convex and concave functionals defined over two convex sets $C$ and $D$ respectively. If $C \cap D$ has an point in the relative interior of both $C$ and $D$, with either the epigraph of $f$ over $C$ or $g$ over $D$ having a non empty interior then :*

$$\inf_{x \in C \cap D} f(x) - g(x) = \sup_{x^* \in C^* \cap D^*} g^*(x^*) - f^*(x^*)$$

*where $x^*$ is a dual vector, $C^*, D^*$ are the conjugate sets defined wrt $(f, C)$ and $(g, D)$ respectively and $f^*$, $g^*$ are the corresponding conjugate functionals (see the exact definitions in section 7.8-7.11 in Luenberger (1997)).*

To that end we make the following choices :

$$f(w) = -\sum_{i=1}^{d} \frac{1}{\sqrt{1 + \frac{T_{\text{off}}}{T} \frac{v_i}{w_i}}}$$

defined over the set $C = [v_1, 1] \times [v_2, 1] \times \cdots \times [v_d, 1]$ and $g$ is the zero function defined over the $d-$dimensional simplex $D = \Delta_d$. With these choices we observe that $C^* = D^* = C^* \cap D^* = R^d$ and $C \cap D$ is the appropriate domain for the optimization equation (37).

The particular choice of $C$ ensures it is straightforward to compute the convex conjugate $f^*$ (it simplifies into $d$ one dimensional optimizations due to the product structure of $C$):

$$f^*(x^*) = \sum_{i=1}^{d} \left( \beta_i(x_i^*) x_i^* + \frac{1}{\sqrt{1 + \frac{T_{\text{off}} v_i}{T \beta_i(x_i^*)}}} \right)$$

where $x^* \in \mathbb{R}^d$ and $x_i^*$ is its $i^{th}$ component. The functions $\beta_i : \mathbb{R} \to \mathbb{R}$ are non-decreasing positive valued functions on $R$ defined by:

$$\beta_i(x) = \begin{cases} v_i, & \text{for } x \leq \frac{-T_{\text{off}}}{2T v_i (1 + \frac{T_{\text{off}}}{T})^{3/2}} \\ z, & \text{for } \frac{-T_{\text{off}}}{2T v_i (1 + \frac{T_{\text{off}}}{T})^{3/2}} \leq x \leq \frac{-T_{\text{off}} v_i}{2T(1 + \frac{T_{\text{off}} v_i}{T})^{3/2}} \\ 1, & \text{for } \frac{-T_{\text{off}} v_i}{2T(1 + \frac{T_{\text{off}} v_i}{T})^{3/2}} \leq x \end{cases}$$

where $z$ is the unique positive real root (which is guaranteed to exist exist under the condition on $x$) to the quartic polynomial:

$$z \left( z + \frac{T_{\text{off}} v_i}{T} \right)^3 - \left( \frac{T_{\text{off}} v_i}{2Tx} \right)^2 = 0.$$

We further observe that $g^*$ for the zero function is given by:

$$g^*(x^*) = \min_{i \in [d]} x_i^*.$$

Applying the fenchel duality theorem (assuming the $v_i$ ensure the regularity conditions are satisfied) we have that:

$$\inf_{\substack{w\in\Delta_d \\ \forall i, w_i\geq v_i}} -\sum_{i=1}^{d}\frac{1}{\sqrt{1+\frac{T_{\text{off}}}{T}\frac{v_i}{w_i}}} = \sup_{x^*\in\mathbb{R}^d}\left\{\min_{i\in[d]}x_i^* - \sum_{i=1}^{d}\left(\beta_i(x_i^*)x_i^* + \frac{1}{\sqrt{1+\frac{T_{\text{off}}v_i}{T\beta_i(x_i^*)}}}\right)\right\}.$$

From this we have that:

$$\sup_{\substack{w\in\Delta_d \\ \forall i, w_i\geq v_i}} \sum_{i=1}^{d}\frac{1}{\sqrt{1+\frac{T_{\text{off}}}{T}\frac{v_i}{w_i}}} = \inf_{x^*\in\mathbb{R}^d}\left\{\sum_{i=1}^{d}\left(\beta_i(x_i^*)x_i^* + \frac{1}{\sqrt{1+\frac{T_{\text{off}}v_i}{T\beta_i(x_i^*)}}}\right) - \min_{i\in[d]}x_i^*\right\}.$$

Now utilizing the definitions of $\beta_i$ and the fact that it is a non-decreasing positive function one can conclude that :

$$\beta_i(x)x + \frac{1}{\sqrt{1+\frac{T_{\text{off}}v_i}{T\beta_i(x)}}}$$

is a non-decreasing function. This immediately implies the minimizer in the dual representation above is going to satisfy $x_i^* = \min_{i\in[d]}x_i^* = x$ for each $i$ and hence we have that:

$$\sup_{\substack{w\in\Delta_d \\ \forall i, w_i\geq v_i}} \sum_{i=1}^{d}\frac{1}{\sqrt{1+\frac{T_{\text{off}}}{T}\frac{v_i}{w_i}}} = \inf_{x\in\mathbb{R}}\left\{\sum_{i=1}^{d}\left(\beta_i(x)x + \frac{1}{\sqrt{1+\frac{T_{\text{off}}v_i}{T\beta_i(x)}}}\right) - x\right\}.$$

Observing that $f^*(x^*)$ is always convex, one concludes that the RHS is an unconstrained convex optimization. Differentiating and using the definitions of $\beta_i$ we can further conclude that the optimal $x$ satisfy the condition:

$$\sum_{i=1}^{d}\beta_i(x) = 1.$$

This concludes the proof. $\qquad\qquad\square$

Using primal and dual forms of $d_e^{lb}$ one may derive the following upper and lower bounds:

**Lemma A.10.** *For $k \in [d]$, such that $v_i = 0$ for $i < k$ and $0 < v_i$ for $i \geq k$, then we have:*

$$\frac{(1-\sum_i v_i)T_{off}}{2v_d(1+\frac{T_{off}}{T})^{3/2}T} + (k-1) + \frac{(d-k+1)}{\sqrt{1+\frac{T_{off}}{T}}} \geq d_e^{lb} \geq (k-1) + \frac{(d-k+1)}{\sqrt{1+\frac{T_{off}(\sum_i v_i)}{T}}}$$

Note that these bounds are tight if $\sum_i v_i \approx 1$ but can be loose if $\sum_i v_i << 1$.

## A.8   Upper and Lower bounds for $\mathcal{R}_{\mathsf{minmax}}(\mathcal{P}^d_{v,T_{\mathsf{off}},T})$.

Using Theorem 4.5 we see that the OOPE regret upper bound when $|\mathcal{A}| = (2d)^d$ and $T = \Omega(max(d\sqrt{T_{\text{off}}}, d^3))$[13] becomes

$$\mathcal{R}(\texttt{OOPE}) = \tilde{O}(\sqrt{d_{eff}dT})$$

---

[13]This ensures that $\tilde{O}(\sqrt{d_{eff}dT})$ is larger than $d^2$.

where $\tilde{O}$ suppresses logarithmic factors in $d, T, T_{\text{off}}$. To use this as an upper bound on $\mathcal{R}_{\text{minmax}}(\mathcal{P}^d_{v,T_{\text{off}},T})$ we will first bound $d_{eff}$ in terms of $v$:

$$d_{eff} = \min\left( \sum_{i=1}^{d} \frac{1}{1 + \frac{T_{\text{off}}}{T} \frac{\lambda_i(V_{\pi_{\text{off}}})}{\max_a ||a||^2}}, \frac{T}{T_{\text{off}}} g(\pi_{\text{off}}) \right)$$

$$= \min\left( \sum_{i=1}^{d} \frac{1}{1 + \frac{T_{\text{off}} v_i}{T}}, \frac{T}{T_{\text{off}}} g(\pi_{\text{off}}) \right)$$

$$= \min\left( \sum_{i=1}^{d} \frac{1}{1 + \frac{T_{\text{off}} v_i}{T}}, \frac{T}{T_{\text{off}}} \max_{a \in \mathcal{A}} \sum_{i=1}^{d} \frac{a_i^2}{\lambda_i(V_{\pi_o})} \right)$$

$$\leq \min\left( \sum_{i=1}^{d} \frac{1}{1 + \frac{T_{\text{off}} v_i}{T}}, \frac{T}{T_{\text{off}}} \max_{w \in \Delta_d} \sum_{i=1}^{d} \frac{w_i}{v_i} \right)$$

$$= \min\left( \sum_{i=1}^{d} \frac{1}{1 + \frac{T_{\text{off}} v_i}{T}}, \frac{T}{T_{\text{off}} v_1} \right)$$

Thus we have that

$$\tilde{O}\left( \sqrt{dT \min\left( \sum_{i=1}^{d} \frac{1}{1 + \frac{T_{\text{off}} v_i}{T}}, \frac{T}{T_{\text{off}} v_1} \right)} \right) \geq \mathcal{R}_{\text{minmax}}(\mathcal{P}^d_{v,T_{\text{off}},T}) \geq \theta(d_e^{lb} \sqrt{T}) \tag{39}$$

In the case where $v_i \geq \frac{c}{d}$ where $c (< 1)$ is a small constant, for all $i$ we have from the above bounds that:

$$\tilde{O}(dT/\sqrt{T_{\text{off}} c}) \geq V(\mathcal{P}^d_{v,T_{\text{off}},T}) \geq \theta(dT/\sqrt{T + T_{\text{off}}})$$

which is tight upto logarithmic factors when $T = o(T_{\text{off}})$ and $c$ is bounded away from zero.
In the case where $v_i = 0$ for all $i < k$ and $v_k > c/d$ for $i \geq k$, with $k = \Omega(d)$, $T = O(T_{\text{off}})$ and $T_{\text{off}}$ is large, we have that:

$$\tilde{O}(d\sqrt{T}) = \tilde{O}(\sqrt{dT(k-1)}) \geq \mathcal{R}_{\text{minmax}}(\mathcal{P}^d_{v,T_{\text{off}},T}) \geq \theta((k-1)\sqrt{T}) = \Omega(d\sqrt{T})$$

which is again tight upto logarithmic factors since $k \leq d$. Note that if $k = o(d)$ then:

$$\tilde{O}(\sqrt{dT(k-1)}) \geq \mathcal{R}_{\text{minmax}}(\mathcal{P}^d_{v,T_{\text{off}},T}) \geq \theta((k-1)\sqrt{T})$$

and hence there is a multiplicative gap of $\sqrt{d/k}$ between the upper and lower bounds.

**Summary**: The above calculations show that we are tight when all directions are well explored or if quite a large number of directions remain under-explored in the offline data. We remark that the there is a multiplicative gap of $O(\sqrt{d/k})$ between our current upper bound and lower bound in the regime where a few directions ($k = o(d)$ in the above example) are under-explored ($v_{k-1} = o(d^{-1})$) in the offline data.

### A.9 Regret bound for Warm Started LinUCB

In warm-started LinUCB with offline data we have the UCB-index defined as:

$$UCB_t(a) = \underset{\tilde{\theta} \in \mathcal{C}_t}{argmax} \ \langle \tilde{\theta}, a \rangle$$

where $\mathcal{C}_t = \{\tilde{\theta} : ||\tilde{\theta} - \hat{\theta}_t||_{V_t} \leq \beta_t\}$ is confidence interval. This usual elliptical confidence interval is warm started with offline data $V_t = \gamma I + T_{\text{off}} V_{\pi_{\text{off}}} + \sum_{s=1}^{t} a_s a_s^t$ and the algorithm plays the arm:

$$A_t \in \underset{a \in \mathcal{A}}{argmax} \ UCB_t(a).$$

**Proposition A.11.** *Assuming that for all $\theta \in \Theta$, we have $||\theta||_{V_0} \leq m$, where $m$ is some known constant, then regret for warm started LinUCB is given by :*

$$\mathcal{R}(\mathtt{UCB}) \leq \sqrt{8T\beta_T d_e^U \log\left(1 + \frac{T \max ||a||^2}{\lambda_1(V_0)}\right)}$$

*where*

$$\sqrt{\beta_t} = m + \sqrt{2\log(T) + 2d_e^U \log\left(1 + \frac{T \max ||a||^2}{\lambda_1(V_0)}\right)}$$

*and*

$$d_e^U := \max\left\{ i \in [d] : \quad (i-1)\lambda_i(V_0) \leq \frac{T \max ||a||^2}{\log\left(1 + \frac{T \max ||a||^2}{\lambda_1(V_0)}\right)} \right\}$$

*Here $V_0$ refers to Grammian matrix warm started using the entire offline data.*

*Remark* A.12. The proof utilizes a result of Valko et al. (2014), where we set $V_0 = T_{\text{off}} V_{\pi_{\text{off}}}$.

*Proof.* Let $V_t = V_0 + \sum_{s=1}^t A_s A_s^t$. Let the eigenvalues be $\delta_i + \nu_i$ and $\nu_i$, of $V_t$ and $V_0$ respectively. Then

$$\log\left(\frac{det(V_t)}{det(V_0)}\right) = \sum_{i=1}^d \log\left(1 + \frac{\delta_i}{\nu_i}\right)$$

$$\leq \sum_{i=1}^{d_e^U} \log\left(1 + \frac{T \max ||a||^2}{\nu_1}\right) + \frac{T \max ||a||^2}{\nu_{d_e^U + 1}}$$

$$\leq 2\sum_{i=1}^{d_e^U} \log\left(1 + \frac{T \max ||a||^2}{\nu_1}\right)$$

The first inequality follows from definition of $d_e^U$ and the fact that $\sum_i \delta_i \leq T \max ||a||^2$. The result then is established from the standard analysis of LinUCB (see Theorem 19.2, Lemma 19.4 and Theorem 20.4 in Lattimore & Szepesvári (2020)). $\quad\square$

## A.10 Example where warm started LinUCB bound is weaker than `OOPE`.

Consider the setting where $\mathcal{A} = \{\pm 1\}^d$ and $\Theta = \{\pm\sqrt{\frac{d}{(T_{\text{off}}+T)}}\}^d$. Let each arm be uniformly pulled in the offline data, that is, $\pi_{\text{off}}(a) = \frac{1}{|\mathcal{A}|}$. Then $\lambda_k(V_{\pi_{\text{off}}}) = 1$ for all $k$. Let $V_0 = T_{\text{off}} V_{\pi_{\text{off}}} + \gamma I$ (we choose $\gamma = O(T)$). Now:

$$||\theta||_{V_0}^2 = (T_{\text{off}} + \gamma)||\theta||_2^2$$
$$= \frac{d^2(T_{\text{off}} + \gamma)}{T_{\text{off}} + T}$$
$$\leq d^2$$

Hence we get that $m = d$ and $d_e^U \leq 2$. This implies $\sqrt{\beta_T} = O(d\sqrt{\log(T)})$. Therefore $\mathcal{R}(\mathtt{UCB}) \leq O(d\sqrt{T\log(T)\log(1 + \frac{Td}{T_{\text{off}}+\gamma})})$ which for $T_{\text{off}} \gg T$ simplifies to $\mathcal{R}(\mathtt{UCB}) \leq O(d^{3/2}T/\sqrt{T_{\text{off}} + \gamma})$.

We have the well-explored setting of offline data with $g_{\mathcal{A}}(\pi_{\text{off}}) = d$ and as a result `OOPE`'s regret bound becomes $\mathcal{R}(\mathtt{OOPE}) \leq O(\frac{dT}{\sqrt{T_{\text{off}}}} + d^2)$ which improves over LinUCB rate by the multiplicative factor of $\sqrt{d}$ and is important in moderate to low dimension ($d \leq 50$) regime.

# B    Proof of results in Section 5

**Motivating the dual:** We will adapt the analysis found in chapter 2 and 3 of Todd (2016). We introduce the *primal* optimization problem

$$\mathcal{P}(\mathcal{V}, \alpha, c) := \min_{H \succ 0} \quad -\log(\det(H))$$

$$\text{s.t} \quad (1-\alpha)a^t H a + \alpha Tr(V_{\pi_{\text{off}}} H) \leq c, \quad \forall a \in \mathcal{V} \subseteq \mathcal{A}.$$

We shall show this optimization has an optimizer and will be denoted by $H_\alpha^*(\mathcal{V}, c)$. We observe that $\mathcal{P}(\mathcal{A}_l, 0, d)$ is the standard MVEE problem for the set of "live" arms in phase $l$, with optimal solution denoted by $H_0^*(\mathcal{A}_l, d)$. Mostly we set $c = d$, in what follows. Let us try to motivate a duality for $\mathcal{P}(\mathcal{V}, \alpha, d)$. The Lagrangian for the minimization is

$$L(H, u) := -\log(\det(H)) + \sum_{a \in \mathcal{V}} u_a((1-\alpha)a^t H a + \alpha Tr(V_{\pi_{\text{off}}} H) - d)$$

for the multipliers $u_a \geq 0, \forall a \in X$. For any $H$ feasible we have

$$L(H, u) \leq -\log(\det(H)).$$

Differentiating wrt to H we get

$$\nabla_H L(H, u) = -H^{-1} + (1-\alpha) \sum_{a \in \mathcal{V}} u_a a a^t + \alpha V_{\pi_{\text{off}}}(\sum_a u_a)$$

Setting this to zero we get

$$H_\alpha^*(\mathcal{V}, d) = \left[(1-\alpha) \sum_{a \in \mathcal{V}} u_a a a^t + +\alpha V_{\pi_{\text{off}}}(\sum_a u_a)\right]^{-1}.$$

Substituting this back into the $L(H, u)$ we have

$$\min_H L(H, u) = \log\left(\det\left([(1-\alpha) \sum_{a \in \mathcal{V}} u_a a a^t + \alpha V_{\pi_{\text{off}}}(\sum_a u_a)]\right)\right) + d - (\sum_a u_a)d.$$

Lagrangian duality then tells us

$$\mathcal{P}(\mathcal{V}, \alpha, d) = \max_{u \geq 0} \quad \log\left(\det\left([(1-\alpha) \sum_{a \in \mathcal{V}} u_a a a^t + \alpha V_{\pi_{\text{off}}}(\sum_a u_a)]\right)\right) + d - (\sum_a u_a)d.$$

Now let $u_a = (\sum_a u_a)\pi(a)$, where $\pi(a)$ is from the probability simplex on $\mathcal{V}$. Then letting $\sum_a u_a = t$ we have

$$\mathcal{P}(\mathcal{V}, \alpha, d) = \max_{\pi \in \Delta(\mathcal{V})} \max_t \quad d(\log(t) - t) + d + \log\left(\det\left((1-\alpha)V_\pi + \alpha V_{\pi_{\text{off}}}\right)\right).$$

We observe $\log(t) - t$ is maximized at $t = 1$ and this gives the dual:

$$\mathcal{P}(\mathcal{V}, \alpha, d) = \max_{\pi \in \Delta(\mathcal{V})} \log\left(\det\left((1-\alpha)V_{\pi_n} + \alpha V_{\pi_{\text{off}}}\right)\right).$$

We make this rigorous in the next subsection.

## B.1   Proof of Lemma 5.1 (Strong Duality in the OO setting).

Define the *dual* problem as:

$$\mathcal{D}(\mathcal{V}, \alpha) := \max_{\pi \in \Delta(\mathcal{V})} \log\left(\det\left((1-\alpha)V_\pi + \alpha V_{\pi_{\text{off}}}\right)\right).$$

Let $d(\pi, \mathcal{V}, \alpha)$ denote the dual value for any feasible $\pi$ (or the information gain of a design $\pi$) in the dual problem and $p(H, \mathcal{V}, \alpha, d)$ denote the value for any feasible $H$ in the primal problem.

**Proposition B.1** (Weak duality). *For any primal feasible $H$ and dual feasible $\pi$ we have*

$$p(H, \mathcal{V}, \alpha, d) \geq d(\pi, \mathcal{V}, \alpha).$$

*Proof.* We have

$$p(H, \mathcal{V}, \alpha, d) - d(\pi, \mathcal{V}, \alpha) = -\log(\det(H)) - \log(\det((1-\alpha)V_\pi + \alpha V_{\pi_{\text{off}}}))$$
$$= -\log(\det(H((1-\alpha)V_\pi + \alpha V_{\pi_{\text{off}}})))$$

Now the matrix inside the log(det) function has the same eigenspectrum as a corresponding PD matrix. Let $\lambda_j$ denote its eigenvalues then

$$p(H, \mathcal{V}, \alpha, d) - d(\pi, \mathcal{V}, \alpha) = -\log(\det(H((1-\alpha)V_\pi + \alpha V_{\pi_{\text{off}}})))$$
$$= -\log(\prod_{j=1}^{d} \lambda_j)$$
$$= -d \quad \log((\prod_{j=1}^{d} \lambda_j)^{1/d})$$
$$\geq -d \log\left(\frac{\sum_j \lambda_j}{d}\right)$$
$$\geq 0.$$

The first inequality is AM-GM inequality while the second one is because

$$Tr(H((1-\alpha)V_\pi + \alpha V_{\pi_{\text{off}}})) = \sum_a \pi(a)((1-\alpha)a^t Ha + \alpha Tr(V_{\pi_{\text{off}}}H)) \leq d$$

and the primal feasibility of $H$. $\qquad\square$

We prove the strong duality next, that is

$$p(H_\alpha^*(\mathcal{V}, d), \mathcal{V}, \alpha, d) = d(\pi^*, \mathcal{V}, \alpha)$$

where $\pi^*$ is the optimal solution to $\mathcal{D}(\mathcal{V}, \alpha)$ and $H_\alpha^*(\mathcal{V}, d)$ the optimal solution to the primal $\mathcal{P}(\mathcal{V}, \alpha, d)$. Now we show the proof of strong duality.

***Proof of Strong Duality (Lemma 5.1).*** For $\epsilon(> 0)$ small enough we know that $\epsilon I$ is primal feasible. Thus we can restrict $H$ by adding the constraint $-\log(\det(H)) \leq -\log(\det(\epsilon I))$ without changing the optimization. This restriction means the $H$ must be strictly positive definite and cannot be arbitrarily close to semi-definiteness. Also with this restriction the feasible set has become closed.
Now as $\mathcal{A}$ is assumed to span the entire space, we assume there exists $\mu_j > 0$ such that $\mu_j e_j$ is a convex combination of $\{\pm a\}$. Thus as the ellipsoid is symmetric between $\pm a$ we have:

$$(1-\alpha)(\mu_j e_j)^t H(\mu_j e_j) + \alpha Tr(V_{\pi_{\text{off}}}H) \leq d$$
$$h_{jj}(1-\alpha)\mu_j^2 + \alpha Tr(V_{\pi_{\text{off}}}H) \leq d$$
$$\implies h_{jj} \leq \frac{d}{1-\alpha}.$$

This implies there is a uniform bound on the trace of $H$ and hence on the spectral norm. Thus the feasible region for $\mathcal{P}(\mathcal{V}, \alpha, d)$ is bounded. Thus, it is compact. As the functional is continuous in $H$, the minima is attained.
The uniqueness of the optima follows from the strict convexity of the objective.
Now we apply the Karush-Kuhn-Tucker (KKT) conditions to get:

$$-\tau H^{-1} + (1-\alpha)\sum_a u_a aa^t + \alpha V_{\pi_{\text{off}}} = 0,$$
$$u_a((1-\alpha)a^t Ha + \alpha Tr(V_{\pi_{\text{off}}}H) - d) = 0$$

for multipliers $\tau, u_a$. Multiplying the first equation with $H$ and taking trace we get:

$$-d\tau + (1-\alpha)\sum_a u_a a^t H a + \alpha \sum_a u_a Tr(V_{\pi_{\text{off}}} H) = 0$$

which with complementary slackness gives

$$-\tau d + d\left(\sum_a u_a\right) = 0. \implies \tau = \sum_a u_a$$

Suppose $u_a = 0$ for all $a$. Then the KKT conditions imply that $\alpha V_{\pi_{\text{off}}} = 0$ which is impossible. Thus we can set $\tau = 1$ by suitably scaling the multipliers $u_a$. Thus these $u_a$ are dual feasible. Further the KKT conditions imply

$$H_\alpha^*(\mathcal{V}, d) = \left((1-\alpha)\sum_a u_a aa^t + \alpha V_{\pi_{\text{off}}}\right)^{-1}$$

Further we note that

$$-\log(\det(H_\alpha^*(\mathcal{V}, d))) = \log\left(\det\left((1-\alpha)\sum_a u_a aa^t + \alpha V_{\pi_{\text{off}}}\right)\right)$$

that is the primal feasible $H_\alpha^*(\mathcal{V}, d)$ has the same primal objective value as the dual feasible $u$ for the dual objective. By weak duality then there is no duality gap and strong duality holds. $\square$

We characterize the optimality conditions for $\mathcal{P}(\mathcal{V}, \alpha, d)$ and $\mathcal{D}(\mathcal{V}, \alpha)$ in the following proposition

**Proposition B.2** (Optimality conditions). *Necessary and sufficient conditions for a PD matrix $H$ and $\pi$ to be optimal for $\mathcal{P}(\mathcal{V}, \alpha, d)$ and $\mathcal{D}(\mathcal{V}, \alpha)$ respectively are:*

- $\sum_a \pi(a) = 1$ *and* $(1-\alpha)a^t H a + \alpha V_{\pi_{off}} \leq d \ \forall a \in X$.
- $H = ((1-\alpha)\sum_a \pi(a)aa^t + \alpha V_{\pi_{off}})^{-1}$
- $(1-\alpha)a^t H a + \alpha V_{\pi_{off}} = d$ *whenever* $\pi(a) > 0$.

*Proof.* Condition (a) is just the primal and dual feasibility conditions. From strong duality we know that when $H$ and $\pi$ are optimal for the primal and dual respectively, necessarily and sufficiently only when there is no duality gap. But from the proof of weak duality we know this happens only when all the eigenvalues of $H((1-\alpha)\sum_a \pi(a)aa^t + \alpha V_{\pi_{\text{off}}})$ are equal and its trace is equal to $d$. This shows that $H((1-\alpha)\sum_a \pi(a)aa^t + \alpha V_{\pi_{\text{off}}}) = I$ and hence condition (b) follows.
Moreover from the identity below we have

$$Tr(H((1-\alpha)V_\pi + \alpha V_{\pi_{\text{off}}})) = \sum_a \pi(a)((1-\alpha)a^t H a + \alpha Tr(V_{\pi_{\text{off}}} H)) \leq d$$

But since $H$ is feasible it must be the case that $(1-\alpha)a^t H a + \alpha V_{\pi_{\text{off}}} = d$ whenever $\pi(a) > 0$ and hence (c) is also true. $\square$

### B.2 Algorithm for $O(d)$-initialization for Frank-Wolfe.

The initializing procedure for the Frank-Wolfe (FW) approximation used in section 5 of the main paper was first suggested in Betke & Henk (1993) and later adapted to the $D$-optimal design setting by Kumar & Yildirim (2005).

In the initialization procedure an arbitrary orthogonal direction $c$ to the vectors in set $B$ is chosen. Then the arm $a$ is added to the set $B$ such that it has the maximum projection (in absolute value) along the direction $c$. This inductive procedure then keeps adding arms to set $B$ until all the possible $d$ directions are exhausted.

Define the set $\underline{B} = \cup_{a \in B}\{a, -a\}$. Similarly define $\underline{A_l}$. This construction ensures that the set $\text{conv}(\underline{B})$ is contained in the $\text{conv}(\underline{A_l})$. Further, an induction argument shows the following result:

---

**Algorithm 3** O(d) initialization.

---

**Input:** $\mathcal{A}_l$.
$c \leftarrow e_1$, $B \leftarrow \emptyset$.
**for** $i \in 1 : d$ **do**
    $a \leftarrow \underset{a \in \mathcal{A}_l}{\text{argmax}} |\langle c, a \rangle|$.
    $B \leftarrow B \cup \{a\}$.
    $c \leftarrow$ non-zero vector from orthogonal complement of $B$.
**end for**
$\pi_l^{(0)} \leftarrow Unif(B)$.
**return:** $\pi_l^{(0)}$.

---

**Proposition B.3** (Theorem 2 in Betke & Henk (1993)). *Under the initialization procedure above we have the following bounds:*

$$\text{vol}(\text{conv}(\underline{\mathcal{A}_l})) \geq \text{vol}(\text{conv}(\underline{B})) \geq \frac{1}{d!} \text{vol}(\text{conv}(\underline{\mathcal{A}_l})).$$

The above relation will be used in the proof of Proposition 5.2. Intuitively, the bound in Proposition B.3 is true because $B$ is a representative subset of $\mathcal{A}_l$.

### B.3 Proof of Initialization Gap (Proposition 5.2).

Recall from Definition 5.3 in section 2 that $H(\pi) = ((1 - \alpha) \sum_a \pi(a) aa^t + \alpha V_{\pi_{\text{off}}})^{-1}$. We now introduce the notion of $\epsilon$-feasibility that proves useful in analyzing Algorithm 2:

**Definition B.4.** A dual feasible $\pi$ is said to $\epsilon$-primal feasible if $H(\pi)$ satisfies, $\forall a \in \mathcal{V}$,

$$(1 - \alpha) a^t H(\pi) a + \alpha Tr(H(\pi) V_{\pi_{\text{off}}}) \leq (1 + \epsilon) d$$

and if moreover, $\forall a$ such that $\pi(a) > 0$, it satisfies

$$(1 - \alpha) a^t H(\pi) a + \alpha Tr(H(\pi) V_{\pi_{\text{off}}}) \geq (1 - \epsilon) d$$

we call $\pi$ $\epsilon$-approximately optimal.

Now we give the following simple bound

**Proposition B.5** (Dual Bound). *If $\pi$ is $\epsilon$-primal feasible then $\pi$ is dual feasible and $(1 + \epsilon)^{-1} H(\pi)$ is primal feasible and both are within $d \log(1 + \epsilon)$ of their optimal value.*

*Proof.* We have

$$\begin{aligned}
p\left((1 + \epsilon)^{-1} H(\pi), \mathcal{V}, \alpha, d\right) - d(\pi, \mathcal{V}, \alpha) &= d \log(1 + \epsilon) + \log\left(\det\left((1 - \alpha) \sum_a \pi(a) aa^t + \alpha V_{\pi_{\text{off}}}\right)\right) \\
&\quad - \log\left(\det((1 - \alpha) \sum_a \pi(a) aa^t + \alpha V_{\pi_{\text{off}}})\right) \\
&= d \log(1 + \epsilon)
\end{aligned}$$

But as $p\left((1 + \epsilon)^{-1} H(\pi), \mathcal{V}, \alpha, d\right) \geq p(H_\alpha^*(\mathcal{V}, d), \mathcal{V}, \alpha, d) = d(\pi^*, \mathcal{V}, \alpha) \geq d(\pi, \mathcal{V}, \alpha)$ from strong duality we get the desired result. $\square$

We recall the following definition of slack given in section 5 where $\tilde{\pi} = (1 - \alpha)\pi + \alpha \pi_{\text{off}}$:

$$\delta(\pi) = \frac{(1 - \alpha) g_{\mathcal{A}_l}(\tilde{\pi}) + \alpha \sum_{a \in \mathcal{A}} \pi_{\text{off}}(a) \|a\|^2_{V_{\tilde{\pi}}^{-1}}}{d} - 1.$$

then we have the following proposition:

**Proposition B.6.** *If $\pi$ is uniform over a subset of $\mathcal{V}$ with size $m$, then $\delta(\pi) \leq m - 1$ and $d(\pi^*, \mathcal{V}, \alpha) - d(\pi, \mathcal{V}, \alpha) \leq d \ \log(m)$.*

*Proof.* As $\pi$ is uniform we have $\sum_a \pi(a) w_a(\pi) = d$ (recall $w_a, w_{a^+}$ were defined in Line 2,3 of Algorithm 2) and hence $w_{a_+} \leq dm$. As $\delta(\pi) = \frac{w_{a+}}{d} - 1$, we have $\delta(\pi) \leq m - 1$. Then the dual bound in Proposition B.5 gives that $d(\pi^*, \mathcal{V}, \alpha) - d(\pi, \mathcal{V}, \alpha) \leq d \ \log(m)$. $\qquad\square$

We next define feasibility relation between the primal problem in the purely online setting and in the online with offline setting:

**Lemma B.7** (Feasibility relation)**.** *We have the following two feasibility results when $\alpha \in [0, 1)$:*

1. *The optimal solution $H_\alpha^*(\mathcal{V}, d)$ to the primal problem $\mathcal{P}(\mathcal{V}, \alpha, d)$ is feasible for the primal problem $\mathcal{P}\left(\mathcal{V}, 0, \frac{d - \alpha Tr(H_\alpha^*(\mathcal{V}, d) V_{\pi_{off}})}{1 - \alpha}\right)$.*

2. *The optimal solution $H_0^*(\mathcal{V}, d)$ to the primal problem $\mathcal{P}(\mathcal{V}, 0, d)$ is feasible for the primal problem $\mathcal{P}(\mathcal{V}, \alpha, d)$.*

*Proof.* (1) As $H_\alpha^*(\mathcal{V}, d)$ is feasible for $\mathcal{P}(\mathcal{V}, \alpha, d)$ we have that

$$(1 - \alpha) a^t H_\alpha^*(\mathcal{V}, d) a + \alpha Tr(H_\alpha^*(\mathcal{V}, d) V_{\pi_{off}}) \leq d$$

for all $a \in \mathcal{V}$. Thus by simple manipulation of terms we have

$$a^t H_\alpha^*(\mathcal{V}, d) a \leq \frac{d - \alpha Tr(H_\alpha^*(\mathcal{V}, d) V_{\pi_{off}})}{1 - \alpha}.$$

We observe that $d > \alpha Tr(H_\alpha^*(\mathcal{V}, d) V_{\pi_{off}})$ and $1 - \alpha > 0$, and as $H_\alpha^*(\mathcal{V}, d) V_{\pi_{off}} \succ 0$ we see that $H_\alpha^*(\mathcal{V}, d)$ is feasible for $\mathcal{P}\left(\mathcal{V}, 0, \frac{d - \alpha Tr(H_\alpha^*(\mathcal{V}, d) V_{\pi_{off}})}{1 - \alpha}\right)$ and hence (1) is proved.

(2) We note that $\mathcal{P}(\mathcal{V}, 0, d)$ is the standard MVEE problem. Thus its optimal solution $H_0^*(\mathcal{V}, d)$ satisfies

$$a^t H_0^*(\mathcal{V}, d) a \leq d$$

for all $a \in \mathcal{V}$. Using this and the identity $Tr(H_0^*(\mathcal{V}, d) V_{\pi_{off}}) = \sum_a \pi_{off}(a) a^t H_0^*(\mathcal{V}, d) a$ we get that

$$
\begin{aligned}
&(1 - \alpha) a^t H_0^*(\mathcal{V}, d) a + \alpha Tr(H_0^*(\mathcal{V}, d) V_{\pi_{off}}) \\
\leq &(1 - \alpha) d + \alpha \sum_a \pi_{off}(a) a^t H_0^*(\mathcal{V}, d) a \\
\leq &(1 - \alpha) d + \alpha \sum_a \pi_{off}(a) d \\
= \ & d
\end{aligned}
$$

Using the fact that $H_0^*(\mathcal{V}, d) \succ 0$ we have that $H_0^*(\mathcal{V}, d)$ is feasible for $\mathcal{P}(\mathcal{V}, \alpha, d)$ and hence (2) is proved. $\quad\square$

We note that since $V_{\pi_{off}}$ is non-singular then the primal problem $\mathcal{P}(\mathcal{V}, 1, d)$ has well defined solution $H_1^*(\mathcal{V}, d) = (V_{\pi_{off}})^{-1}$. If it isn't then the optimal objective value is $-\infty$ with no optimizer. In the case where the optimizer is well-defined we can straightforwardly extend the above Lemma B.7 for the case $\alpha = 1$.

Let us show the invariance to scale $c$ for the volume of MVEE

**Lemma B.8** (Scale invariance of MVEEs)**.** *For all $c > 0$ we have that*

$$vol(\xi(H_0^*(\mathcal{V}, c), c)) = vol(\xi(H_0^*(\mathcal{V}, d), d)).$$

*Proof.* We observe that $\frac{d}{c}H_0^*(\mathcal{V}, c)$ is feasible for $\mathcal{P}(\mathcal{V}, 0, d)$ and $\frac{c}{d}H_0^*(\mathcal{V}, d)$ is feasible for $\mathcal{P}(\mathcal{V}, 0, c)$. Thus we have

$$-\log(\det(\frac{d}{c}H_0^*(\mathcal{V}, c))) \geq -\log(\det(H_0^*(\mathcal{V}, d)))$$

$$-\log(\det(\frac{c}{d}H_0^*(\mathcal{V}, d))) \geq -\log(\det(H_0^*(\mathcal{V}, c))).$$

From these we get the inequalities,

$$\left(\frac{d}{c}\right)^d \det(H_0^*(\mathcal{V}, c)) \leq \det(H_0^*(\mathcal{V}, d))$$

and

$$\left(\frac{c}{d}\right)^d \det(H_0^*(\mathcal{V}, d)) \leq \det(H_0^*(\mathcal{V}, c)).$$

These then imply

$$\left(\frac{c}{d}\right)^d \det(H_0^*(\mathcal{V}, d)) = \det(H_0^*(\mathcal{V}, c)).$$

By the volume formula for ellipsoids we have :

$$\begin{aligned}
vol(\xi(H_0^*(\mathcal{V}, c), c)) &= \frac{c^{d/2}B_d}{\sqrt{\det(H_0^*(\mathcal{V}, c))}} \\
&= \frac{c^{d/2}B_d}{\sqrt{\left(\frac{c}{d}\right)^d \det(H_0^*(\mathcal{V}, d))}} \\
&= \frac{d^{d/2}B_d}{\sqrt{\det(H_0^*(\mathcal{V}, d))}} \\
&= vol(\xi(H_0^*(\mathcal{V}, d), d)).
\end{aligned}$$

$\square$

Now we are ready to give a proof of Proposition 5.2:

**Proof of Proposition 5.2.** In light of the formula for a volume of an ellipsoid $\xi(H, c)$,

$$\mathrm{vol}(\xi(H, c)) = \frac{c^{d/2}B_d}{\sqrt{\det(H)}},$$

it is clear that the primal problem $\mathcal{P}(\mathcal{V}, \alpha, d)$ is equivalent to minimizing the volume of the ellipsoid $\xi(H, d)$ with constraint on $H$ such that $(1-\alpha)a^t Ha + \alpha Tr(HV_{\pi_o}) \leq d$ for all $a \in \mathcal{V}$.

Setting $\mathcal{V} = B$, where $B$ is the support of initialization procedure described in section B.2 for the set $\mathcal{A}_l$, from Lemma B.7 part (1) we get:

$$-\log(\det(H_\alpha^*(B, d))) \geq -\log\left(\det\left(H_0^*\left(B, \frac{d - \alpha Tr(H_\alpha^*(B, d)V_{\pi_{\mathrm{off}}})}{1-\alpha}\right)\right)\right)$$

which implies then that

$$\begin{aligned}
vol\left(\xi\left(H_\alpha^*(B, d), \frac{d - \alpha Tr(H_\alpha^*(B, d)V_{\pi_{\mathrm{off}}})}{1-\alpha}\right)\right) \geq \\
vol\left(\xi\left(H_0^*\left(B, \frac{d - \alpha Tr(H_\alpha^*(B, d)V_{\pi_{\mathrm{off}}})}{1-\alpha}\right), \frac{d - \alpha Tr(H_\alpha^*(B, d)V_{\pi_{\mathrm{off}}})}{1-\alpha}\right)\right).
\end{aligned}$$

Now from scale invariance of Lemma B.8 we know that

$$vol\left(\xi\left(H_0^*\left(B, \frac{d - \alpha Tr(H_\alpha^*(B,d)V_{\pi_{\text{off}}})}{1-\alpha}\right), \frac{d - \alpha Tr(H_\alpha^*(B,d)V_{\pi_{\text{off}}})}{1-\alpha}\right)\right) = vol(\xi(H_0^*(B,d),d))$$

and from the volume formula for ellipsoids we have the fact that

$$vol\left(\xi\left(H_\alpha^*(B,d), \frac{d - \alpha Tr(H_\alpha^*(B,d)V_{\pi_{\text{off}}})}{1-\alpha}\right)\right) =$$

$$\left(\frac{d - \alpha Tr(H_\alpha^*(B,d)V_{\pi_{\text{off}}})}{d(1-\alpha)}\right)^{d/2} vol(\xi(H_\alpha^*(B,d),d))$$

using which we get the inequality

$$\left(\frac{d - \alpha Tr(H_\alpha^*(B,d)V_{\pi_{\text{off}}})}{d(1-\alpha)}\right)^{d/2} vol(\xi(H_\alpha^*(B,d),d)) \geq vol(\xi(H_0^*(B,d),d)).$$

Now as the $\xi(H_0^*(B,d),d)$ is the MVEE of $B$ we have that $vol(\xi(H_0^*(B,d),d)) \geq vol(\text{conv}(\underline{B}))$, where $\underline{B}$ is as defined in Appendix B.2 and hence

$$\left(\frac{d - \alpha Tr(H_\alpha^*(B,d)V_{\pi_{\text{off}}})}{d(1-\alpha)}\right)^{d/2} vol(\xi(H_\alpha^*(B,d),d)) \geq vol(\text{conv}(\underline{B})).$$

As remarked in in Appendix B.2, the $O(d)$ initialization procedure there has the property $vol(\text{conv}(\underline{B})) \geq \frac{1}{d!}vol(\text{conv}(\underline{\mathcal{A}_l}))$ we get that

$$\left(\frac{d - \alpha Tr(H_\alpha^*(B,d)V_{\pi_{\text{off}}})}{d(1-\alpha)}\right)^{d/2} vol(\xi(H_\alpha^*(B,d),d)) \geq \frac{1}{d!}vol(\text{conv}(\underline{\mathcal{A}_l})).$$

Now John's theorem (see Theorem 1.1 in Todd (2016)) on MVEE states that for any finite set $\mathcal{C}$ we have $\frac{1}{d}\text{MVEE}(\underline{\mathcal{C}}) \subset \text{conv}(\underline{\mathcal{C}})$ and using the fact that $\text{MVEE}(\underline{\mathcal{C}}) = \text{MVEE}(\mathcal{C})$ gives us

$$\left(\frac{d - \alpha Tr(H_\alpha^*(B,d)V_{\pi_{\text{off}}})}{d(1-\alpha)}\right)^{d/2} vol(\xi(H_\alpha^*(B,d),d)) \geq \frac{1}{d!d^d}vol(\xi(H_0^*(\mathcal{A}_l,d),d)).$$

Now from Lemma B.7 part 2, with $\mathcal{V} = \mathcal{A}_l$, we have that

$$-\log(\det(H_0^*(\mathcal{A}_l,d))) \geq -\log(\det(H_\alpha^*(\mathcal{A}_l,d)))$$

and hence $vol(\xi(H_0^*(\mathcal{A}_l,d),d)) \geq vol(\xi(H_\alpha^*(\mathcal{A}_l,d),d))$. This give us

$$\left(\frac{d - \alpha Tr(H_\alpha^*(B,d)V_{\pi_{\text{off}}})}{d(1-\alpha)}\right)^{d/2} vol(\xi(H_\alpha^*(B,d),d)) \geq \frac{1}{d!d^d}vol(\xi(H_\alpha^*(\mathcal{A}_l,d),d)).$$

Now using the fact that $Tr(H_\alpha^*(B,d)V_{\pi_{\text{off}}}) \geq 0$ we get that

$$\frac{1}{(1-\alpha)^{d/2}} \geq \left(\frac{d - \alpha Tr(H_\alpha^*(B,d)V_{\pi_{\text{off}}})}{d(1-\alpha)}\right)^{d/2}.$$

Thus, we have

$$vol(\xi(H_\alpha^*(B,d),d)) \geq \frac{(1-\alpha)^{d/2}}{d!d^d}vol(\xi(H_\alpha^*(\mathcal{A}_l,d),d)).$$

We have finally managed to connect the optimal ellipsoids of the initialized set $B$ and the overall set $\mathcal{A}_l$. We have:

$$d(\pi_{l,\text{on}}^*, \mathcal{A}_l, \alpha) - d(\pi_l^{(0)}, B, \alpha) = d(\pi_{l,\text{on}}^*, \mathcal{A}_l, \alpha) - d(\pi^*(B), B, \alpha) + d(\pi^*(B), B, \alpha) - d(\pi_l^{(0)}, B, \alpha).$$

where $\pi^*(B)$ is the optimizer to the dual problem $\mathcal{D}(B, \alpha)$ and that $\pi^*_{l,\text{on}}$ is the optimizer of $\mathcal{D}(\mathcal{A}_l, \alpha)$ (recall the definition of $\pi^*_{l,\text{on}}$ from equation (2)). Then by proposition B.6 we have that $d(\pi^*(B), B, \alpha) - d(\pi_l^{(0)}, B, \alpha) \le d \ln(d)$.

Next we try to bound $d(\pi^*_{l,\text{on}}, \mathcal{A}_l, \alpha) - d(\pi^*(B), B, \alpha)$. From the volume of an ellipsoid formula and strong duality we have the following identities:

$$d(\pi^*(B), B, \alpha) = 2\log(\text{vol}(\xi(H^*_\alpha(B, d), d))) - d \ \log(d) - 2\log(B_d)$$
$$d(\pi^*_{l,\text{on}}, \mathcal{A}_l, \alpha) = 2\log(\text{vol}(\xi(H^*_\alpha(\mathcal{A}_l, d), d))) - d \ \log(d) - 2\log(B_d).$$

where as before $B_d$ is the volume of the unit ball in $\mathbb{R}^d$. Thus

$$d(\pi^*_{l,\text{on}}, \mathcal{A}_l, \alpha) - d(\pi^*(B), B, \alpha) = 2\log\left(\frac{\text{vol}(\xi(H^*_\alpha(\mathcal{A}_l, d), d))}{\text{vol}(\xi(H^*_\alpha(B, d), d))}\right).$$

But we know the ratio of volumes is upper bounded by $\frac{d^d d!}{(1-\alpha)^{d/2}}$ and thus we have

$$d(\pi^*_{l,\text{on}}, \mathcal{A}_l, \alpha) - d(\pi^*(B), B, \alpha) \le 2\log\left(\frac{d^d d!}{(1-\alpha)^{d/2}}\right)$$
$$\le 4d \ \log(d) - d\log(1-\alpha).$$

Thus we have

$$d(\pi^*_{l,\text{on}}, \mathcal{A}_l, \alpha) - d(\pi^*(B), B, \alpha) \le (d \ \log(d)) + (4d \ \log(d) - d\log(1-\alpha))$$
$$= 5d \ \log(d) - d \ \log(1-\alpha).$$

This concludes the proof. $\qquad\qquad\qquad\qquad\qquad\qquad\qquad\qquad\qquad\qquad\qquad\qquad\qquad\qquad\qquad\square$

### B.4   Proof of Lemma 5.4.

*Proof.* The update of FW in algorithm 2 is given by

$$\pi_l^{(t+1)} = (1+\beta)^{-1}(\pi_l^{(t)} + \beta \mathbb{1}_{\{a_+\}})$$

where $\beta = \frac{(w_{a_+} - d)}{(d-1)w_{a_+}}$, $a_+ = \underset{a}{\arg\max} \ \ w_a$ and $w = \left(Tr\left(H(\pi_l^{(t)})\left((1-\alpha)aa^t + \alpha V_{\pi_{\text{off}}}\right)\right)\right)_{a\in\mathcal{A}_l}$.

Then by matrix determinant lemma we have

$$d(\pi_l^{(t+1)}, \mathcal{A}_l, \alpha) = \log \ \det\left((1-\alpha)\frac{(V_{\pi_l^{(t)}} + \beta a_+ a_+^t)}{(1+\beta)} + \alpha V_{\pi_{\text{off}}}\right)$$
$$= \log\left((1+\beta)^{-d}\det\left((1-\alpha)V_{\pi_l^{(t)}} + \alpha V_{\pi_{\text{off}}} + \beta((1-\alpha)a_+ a_+^t + \alpha V_{\pi_{\text{off}}})\right)\right)$$
$$= -d\log(1+\beta) + d(\pi_l^{(t)}, \mathcal{A}_l, \alpha) + \log(\det(I + \beta H(\pi_l^{(t)})((1-\alpha)a_+ a_+^t + \alpha V_{\pi_{\text{off}}})))$$

Now the log-determinant in last term above can be re-written using the eigenvalues $\lambda_k$ of $H(\pi_l^{(t)})((1-\alpha)a_+ a_+^t + \alpha V_{\pi_{\text{off}}})$ as:

$$\log(\det(I + \beta H(\pi_l^{(t)})((1-\alpha)a_+ a_+^t + \alpha V_{\pi_{\text{off}}}))) = \sum_{k=1}^{d} \log(1 + \beta\lambda_k).$$

We observe that as the eigenspectrum of $H(\pi_l^{(t)})((1-\alpha)a_+ a_+^t + \alpha V_{\pi_{\text{off}}})$ is the same as the positive-semidefinite matrix $H^{1/2}(\pi_l^{(t)})((1-\alpha)a_+ a_+^t + \alpha V_{\pi_{\text{off}}})H^{1/2}(\pi_l^{(t)})$ we can conclude that $\lambda_k \ge 0$ for all $k \in [d]$.

Now using Lemma 1 in Merhav (2022) (we set $f(x) = ln(1 + \beta x)$, $a = \sum_k \lambda_k$ and $\mu = \frac{\sum_k \lambda_k}{d}$ in their Lemma 1), which is a *reverse* Jensen type inequality, to get

$$\log(\det(I + \beta H(\pi_l^{(t)})((1-\alpha)a_+ a_+^t + \alpha V_{\pi_{\text{off}}}))) = \sum_{k=1}^{d} \log(1 + \beta \lambda_k)$$

$$\geq \log\left(1 + \beta \sum_{k=1}^{d} \lambda_k\right).$$

But as

$$\log\left(1 + \beta \sum_{k=1}^{d} \lambda_k\right) = \log\left(1 + \beta Tr\left(H(\pi_l^{(t)})((1-\alpha)a_+ a_+^t + \alpha V_{\pi_{\text{off}}})\right)\right)$$

$$= \log(1 + \beta w_{a_+}),$$

we have:

$$d(\pi_l^{(t+1)}, \mathcal{A}_l, \alpha) - d(\pi_l^{(t)}, \mathcal{A}_l, \alpha) \geq -d\log(1 + \beta) + \log(1 + \beta w_{a_+}).$$

Hence:

$$d(\pi_l^{(t+1)}, \mathcal{A}_l, \alpha) - d(\pi_l^{(t)}, \mathcal{A}_l, \alpha) \geq (d-1)\log\left(\frac{(d-1)w_{a_+}}{d(w_{a_+} - 1)}\right) + \log\left(\frac{w_{a_+}}{d}\right). \tag{40}$$

Recall that:

$$\delta(\pi_l^{(t)}) := \frac{w_{a_+}(\pi_l^{(t)})}{d} - 1.$$

Using this the inequality in equation (40) is re-written to get

$$d(\pi_l^{(t+1)}, \mathcal{A}_l, \alpha) - d(\pi_l^{(t)}, \mathcal{A}_l, \alpha) \geq \log(1 + \delta(\pi_l^{(t)})) - (d-1)\log\left(1 + \frac{\delta(\pi_l^{(t)})}{(d-1)(1 + \delta(\pi_l^{(t)}))}\right)$$

$$\geq \log(1 + \delta(\pi_l^{(t)})) - \frac{\delta(\pi_l^{(t)})}{1 + \delta(\pi_l^{(t)})}$$

$$:= m(\delta(\pi_l^{(t)}))$$

This completes the proof of Lemma 5.4. $\qquad\square$

Now $m(\delta)$ satisfies the following simple properties (see lemma 3.6 in Todd (2016) for a proof)

**Lemma B.9.** *We have*

1. $m(\delta)$ *is increasing if* $\delta \geq 0$ *and decreasing if* $\delta < 0$.

2. *for* $\delta \geq \delta_0$, $m(\delta) \geq \left(1 - \frac{\delta_0}{(1+\delta_0)(ln(1+\delta_0))}\right) ln(1+\delta)$.

3. *for* $|\delta| \leq 1/2$, $m(\delta) \geq 2/7\delta^2$.

## B.5 Proof of Proposition 5.5.

*Proof.* Consider an iterate $\pi_l^{(t)}$ such that $\delta(\pi_l^{(t)}) \geq 1$. Set $\gamma_t = d(\pi_{l,\text{on}}^*, \mathcal{A}_l, \alpha) - d(\pi_l^{(t)}, \mathcal{A}_l, \alpha)$. Then

$$\gamma_t - \gamma_{t+1} = d(\pi_l^{(t+1)}, \mathcal{A}_l, \alpha) - d(\pi_l^{(t)}, \mathcal{A}_l, \alpha)$$

$$\geq \log(1 + \delta(\pi_l^{(t)})) - \frac{\delta(\pi_l^{(t)})}{1 + \delta(\pi_l^{(t)})}$$

$$\geq \left(1 - \frac{\delta_0}{(1+\delta_0)(ln(1+\delta_0))}\right) \log(1 + \delta(\pi_l^{(t)}))$$

$$\geq \left(\frac{1}{d} - \frac{\delta_0}{d(1+\delta_0)(ln(1+\delta_0))}\right) \gamma.$$

The first inequality follows from Lemma 5.4, the second inequality from Lemma B.9 (2), and the third from proposition B.5. We set $k(\delta_0) := \left(1 - \frac{\delta_0}{(1+\delta_0)(ln(1+\delta_0))}\right)$. We have:

$$\gamma_{t+1} \leq \left(1 - \frac{k(\delta_0)}{d}\right)\gamma_t \leq exp\left(-\frac{k(\delta_0)}{d}\right)\gamma_t.$$

Since the initialization has an upper bound of $5d\,ln(d) - d\,ln(1-\alpha)$ then within

$$\frac{d}{k(\delta_0)}ln\left(\frac{d}{\delta_0}ln\left(\frac{d^5}{1-\alpha}\right)\right)$$

iterations $\gamma_t$ is at most $\delta_0$. $\qquad \square$

## B.6 Proof of Proposition 5.7.

Since $\pi_l^{(t)}$ satisfies $\delta(\pi_l^{(t)}) \leq \frac{d_{\text{eff}}}{d}$ we have by definition of $\delta$ that:

$$(1-\alpha_l)||a||^2_{V^{-1}_{\tilde{\pi}_l^{(t)}}} + \alpha_l \sum_a \pi_{\text{off}}(a)||a||^2_{V^{-1}_{\tilde{\pi}_l^{(t)}}} \leq d(1+\frac{d_{\text{eff}}}{d}).$$

for each $a \in \mathcal{A}_l$. We now observe that similar to the proof of Lemma 4.4 we have:

$$d - \alpha_l \sum_a \pi_{\text{off}}(a)||a||^2_{V^{-1}_{\tilde{\pi}_l^{(t)}}} = Tr\left(\left(I + \frac{\alpha_l}{1-\alpha_l}V_{\pi_{\text{off}}}V^{\dagger}_{\pi_l^{(t)}}\right)^{-1}\right)$$

$$\leq d_{\text{eff}}.$$

This gives us the bound:

$$(1-\alpha_l)||a||^2_{V^{-1}_{\tilde{\pi}_l^{(t)}}} \leq 2d_{\text{eff}}$$

for each $a \in \mathcal{A}_l$ and completes the proof of the lemma.

## B.7 Proof of Theorem 5.8.

*Proof.* The proof is quite similar to the proof of Theorem 4.5. The only difference is we allow FW to solve only up to $\delta = \frac{d_e}{d}$ instead of setting $\delta = 0$. The upper bound on the confidence width is supplied by Proposition 5.7 instead of Lemma 4.4. Now using much the same techniques to arrive at equation (28) we get the following regret bound:

$$\mathcal{R}(\texttt{OOPE-FW}) \leq vT + 48d_{\text{eff}}\log(4l^2_{max}|\mathcal{A}|T)\left(\sum_{l=1}^{l_M}2^l\right) + \sum_{l=1}^{l_M}8\epsilon_l(|\text{supp}(\pi_l^{(t)})|) \qquad (41)$$

Now from Theorem 5.5 we know that $|\text{supp}(\pi_l^{(t)})|$ is upper bounded by $4d\log(d\log(\left(\frac{d^5}{1-\alpha}\right))) + d + \frac{28d}{\delta}$. Using this bound instead of $d(d+1)/2$ bound we get in much the same way as Step 5 in Theorem 4.5's proof:

$$\mathcal{R}(\texttt{OOPE-FW}) \leq vT + 96d_{\text{eff}}\log(4l^2_{max}|\mathcal{A}|T)\min\left\{\frac{8}{v}, \sqrt{4 + \frac{T}{d_{\text{eff}}\log(4|\mathcal{A}|T)}}\right\} + 8d + \frac{224d^2}{d_{\text{eff}}}$$

$$+ 32d\log\left(d\log\left(\left(\frac{d^5(T+T_{\text{off}})}{T}\right)\right)\right)$$

Now optimizing over $v$ we get that:

$$\mathcal{R}(\texttt{OOPE-FW}) \leq 32\sqrt{3d_{\text{eff}}T\log(4l^2_{max}|\mathcal{A}|T)} + 8d + \frac{224d^2}{d_{\text{eff}}} + 32d\log\left(d\log\left(\left(\frac{d^5(T+T_{\text{off}})}{T}\right)\right)\right)$$

and this concludes the proof. $\qquad \square$

