# OpenReview forum: "Regret minimization in Linear Bandits with offline data via extended D-optimal exploration."
_TMLR — Accepted by TMLR_

### Review · Reviewer_Wycu · 2025-12-20

**Summary Of Contributions:**

This paper studies stochastic linear bandits when the learner has access to an offline dataset collected before the online interaction. The central idea in the proposed algorithm Offline-Online Phased Elimination (OOPE) is to integrate the offline information into a phased-elimination framework by using an “extended” D-optimal experimental design: in each exploration phase, the method chooses an online sampling distribution that maximizes the log-determinant of a Gram matrix formed by mixing the offline Gram matrix with the phase’s online design.

The analysis introduces an effective dimension $d_\text{eff}$ (driven by the offline Gram eigen-spectrum / offline coverage) and derives a regret guarantee that smoothly interpolates between (i) the standard pure-online regret when offline data is uninformative, and (ii) substantially smaller regret when the offline data is abundant and well-conditioned. The paper also develops minimax-style lower bounds for problem classes parameterized by the normalized eigen-spectrum of the offline Gram matrix, and shows matching (up to logs/additive terms) upper/lower bounds in regimes where the offline data is either well explored or severely under-explored. Finally, to address the potentially large $O(d^2)$ additive term coming from the support size of exact D-optimal designs, the paper proposes a Frank–Wolfe approximation (OOPE-FW) that improves the dimension-dependent additive term in certain regimes.

Key strengths:
The paper is well written and easy to follow, and it studies an interesting problem. It proposes novel algorithms with theoretical guarantees that explicitly quantify how the quality of offline data affects regret. In addition, the paper derives lower bounds tailored to the offline–online setting, providing a sanity check and clarifying when performance improvements are fundamentally possible. The Frank–Wolfe–based variant is a valuable addition, particularly for high-dimensional settings or regimes where support size is a critical concern.

Weaknesses:
- The theoretical analysis assumes that the offline data are collected using a fixed (non-adaptive) design and that there is no distribution shift between the offline and online phases. As acknowledged by the paper, these are strong assumptions that limit the algorithm’s practical applicability.
- There remains a gap between the upper and lower bounds in regimes where only a small number of directions are poorly explored (i.e., moderately explored regimes).
- The analysis avoids reusing offline samples across phases in order to maintain independence. This choice is conservative and differs from how the method would likely be implemented in practice.
-  The empirical evaluation is brief and primarily illustrates the qualitative benefits of increasing the amount of offline data, without providing broader comparisons against strong baselines (although a few basic ones are included).

**Audience:**

Yes

**Audience Explanation:**

The offline-to-online setting for bandits is broadly relevant to modern ML workflows where large logged datasets exist and online interaction is expensive. The results should be of interest to readers working on bandits/contextual bandits, offline RL to online fine-tuning, exploration under logging policies, and connections between optimal experimental design and online learning.

**Broader Impact Concerns:**

I see no impact concern.

**Claims And Evidence:**

Yes

**Claims Explanation:**

The submission’s main claims are theoretical (algorithm design + regret upper bounds + minimax lower bounds), and the paper states these results precisely and provides detailed proofs in the appendix. The dependence of the regret on an effective dimension that captures offline data coverage is well-motivated in the analysis and is reflected in the stated bounds, including the reduction to the standard pure-online rate when offline data is not informative.

In addition, the minimax lower bounds parameterized by the normalized offline eigen-spectrum are a strong part of the paper: they help justify that improvements beyond the classic $O(d \sqrt T )$ behavior are only possible when the offline data meaningfully covers the action space, and they delineate regimes where the proposed algorithm is (near-)optimal.

For the experimental section, it provides qualitative evidence consistent with the theory (more offline samples and broader offline support reduce online regret), but it is relatively limited and would be more convincing with stronger baselines and a more extensive study. This affects the strength of the empirical evidence, but does not undermine the core theoretical claims.

**Requested Changes:**

- Please provide more guidance and intuition on $d_{\text{eff}}$. Its current definition is somewhat involved, and it would be helpful to include a short paragraph explaining how to interpret $d_{\text{eff}}$ in practice and how it might be estimated.
- If possible, please add a brief discussion on how to make the method horizon-free (e.g., via a doubling trick), since the algorithm currently relies on knowing $T$ in advance for parameter selection.
 - The paper identifies a gap between upper and lower bounds when only a few directions are poorly explored. Even if closing this gap is beyond the scope of the current work, it would be valuable to have a clearer discussion of its underlying causes and whether it is believed to stem from analytical looseness or to reflect a fundamental limitation.

Expand the empirical evaluation:
- Include comparisons against strong baselines, such as pure online phased elimination, to more clearly highlight the benefit of leveraging offline data, as well as any relevant recent offline-online baselines if they are applicable.
- Consider adding plots that compare OOPE and OOPE-FW to illustrate when the Frank-Wolfe variant is advantageous (e.g., as the dimension $d$ increases or when the $O(d^2)$ term becomes dominant).


Minor:
- Potential typo: Figure 3 uses  OOPEFW, but should likely be  OOPE-FW.

---

> ### Author Response · Authors · 2026-03-23
> **Response to Reviewer Wycu**
>
> We thank the reviewer for carefully engaging with the work. Please find our response to the points raised:
>
> 1) **Non-adaptive offline design**: We agree it is important to extend to cases where the offline data has been collected adaptively. However, this necessarily complicates the analysis, since the input datum must also explain "how" this offline data was collected. Hence, we chose to focus on non-adaptive offline designs and leave this important extension for future work.
>
> 2) **Reusing offline samples**: We have chosen to chunk the offline data across various faces instead of reusing all of it for each phase. This ensures we can derive concentration results that are crucial for the functioning of OOPE. With reuse, we are not able to derive similar concentration results. For our empirical implementation, we have faithfully reproduced the same pseudocode we show in the paper. However, we have experimented with reused offline data, and the performance was quite uneven; sometimes it outperforms OOPE, and at other times incurs much more regret. We attribute this unevenness to the lack of concentration guarantees.  We also mention that in the minimax optimal sense, OOPE already is provably optimal in many regimes, and hence, reuse is unlikely to significantly improve the theoretical guarantees in such scenarios.
>
> 3) **$d_{eff}$ Intuition and Estimation**: $d_{eff}$ represents the effective reduced dimension the algorithm has left to explore given the offline data. So if the offline data is plentiful and well-explored, then we expect it to be small, and when many directions are underexplored in offline data, we correspondingly expect it to be large. We can estimate the $d_{eff}$ from the knowledge of offline data (in particular $T_{off}$ and $V_{\pi_{off}}$) and the online horizon length $T$.
>
> 4) **Making OOPE Horizon-Free**: We can make the algorithm horizon-free by using a variant of the doubling trick with geometrically increasing episode horizons with full restarts. This variant, among others, is discussed in detail in the work of Besson and Kaufmann (2018). They show that this approach preserves the minimax $O(\sqrt{T})$ type bounds at the cost of absolute constant multipliers ($3.41$) to the regret bound. For an $i^{th}$ episode we set the offline fraction as $\alpha^{i}=\frac{T_{off}}{T^{i}+T_{off}}$ and estimate $d_{eff}$ based on this guessed horizon $T^{i}$ for this episode. We note that we are reusing the entire offline data for each episode. This preserves the eigenspectrum of the offline data across all episodes and is why we prefer the "full" restart variant of the doubling trick.
>
> 5) **Gap between upper and lower regret bounds**: We believe the gap is purely analytical and essentially boils down to how we upper-bound the term $\lambda_{d}(V\_{\pi^{*}\_{l,on}})$  by $\max\_a ||a||^2$ in proof of Lemma 4.4 in Appendix A.4. This bound does not reflect the dependence on the offline eigenspectrum and is just a uniform bound. This bound can be tight when the offline data has many underexplored directions, but it is quite weak in more well-explored settings. Improving this bound, to incorporate the offline spectrum in a more nuanced way, is challenging, and we leave it as further work.
>
> 6) **Comparisons with other Experimental Baseline**: In Figure 1, we update it by plotting the regret of pure online phased elimination that ignores the offline data. This shows how significantly OOPE outperforms vanilla pure phased elimination in the OO setting.  Most of the approaches proposed in the literature for OO setting in linear bandits involve warm-starting LinTS (Hamidi and Bayati, Oetomo) and LinUCB (Valko et al). We have already shown in Figure 2 (and theoretically in Section 4.3)  that we are comparable and at times outperform these benchmarks.
>
> 7) **Regimes where OOPE-FW outperforms OOPE**: We added several experiments demonstrating when OOPE-FW outperforms OOPE and vice-versa. We observe that there is no difference for short horizons (due to no meaningful elimination phase) but for medium and longer horizons, depending on the value of the effective dimension, there are considerable differences. For large $d\_{eff}$, OOPE outperforms OOPE-FW. This is because, for poor quality offline data, we are essentially back in pure online regret setting, where, in terms of regret, OOPE outperforms because of more precise exploration. For moderate $d_{eff}$, OOPE-FW outperforms OOPE and is due to the better support terms of $O(d^2/d\_{eff})$.

---

> ### Author Response · Authors · 2026-03-23
> **contd. response to Reviewer Wycu**
>
> In response to various reviewers' comments, we have made a number of changes to the paper. The following changes are pertinent for your comments:
>
> $\underline{\text{Point 3)}}:$  We have provided a paragraph mentioning in detail the aforementioned intuition and estimation for $d_{eff}$ in bottom of page 7. We also provide examples where we calculate the $d_{eff}$ for specific action-sets and offline data distribution.  We hope this clarifies the notion of $d_{eff}$ and why it plays an important role in our analysis.
>
> $\underline{\text{Point 4)}}:$  We add Remark 4.7 in bottom of page 11 describing how to make OOPE horizon free.
>
> $\underline{\text{Point 5)}}:$ We add Remark 4.12 explaining in detail why we believe the gap between upper and lower bounds is only analytical and stems from a potentially loose bound we employ in Lemma 4.4.
>
> $\underline{\text{Point 6)}}:$ We updated Figure 1 with pure online phased elimination to benchmark the benefits obtained by employing OOPE in the OO setting.
>
> $\underline{\text{Point 7)}}:$  We have added an experimental section where we extensively compare OOPE across varying $d,T, \lambda(V\_{\pi\_{off}})$ and $d\_{eff}$ whose parameters are given in Table 5 and results in Figure 4. We give a detailed discussion on the obtained results.

---

### Review · Reviewer_MUCV · 2026-01-02

**Summary Of Contributions:**

This paper studies online RL in linear bandits setup with additional access to offline data. The authors propose the notion of "effective dimension" to characterize the under-explored dimensions given the offline data. They contribute the OOPE algorithm, which leverages extended D-optimal design to incoroperate the offline information. They derive regret bound for OOPE and also establish lower bound. Besides, they further improve the $O(d^2)$ term by using Frank-Wolfe when solving the optimization problem in OOPE. Lastly, empirical evaluations are provided.

**Additional Comments:**

Assumption 3.4 seems a bit unnatural to me. It may fail to hold when $|\mathcal{A}|$ is larger than $T_{off}$, since $\min_a \pi_{off}(a) \leq 1/|\mathcal{A}|$. Although the authors discuss several ways to "get rid of" it, I'm wondering if there would be better ways to "fix" it?

For example, I guess one possible way is to utilize a "truncated" version of $\pi_{off}$ as the offline policy to leverage, by dropping all the actions with limited coverage by $\pi_{off}$ (and do re-normalization). But I'm not sure whether this can work all the time (maybe not a good idea when $\pi_{off}$ is close to uniform).

**Audience:**

Yes

**Audience Explanation:**

Linear bandit is a classical and important topic in this community. Besides, utilizing offline data to accelerate online learning is a practical scenario and would be interesting to the TMLR's audience.

**Broader Impact Concerns:**

N.A.

**Claims And Evidence:**

Yes

**Claims Explanation:**

The setting, assumptions and algorithms are clearly stated. The presented results look good to me.

**Requested Changes:**

Overall, I found the paper already in good shape. Maybe the authors can improve it by fixing typos if any.

---

> ### Author Response · Authors · 2026-03-23
> **Response to Reviewer MUCV**
>
> We thank the reviewer for engaging with the work. Please find below our response to point raised by you:
>
> 1) **Getting rid of Assumption 3.4**: We agree that this assumption, although mild in practice, is unnatural. We have observed that in offline distributions, violating this assumption, the empirical runs are less stable. We think this is because the concentration inequalities no longer hold, and the behaviour expected on good events is no longer true. We have tried hard to remove this assumption, but have not succeeded. The main challenge has always been that the number of arms that violate this assumption can be quite large. We believe this assumption is not necessary since the strictly needed samples (without the ceil operation) all satisfy the offline samples without an issue. Maybe we need to build on more sophisticated rounding procedures for D-optimal design found in Pukelsheim (2006) and Fiez et al. (2019), but we were unable to do so.

---

### Review · Reviewer_JfdN · 2026-03-08

**Summary Of Contributions:**

This paper studies stochastic linear bandits with access to offline data, aiming to reduce online regret with some prior collected observations. This work proposed an algorithm named OOPE, which uses an extended D-optimal design to leverage offline data and achieves a decent regret bound. This work also shows that the regret bound matches minimax lower bounds depending on the offline data quality based on effective dimension. High-dimenion regime is also studied based on a Frank-Wolfe variant of OOPE.

I feel this work's theoretical results are complete and quite comprehensive, and I can not find any clear flaw of this work. It is interesting to see that the authors deduce both upper bounds and lower bounds and they match each other, which strenthen the contribution of this work.

For experimental results, maybe it is better to add some real-world setups.

**Audience:**

Yes

**Audience Explanation:**

This work studies stochastic linear bandit with access to offline data, which is relevant to TMLR's topics.

**Broader Impact Concerns:**

No ethical or social impact concern as this work is mainly theoretical.

**Claims And Evidence:**

Yes

**Claims Explanation:**

This work provides substantial related work discussion and theoretical deduction. Although I did not check all the details in the Appendix, but the theoretical results look reasonable and strong to me. This paper also provides a suite of experimental results to support their findings.

**Requested Changes:**

I think it would be better if some real-world data experiments can be used in this work.

---

> ### Author Response · Authors · 2026-03-23
> **Response to Reviewer JfdN**
>
> We thank the reviewer for engaging with the work. Please find our response to the point raised by you:
>
> 1) **Adding real-world experiments**: Our work primarily makes theoretical contributions to regret minimisation in linear bandits. We have empirically validated our theoretical claims through synthetic experiments. The ideas from Linear Bandits often guide in designing algorithms for more practical scenarios like contextual bandits. We find that most papers presenting real-world data simulations are for the specific case of contextual bandits in areas like recommendations and online advertising.  In these applications, contextual information is crucial, something our setting does not incorporate. Further to validate the $\tilde{O}(\sqrt{d\_{eff}T})$ style rates requires control over the offline data generation, which is possible in synthetic data but not in a fixed real-world dataset. Almost all prior work on Offline-to-Online Linear Bandits and MABs, like Cheung & Lyu (2024) and Joachims & Shivasamy (2012) have only provided experiments with synthetic data.

---

### Author Response · Authors · 2026-03-23
**Revised paper incorporating feedback from reviewers**

We have made several changes incorporating the valuable feedback given by the reviewers:

$\underline{\text{Intuition for $d_{eff}$}}:$  We have provided a paragraph mentioning in detail the aforementioned intuition and estimation for $d_{eff}$ in bottom of page 7. We also provide examples where we calculate the $d_{eff}$ for specific action-sets and offline data distribution.  We hope this clarifies the notion of $d_{eff}$ and why it plays an important role in our analysis.

$\underline{\text{Making OOPE horizon-free}}:$  We add Remark 4.7 in bottom of page 11 describing how to make OOPE horizon free.

$\underline{\text{Gap in upper and lower bounds}}:$ We add Remark 4.12 explaining in detail why we believe the gap between upper and lower bounds is only analytical and stems from a potentially loose bound we employ in Lemma 4.4.

$\underline{\text{Benchmark against pure online phased elimination.}}:$ We updated Figure 1 with pure online phased elimination to benchmark the benefits obtained by employing OOPE in the OO setting.

$\underline{\text{Experimental results comparing OOPE and OOPE-FW}}:$  We have added an experimental section where we extensively compare OOPE across varying $d,T, \lambda(V\_{\pi\_{off}})$ and $d\_{eff}$ whose parameters are given in Table 5 and results in Figure 4. We give a detailed discussion of the obtained results.

We request the reviewers and editors to please consider these changes when deciding on this submisison.

---

### Decision · Action_Editor_n2s3 · 2026-04-07

**Recommendation:** Accept as is

**Audience:**

Yes

**Audience Explanation:**

As acknowledged by the reviewers, utilizing offline data to accelerate online linear bandit learning is a practical setting. Ideally the paper could be evaluated in real-world datasets, but as the authors acknowledged, this requires a contextual bandit setting which is beyond the scope of the current paper.

**Claims And Evidence:**

Yes

**Claims Explanation:**

The reviewers find the paper technically solid: the proved upper and lower bounds match in a broad regime, and can improve over the baseline of using online learning-only. Experiments provide qualitative evidence consistent with the theory.